# Longitudinal population-level HIV epidemiologic and genomic surveillance highlights growing gender disparity of HIV transmission in Uganda

HIV incidence in eastern and southern Africa has historically been concentrated among girls and women aged 15–24 years. As new cases decline with HIV interventions, population-level infection dynamics may shift by age and gender. Here, we integrated population-based surveillance of 38,749 participants in the Rakai Community Cohort Study and longitudinal deep-sequence viral phylogenetics to assess how HIV incidence and population groups driving transmission have changed from 2003 to 2018 in Uganda. We observed 1,117 individuals in the incidence cohort and 1,978 individuals in the transmission cohort. HIV viral suppression increased more rapidly in women than men, however incidence declined more slowly in women than men. We found that age-specific transmission flows shifted: whereas HIV transmission to girls and women (aged 15–24 years) from older men declined by about one-third, transmission to women (aged 25–34 years) from men that were 0–6 years older increased by half in 2003 to 2018. Based on changes in transmission flows, we estimated that closing the gender gap in viral suppression could have reduced HIV incidence in women by half in 2018. This study suggests that HIV programmes to increase HIV suppression in men are critical to reduce incidence in women, close gender gaps in infection burden and improve men's health in Africa.

Despite the widespread availability of human immunodeficiency virus (HIV) prevention and treatment interventions, there were 1.5 million new HIV infections and 680,000 HIV-associated deaths in 2020[1]. More than half of these new cases and deaths were concentrated in the eastern and southern regions of the African continent, where incidence rates have historically been highest in adolescent girls and young women, aged 15–24 years[2–5]. Although HIV incidence has declined by 43% in eastern and southern Africa since 2010, current HIV service programmes are failing to reduce new cases rapidly enough to meet United Nations health targets for HIV epidemic control[1]. With rising levels of HIV drug

resistance[6,7] and flatlined global investment in HIV control[8], the African HIV epidemic has reached a critical inflection point[9].

Over the past decade, African HIV control programmes, including the US President's Emergency Plan for AIDS Relief (PEPFAR), have focused on expanding treatment coverage in people with HIV and reducing HIV infections among adolescent girls and young women[10,11]. However, recent data from Africa indicate that the mean age of infection is shifting[12,13] and incidence rates are declining faster in men than in women[14,15], suggesting that the age and gender structure of the African HIV epidemic is evolving. Here, we integrate 15 years of data on

✉e-mail: jkagayi@rhsp.org; mgrabow2@jhu.edu; oliver.ratmann@imperial.ac.uk

**Fig. 1 | Time trends in age-specific HIV incidence rates for men and women in Rakai, Uganda. a**, Location of RCCS in south-central Uganda. Study outcomes are reported for all RCCS communities located inland to Lake Victoria across nine survey rounds. **b**, Number of RCCS participants in the census-eligible population of age 15–49 years by survey round. **c**, Estimated mean HIV incidence rates per 100 PY of exposure in uninfected individuals (line) by 1-year age band, gender (colours) and survey round, along with 95% confidence intervals (ribbon), and median age of incident cases (arrowhead). **d**, Estimated median contribution to incidence cases in the study population (line) by 1-year age band, gender (colours) and survey round, along with 95% confidence intervals (ribbon). Throughout all subfigures, incidence estimates are based on $n = 1,117$ individuals in the incidence cohort. Basemap in **a** from OpenStreetMap under a Creative Commons license CC BY-SA 2.0.

HIV incidence and onward transmission to show how the drivers of the heterosexual African HIV epidemic are changing by age and gender. We focus on a study population aged 15 to 49 years with an HIV risk profile typical across eastern and southern Africa[16,17], living in 36 semi-urban and rural agrarian communities that are part of the population-based Rakai Community Cohort Study (RCCS) in south-central Uganda[18] (Fig. 1a). We followed individuals in the RCCS who were HIV seronegative and documented new infection events. We also deep sequenced HIV virus longitudinally from people with viremic HIV. This enabled us to infer directed transmission networks across age and gender[19,20], and

investigate the time trends in infection dynamics and transmission networks during mass scale-up of HIV services in Africa[1].

## Results

### HIV incidence is declining faster in men than women

From 23 September 2003 to 22 May 2018, 38,749 participants were enrolled in RCCS[14]. Of these participants, 22,724 tested HIV seronegative at first survey, and contributed an estimated 127,217 person-years (PY) of follow-up (Fig. 1b, Supplementary Tables 1 and 2). Study participants were enrolled following population census, household enumeration

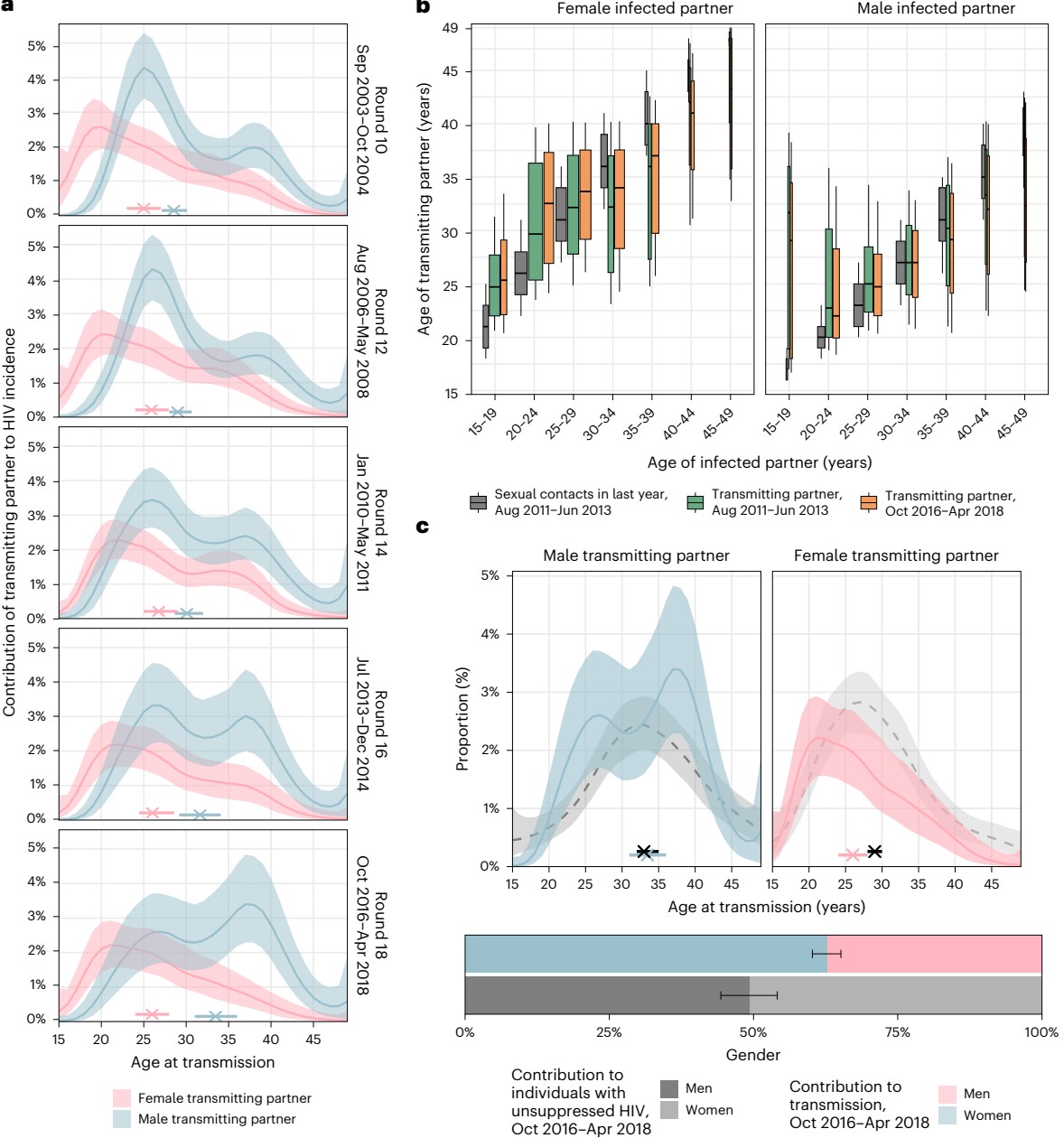

**Fig. 2 | Time trends in age-specific sources of HIV infections in women and men. a**, Estimated age distributions of transmitting partners (posterior median: line; 95% credible interval: ribbon), along with the median age at transmission (posterior median: cross; 95% credible interval: bar). Age contributions sum to 100% for each round, summing over men and women. **b**, Estimated age distributions of transmitting partners by 5-year age bracket of infected partners (posterior median: thick black bar in boxplots; 50% interquartile range: height of box; 80% credible intervals: whiskers in boxplots). The width of the boxplots is proportional to the total infections in each recipient group. For reference, posterior estimates of the age distributions of sexual contact partners of men and women by 5-year age bands in the past 12 months in the same communities are shown in dark grey (estimates visualized in the same manner). **c**, Comparison of the age contributions to transmitting partners (colour) to the age contributions to men and women with unsuppressed HIV (posterior median: dashed black line; 95% credible interval: ribbon), along with median age (posterior median: cross; 95% credible interval: bar). Age contributions sum to 100% for men and women combined. Throughout all subfigures, transmission flow estimates are based on n = 227 heterosexual source–recipient pairs identified among n = 1,978 individuals in the transmission cohort and n = 1,117 individuals in the incidence cohort.

and informed consent in nine survey rounds of approximately 18 months' duration, herein denoted as survey rounds 10–18 (Methods and Extended Data Fig. 1).

In total, we observed 1,117 incident HIV infections (Supplementary Tables 3 and 4 and Extended Data Fig. 2). Figure 1c shows that incidence rates among men in inland communities fell rapidly from 1.05 (95% confidence interval (CI) 1.03–1.08) per 100 PY in 2003 (survey round 10) by 67.8% (66.2–69.2) to 0.34 (0.33–0.35) per 100 PY in 2018 (survey

round 18), with no substantial shift in the median age of male incident infection (blue arrowheads in Fig. 1c). In young women aged 15–24 years, incidence rates fell similarly rapidly from 1.42 (1.35–1.5) per 100 PY in 2003 by 74.5% (71.6–77.1) to 0.36 (0.33–0.40) per 100 PY in 2018. However, among women aged 25–34, declines in HIV incidence were substantially slower (from 1.51 (1.45–1.57) per 100 PY in 2003 by 43.9% (40.5–47.4) to 0.84 (0.80–0.89) per 100 PY in 2018), and similarly in women aged 35–49 (from 0.90 (0.85–0.94) per 100 PY in 2003 by

**Table 1 | HIV prevalence, viral suppression, transmission sources and impact of counterfactual interventions focused on closing the suppression gap in men by age of male partner, round 18, October 2016 to April 2018**

| Age | HIV prevalence in men (% in age bracket) | Men with HIV who have unsuppressed virus (% in age bracket) | Male–female difference in the proportion of individuals with HIV who have unsuppressed virus (difference) | Contribution of age group to all men with unsuppressed virus (%) | Contribution of age group to all transmitting male partners (%) | Closing half the suppression gap | | Closing the suppression gap | | 95-95-95 in men | |
|---|---|---|---|---|---|---|---|---|---|---|---|
| | | | | | | Contribution of age group to additional number of men with unsuppressed virus in counterfactual (%) | Predicted reduction in incidence in women in round 18 (% of actual incidence) | Contribution of age group to additional number of men with unsuppressed virus in counterfactual (%) | Predicted reduction in incidence in women in round 18 (% of actual incidence) | Contribution of age group to additional number of men with unsuppressed virus in counterfactual (%) | Predicted reduction in incidence in women in round 18 (% of actual incidence) |
| 15–19 | 0.8 | 73.2 | 36.8 | 5.3 | 1.4 | 5.3 | 21.9 | 5.3 | 43.7 | 7.4 | 72.2 |
| | (0.4–1.3) | (56.6–86.2) | (16.8–54.2) | (3.0–8.8) | (0.4–3.5) | (2.5–8.7) | (21.2–22.8) | (2.5–8.7) | (42.4–45.6) | (5.2–9.9) | (68.8–74.6) |
| 20–24 | 2.1 | 66.5 | 26.6 | 9.0 | 12.7 | 7.1 | 24.7 | 7.1 | 49.6 | 12.2 | 60.4 |
| | (1.6–2.7) | (55.5–76.6) | (14.0–38.2) | (6.5–11.8) | (7.5–19.2) | (3.7–10.9) | (23.2–26.2) | (3.7–10.9) | (46.6–52.7) | (9.4–15.6) | (55.6–65.0) |
| 25–29 | 6.2 | 53.5 | 23.2 | 17.9 | 20.2 | 15.0 | 25.1 | 15.0 | 50.5 | 22.7 | 58.4 |
| | (5.2–7.3) | (45.3–61.4) | (13.7–32.6) | (14.6–21.5) | (13.8–27.5) | (9.1–21.5) | (23.9–26.6) | (9.1–21.5) | (47.9–53.5) | (18.2–28.1) | (54.0–62.4) |
| 30–34 | 12.5 | 39.6 | 19.2 | 24.3 | 19.4 | 21.3 | 25.2 | 21.3 | 50.7 | 26.9 | 58.7 |
| | (10.8–14.2) | (33.6–45.8) | (12.1–26.1) | (20.8–28.0) | (13.3–27.3) | (13.7–27.7) | (23.6–27.0) | (13.7–27.7) | (47.4–54.3) | (22.2–31.5) | (53.0–64.0) |
| 35–39 | 16.4 | 28.9 | 17.6 | 21.3 | 25.8 | 24.0 | 26.9 | 24.0 | 54.3 | 18.6 | 52.9 |
| | (14.6–18.2) | (23.3–35.0) | (11.2–24.3) | (17.9–24.8) | (18.4–34.9) | (16.9–31.1) | (24.8–28.7) | (16.9–31.1) | (49.8–58.1) | (13.5–23.6) | (46.6–60.0) |
| 40–44 | 16.8 | 21.9 | 14.6 | 13.8 | 14.9 | 17.7 | 28.3 | 17.7 | 57.7 | 8.3 | 42.5 |
| | (14.8–18.9) | (16.3–28.2) | (8.5–21.5) | (10.6–17.1) | (9.2–21.8) | (10.9–23.9) | (25.8–30.0) | (10.9–23.9) | (52.4–61.1) | (2.6–13.4) | (35.2–51.9) |
| 45–49 | 16.4 | 19.4 | 12.2 | 8.2 | 4.4 | 9.6 | 27.1 | 9.6 | 56.1 | 3.7 | 40.3 |
| | (14.1–19.1) | (13.0–27.0) | (4.5–20.5) | (5.5–11.3) | (1.7–8.5) | (2.7–15.8) | (24.6–29.0) | (2.7–15.8) | (50.1–59.7) | (0.0–8.5) | (29.4–55.1) |
| Total | 8.0 | 33.9 | 14.8 | 100 | 100 | 100 | 25.1 | 100 | 50.6 | 100 | 58.4 |
| | (7.4–8.6) | (29.7–38.3) | (10.0–19.6) | | | | (24.2–26.2) | | (48.6–52.8) | | (54.9–61.7) |

37.4% (31.9–42.6) to 0.56 (0.52–0.60) per 100 PY in 2018, resulting in a progressive, substantial shift in the median age of infection in women from 23.4 (22.6–24.1) in 2003 to 28.2 (27.1–29.2) in 2018 (Fig. 1c,d). Progress in reducing HIV incidence thus continues to be substantially slower in women[14,15], especially among those aged 25 years and above.

### The proportion of transmission from men is increasing

To characterize the population transmission flows by age and gender that underly observed shifts in incidence, we deep sequenced virus from 1,978 participants with HIV[19] (Supplementary Table 5). By embedding genomic surveillance into a population-based cohort study, deep-sequence sampling coverage was high relative to typical pathogen sequencing studies, which is essential for reconstructing transmission events[20–24]. We characterized the phylogenetic ordering between multiple viral variants from individuals and estimated the direction of transmission with phyloscanner[21,25] (Methods). We identified 236 heterosexual source–recipient pairs that were phylogenetically close and exhibited, in combination with data on last negative and first positive tests, strongly consistent evidence of the direction of transmission (Methods and Extended Data Fig. 3). We further estimated the likely infection date from deep-sequence data[26], which enabled us to place the source–recipient pairs in calendar time and consider their age at the time of infection (Extended Data Fig. 4). Of the 236 heterosexual source–recipient pairs, we retained in total 227 pairs in whom transmission was estimated to have occurred during the study period.

Deep-sequence phylogenetics cannot prove direction of transmission between two persons[21], but in aggregate these data are able to capture heterosexual HIV transmission flows at a population level[20,27]. We estimated population-level transmission flows adjusting for detection probabilities with semi-parametric Poisson flow regression models[28], and under the constraint that the transmission flows needed to closely match the age- and gender-specific incidence dynamics shown in Fig. 1 (Methods, Extended Data Fig. 5 and Supplementary Table 6). The fitted model was consistent with all the available data (Extended Data Fig. 6). Figure 2a shows the age profile of the estimated male and female sources of infection, such that the male plus the female sources sum to

100% for each survey round. Overall, we found that the contribution of men to onward transmission increased progressively from 57.9% (56.2–59.6) in 2003 to 62.8% (60.2–65.2) in 2018, indicating that HIV transmission is now more disproportionately driven by men than has been the case previously.

### Transmissions from men are shifting to older ages

The age profile of the population-level sources of infection characterizes the major age groups that sustain transmission[29]. We find that the age of transmitting male partners progressively increased from a median age of 28.5 (27.1–30.1) years in 2003 to 33.5 (31.0–36.0) years in 2018 (Table 1 and Fig. 2a), and this increase in the age of transmitting male partners was largest in transmissions to women aged 20–24 years (Fig. 2b). By contrast, the median age of female transmitting partners remained similar (from 25.0 (23.0–27.0) years in 2003 to 26.0 (24.0–28.0) years in 2018), corresponding to our earlier observations that the age of male incident infections also remained similar during the observation period.

Over time, substantially fewer infections occurred in adolescent girls and young women aged 15–24 years. In 2003 the largest transmission flows were to women aged 15–24 years from male partners 0–6 years older (15.5% (12.3–18.9)) and from male partners more than 6 years older (16.0% (12.7–19.2); Supplementary Table 7). By 2018, these transmission flows declined by approximately one-third, with 8.1% (5.6–11.0) to women aged 15–24 years from male partners aged 0–6 years older, and 12.1% (9.3–15.2) to women aged 15–24 years from male partners aged more than 6 years older. In those infections in adolescent girls and young women that occurred in 2018, the median age difference between incident infections in adolescent girls and young women and their transmitting male partners was 9.0 (7.0–12.0) years (Fig. 2b and Supplementary Table 7), similarly as in a phylogenetic study from KwaZulu-Natal in South Africa[30]. This prompted us to estimate for comparison age-specific sexual contact patterns within RCCS communities (Methods and Supplementary Table 8). In 2018, the median age difference between adolescent girls and young women and their male sexual partners was 3.6 (3.5–3.9) years. Our data thus indicate that

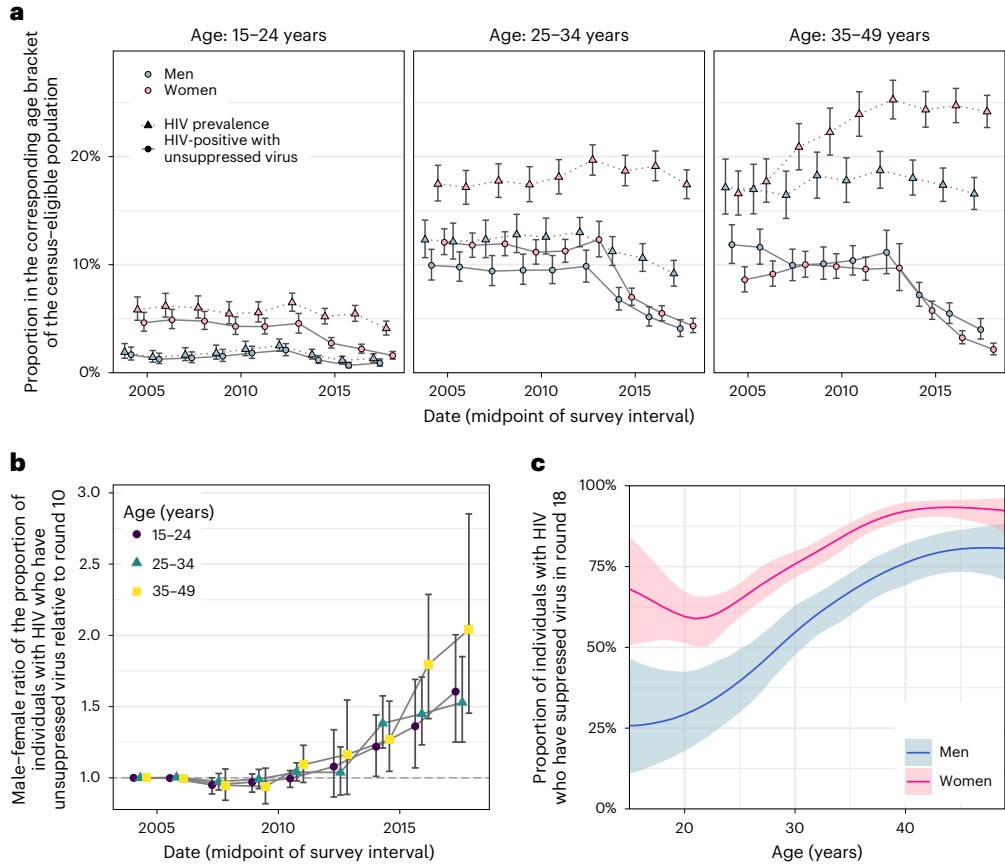

**Fig. 3 | Changes in population-level suppression of HIV viral load. a**, Estimated trends of HIV prevalence and the proportion of census-eligible individuals in three age brackets that remain virally unsuppressed, defined as viral load above 1,000 copies per ml of plasma (posterior median: dots; 95% credible interval: error bars), combining data from participants and from first-time participants as proxy of non-participants. **b**, Male-to-female ratio of changes in population-level viral load suppression relative to survey round 10 (posterior median: dots;

95% credible interval: error bars). **c**, Estimated viral suppression rates by 1-year age band (*x* axis) and gender (colour) for survey round 18 (posterior median: dots; 95% credible interval: error bars). Throughout all subfigures, estimates are based on data from *n* = 38,749 participants including *n* = 3,265 participants with HIV and with measured viral load. First-time participants were used as proxies of individuals who did not participate in the survey.

the main transmission flow into adolescent girls and young women is through contacts with considerably older men as compared to their typical sexual contacts[30,31], and that while this transmission flow has weakened overall, it remains the predominant mode of infection in adolescent girls and young women.

By 2018, the largest share of transmission flows shifted to women aged 25–34 years, from male partners 0–6 years older. In 2003, transmissions to women 25–34 years from these transmitting partners accounted for 7.7% (6.2–9.3) of all transmissions, and by 2018 the share of these flows increased by half to 12.0% (9.1–15.0; Supplementary Table 7). We also find that the transmission flows to women aged 35 years and above increased (Supplementary Table 7, also indicated by wider boxplots in Fig. 2b).

Our data suggest further deviations in age-specific transmission flows from the typical sexual contact patterns within study communities. For all women aged 30 years and older, we estimate their male transmitting partners were of similar age with a posterior interquartile age range of 30.3–38.0 years in 2018, whereas for comparison the typical sexual contact partners of these women were older with a posterior interquartile age range of 40.0–42.7 years. These findings explain the unexpected age profile of male transmitting partners (Fig. 3c) that concentrates in men aged 25–40 years instead of extending to progressively older men (Extended Data Fig. 7). Our observations are in line with recent studies from Zambia[20] and South Africa[32] that show having a male partner aged 25–40 years

rather than having an age gap between partners is associated with increased transmission risk.

The transmission flows into men remained similar over time (Fig. 2b). In 2018, the largest transmission flow was to men aged 25–34 years from transmitting female partners of similar age that were 0–6 years older (10.6% (8.9–12.3)).

## Gender gaps in viral suppression are increasing

We next placed the reconstructed shifts in transmission dynamics into the wider context of rapidly expanding HIV treatment during the observation period[14]. We measured viral load from 2011 (survey round 15) among almost all participants with HIV[33] (Supplementary Table 1 and Extended Data Fig. 8). Following World Health Organization (WHO) criteria[34], individuals with viral load measurements below 1,000 copies per ml of plasma were considered viremic (Methods and Supplementary Table 9). By 2018, we find that the proportion of men and women who were viremic was entirely decoupled from HIV prevalence in that while the proportion of women with HIV was substantially higher than in men, the proportion of viremic women was similar or lower than in men (Fig. 3a). We quantified these trends with the male-to-female ratio of the proportion of viremic individuals relative to 2003 levels, which has been progressively increasing in all age groups (Fig. 3b). This suggests[35] that faster rises in female HIV suppression could explain in part the faster declines in male incidence rates as higher rates of antiretroviral treatment (ART) uptake and virus

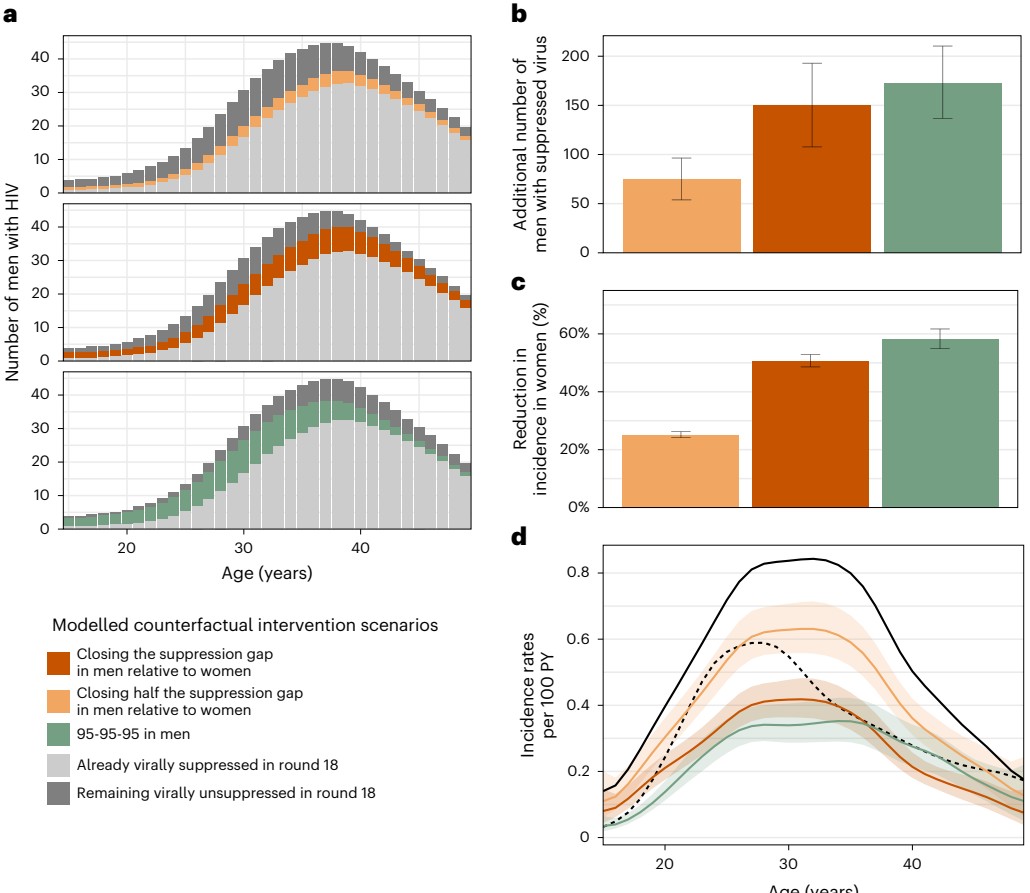

**Fig. 4 | Counterfactual modelling scenarios predicting the effect of interventions to increase HIV suppression in men on incidence reductions in women. a,b**, Estimated additional number of men with HIV in the census-eligible population in round 18 that already had suppressed virus (light grey), those who would have achieved viral suppression in the counterfactual intervention scenarios (colour), and those who would have remained with unsuppressed virus in the counterfactuals (dark grey) (posterior median: bars; 95% credible interval: error bars). **c**, Reduction in incidence in women of the census-eligible population in round 18 under the counterfactual targeted scenarios (posterior median: bars; 95% credible interval: error bars). **d**, Estimated incidence rates among women in the census-eligible population in round 18 (black solid line) and the counterfactual scenarios (colour), with incidence rates among men in round 18 shown as reference (black dashed line) (posterior median: lines; 95% credible intervals: ribbons). Throughout all subfigures, estimates are based on data from *n* = 15,053 participants in survey round 18, including *n* = 110 individuals in the incidence cohort in round 18, *n* = 432 individuals with HIV and with measured viral load in round 18, and *n* = 61 heterosexual source−recipient pairs in rounds 16−18, and information inferred through hierarchical models from all individuals in earlier rounds.

suppression in women mean that male partners are less likely to become infected, whereas men's higher rates of unsuppressed virus mean they are more likely to transmit to female partners (Extended Data Fig. 9). These trends have by 2018 accumulated to a substantial gap in suppression levels in men compared to women (Table 1 and Fig. 3c).

### Men contribute disproportionally to transmission

Combining phylogenetics with the virus suppression data also allowed us to compare transmission with population-level infectiousness as measured through viremic individuals (Table 1 and Fig. 2c). In 2018, the contribution of men to viremic individuals was 49.2% (44.3–54.1). For the same time period, we found that the contribution of men to transmission was consistently higher (62.8% (60.2–65.2)), indicating that men contribute more to transmission than population viral load suggests. These findings are compatible with generally higher viral load in men than women[33,36], which is expected to lead to higher transmission rates per sex act from men than women, heterogeneous contact patterns[37], higher biological susceptibility of women to HIV infection in heterosexual contacts[38,39], but also lower susceptibility of men to HIV infection following voluntary medical male circumcision[40].

### Policy implications

It has been previously demonstrated that people with HIV who are on ART and maintain suppressed virus do not transmit HIV[41,42]. On this basis, we quantified the effect that closing the gap in male−female virus suppression levels could have had on HIV transmission flows. Specifically, we parameterized the transmission flow model in terms of HIV seronegative individuals who are susceptible to infection and individuals with unsuppressed HIV who remain infectious. Thus, we were able to use the fitted model to estimate the impact of fewer individuals with unsuppressed HIV on evolving HIV transmission in counterfactual, modelled intervention scenarios (Methods). We considered the impact of three hypothetical scenarios: first, the impact of reducing by half the gap in the proportion of men with suppressed virus as compared to women ('closing half the suppression gap in men') at the end of the observation period in 2018 (Fig. 3c); second, the impact of achieving the same virus suppression levels in men with HIV as in women in 2018 ('closing the suppression gap in men'); and third−for reference−achieving the UNAIDS (Joint United Nations Programme on HIV/AIDS) 95-95-95 target that 86% of men (0.95 × 0.95 × 0.95) with HIV reach viral suppression in all age groups in 2018[43]. Table 1 and Fig. 4a describe the age-specific male counterfactual viral suppression targets

of each scenario, and place these into the context of prevalence, suppression and transmission. Overall, we found slightly older men would have reached suppression in the scenarios closing the suppression gap as compared to the UNAIDS 95-95-95 scenario. We predict that in the UNAIDS 95-95-95 scenario, an additional 172.6 (136.8–210.0) men with HIV would have reached viral suppression in 2018 (Fig. 4b) and this would have resulted in a 58.4% (54.9–61.7) additional reduction in HIV incidence in women in 2018 (Fig. 4c), which is in good agreement with the contribution of 95-95-95 interventions to projected incidence reductions for all of eastern and southern Africa under the mathematical models used to inform the global HIV prevention strategy[44]. In the scenario closing half the suppression gap in men, an additional 75.1 (53.9–96.0) men with HIV would have reached viral suppression in 2018 and resulted in a 25.1% (24.2–26.2) additional reduction in HIV incidence in women in 2018. In the scenario closing the entire suppression gap in men, an additional 150.2 (107.8–193.0) men with HIV would have reached viral suppression in 2018 and resulted in a 50.6% (48.6–52.8) additional reduction in HIV incidence in women in 2018 (Fig. 4b–c). Thus, all three intervention scenarios involved reaching a small additional number of men compared with the thousands of women with higher risk of HIV acquisition in the same rural and semi-urban study areas[45]. We predict that closing the suppression gap in men would have changed the female-to-male incidence rate ratio from 1.59 (1.38–1.82) to 0.78 (0.69–0.87) in 2018 (Fig. 4d), entirely closing the growing gender disparity in HIV incidence.

## Discussion

Effective HIV interventions and services are essential to bring most African countries on track to end AIDS as a public health threat by 2030 and accelerate progress towards the vision of the UNAIDS 'three zeros' target: zero new HIV infections, zero discrimination and zero AIDS-related deaths[44,46]. Gender inequalities are among the main reasons why global targets on mass scale-up of HIV testing, biomedical interventions and on incidence reductions have not been achieved[47]. Here, we combined population-based incidence with deep-sequence viral phylogenetic surveillance data to characterize how HIV incidence and heterosexual transmission sources have been changing by age and gender in a typical rural and semi-urban African setting. We show that along with increasing availability of HIV services, there have been consistently faster increases in viral suppression in women than in men and an increasing majority of new infections are arising from men. We also document substantial age shifts in HIV incidence and transmission sources, with the primary burden of incidence shifting to older women aged 25–34 years, the primary burden of transmission shifting to male partners aged 30–39 years, and the relative contribution of transmission flows to adolescent girls and young women from older men reducing by one-third. Modelling counterfactual improvements in HIV outcomes for men based on the inferred transmission flows during the last survey round in 2016–2018, we find that closing the male gender gap in viral suppression rates could have reduced incident female infections by half in that time period and brought about gender equality in HIV infection burden.

This study evaluated data from one longitudinal surveillance cohort in southern Uganda, but the increasing gender disparities and shifts in age-specific transmission are not unique. Incidence data published over the past decade documents widespread declining incidence across the African continent[17], greater differences in rates of new infections between men and women over calendar time, and rising average age of infection in women[17]. Data from population surveillance studies and HIV treatment and prevention trials shows higher levels of viremia among men compared to women with HIV[48,49], and phylogenetic studies from Botswana[50] and Zambia[20] also report gender disparities in HIV transmission. Together, these observations suggest that the principal characteristics of the evolving HIV epidemic likely apply more broadly in similar rural and semi-urban populations across eastern and southern Africa.

Given that the African HIV epidemic has historically been concentrated among adolescent girls and young women[4,5], programmes and policies rightfully have concentrated on reducing HIV risk in this demographic. Here, we document that most heterosexual transmission is driven by men and that—as incidence is declining—the contribution of men to onward heterosexual transmission is growing, likely due to slower population-level declines in HIV viremia in men. While there are emerging efforts to design male-centred HIV interventions[51,52], African men continue to be overlooked in the design of programmatic services[53,54]. Many factors, including gender norms, mobility and lack of targeted programming to men contribute to lower uptake of HIV services by men[52]. Case finding of men with HIV might be difficult but could be strengthened by expanding access to HIV testing services most likely to reach them, such as through self-testing or assisted partner notification and other social network strategies[53,55,56]. Retention of men with HIV in treatment and care programmes could be improved through male-centred differentiated service delivery. It is well established that improving male engagement in HIV services leads to better health for men[57,58]. We expect additional interventions such as voluntary medical male circumcision, condom promotion or pre-exposure prophylaxis would lead to further reductions in new cases[59].

Our findings are grounded in 15 years of consecutive population-based epidemiologic and molecular surveillance in southern Uganda, enabling us to measure changes in HIV incidence and transmission during a critical period of HIV service scale-up. Though it is typically assumed that age-specific patterns in onward HIV transmission correspond to those of viremia or follow typical sexual contact patterns, we find that this is not always the case. First, men contributed disproportionally more to onward heterosexual transmission than to viremia across all survey rounds during which viremia were measured (Fig. 2c and Extended Data Fig. 7a). Second, older women contributed less to transmission than viremia suggests, an observation that was consistent with attenuating sexual activity of women from age 25 onwards (Extended Data Fig. 7a). Third, young women and young men tended to be infected by transmitting partners who were substantially older than the typical sexual partners of the same population age group (Fig. 2b and Extended Data Fig. 7b). These findings illustrate the central utility of pathogen genomics to track and understand patterns of transmission, especially when interpreted in the context of population-based surveillance data, and when implemented at high enough sequence coverage to reconstruct directed transmission networks.

This study has important limitations. First, not all census-eligible individuals participated in the survey, primarily due to absence for work or school[14] (Extended Data Fig. 1). We used data from first-time participants as proxies of non-participants, but we cannot rule out that non-participants include disproportionally larger populations of people with HIV and/or with different risk profiles. In this case, sensitivity analyses (Supplementary Table 10) indicate that more viremic men would have to be reached in all intervention scenarios for similar HIV incidence reductions in women as in Fig. 4. Second, we were only able to deep-sequence a fraction of all transmission events, and these may not be representative of all transmissions. We characterized sampling probabilities under the assumption that individuals were ever deep sequenced at random within age and gender strata, and found that the sampling probabilities did not differ substantially between strata in each round (Extended Data Fig. 5), so that the estimated transmission flows were not sensitive to our sampling probability adjustments (Supplementary Table 10). Of course, these sampling adjustments are modelled and it is possible that missing data could bias our findings. Third, our error analyses indicate that deep-sequence phylogenetics are not a perfect marker of direction of transmission, with estimated false discovery rates of 16.3% (8.8–28.3%) in this cohort[21]. Fourth, over time some communities were added and others left RCCS (Supplementary Table 2). We repeated our analysis on the subset of 28 continuously surveyed communities, and found similar incidence and transmission

dynamics (Supplementary Table 10). Fifth, our findings on rural and semi-urban populations may not extend to populations with different demographics, risk profiles or healthcare access, and this includes populations in urban or metropolitan areas or key populations.

This study demonstrates shifting patterns in HIV incidence and in the drivers of HIV infection in communities typical of rural and semi-urban east Africa, providing key data for evidence-informed policymaking. We find incidence rates have dropped substantially in women aged 15–24 years from 2003 to 2018, and incidence rates now peak among women aged 25–35 years, consistent with cross-sectional national surveillance data from Uganda[60]. Shifts in women's incidence are the result of an increase in the age of transmitting male partners, and the primary contribution to HIV transmission lies now in men aged 30 years and above. The growing contribution of men to heterosexual transmission is associated with substantially slower declines of unsuppressed viremia in men than in women. We predict successful interventions centred on men that bring suppression rates in men on par with those in women could reduce incidence in women by half and close the gender gap in new infections. These findings reinforce calls for HIV prevention programming and services to give greater priority to reach and retain in care men with HIV as this will improve male health, substantially reduce incidence in women, and close gender gaps in infection burden.

## Methods

### The Rakai Community Cohort Study

**Longitudinal surveillance.** Between September 2003 and May 2018, 9 consecutive survey rounds of RCCS, labelled as survey rounds 10 to 18, were conducted in 36 inland communities in south-central Uganda (Fig. 1, Supplementary Tables 1 and 2, and Supplementary Fig. 1). The results presented in this paper derive from data collected through these surveys, including the population census, the RCCS survey participants, the incidence cohort and the phylogenetic transmission cohort.

RCCS survey methods have been reported previously[14,18]. In brief, for each survey round, the RCCS did a household census, and subsequently invited all individuals who were aged 15-49 years and residents for at least 1 month to participate in the open, longitudinal RCCS survey; and so data collection was not randomized. Data collection was blind relative to previous interactions with individuals or any personal characteristics apart from age and residency status, and any research questions. Eligible individuals first attended group consent procedures, and individual consent was obtained privately by a trained RCCS interviewer. Following consent, participants reported in a private location, typically a tent at the survey hub, on demographics, behaviour, health and health service use. All participants were offered free voluntary counselling and HIV testing as part of the survey. Rapid tests at the time of the survey and confirmatory enzyme immunoassays were performed to determine HIV status. All participants were provided with pre-test and post-test counselling, and referrals of individuals who were HIV-positive for ART. Additionally, all consenting participants, irrespective of HIV status, were offered a venous blood sample for storage/future testing, including viral phylogenetic studies. Supplementary Table 1 summarizes the characteristics of the RCCS participants and HIV-positive participants by age and gender. For the purpose of our analyses, we combined data from three pairs of geographically close areas in peri-urban settings into three communities, and 28 of 36 communities were continuously surveyed over all rounds (Supplementary Table 2). All epidemiologic data collected through RCCS are stored in a database running Microsoft SQL server 2019 and Microsoft Access version 2016.

**Population size estimates.** To characterize changes in population demography, individual-level data on the census-eligible individuals that were obtained during each census were aggregated by gender, 1-year age band (between 15 and 49 years) and survey round (Extended Data Fig. 1a,b, bars). The age reported by household heads in the census surveys tended to reflect grouping patterns towards multiples of five, suggesting that household heads reported ages only approximately. For this reason, we smoothed population sizes across ages independently for every gender and survey round, using locally weighted running line smoother (LOESS) regression methods that fit multiple polynomial regressions in local neighbourhoods as implemented in the R package stats (version 3.6.2) with the span argument set to 0.5 (Extended Data Fig. 1a,b, line). Model fit was assessed visually without a formal test, suggesting that the data met the assumptions of the statistical model.

**Participation rates.** To characterize participation rates, we calculated the proportion of RCCS participants in the census-eligible population by gender, 1-year age band and survey round (Extended Data Fig. 1c,d, bars). Following consent, participants reported either their birth date or current age themselves, and accompanying documentary evidence was requested. There were no obvious age grouping patterns of multiples of 5 among participants. Overall, participation rates were lower in men than women (63% versus 75%). Participation rates also increased with age for both men and women, and were very similar across survey rounds. Considering the grouping patterns by age in the population count data, we again smoothed the participation rates across ages independently for every gender and survey round using LOESS regression as specified above for population size estimation (Extended Data Fig. 1c,d, line). Model fit was assessed visually without a formal test, suggesting that the data met the assumptions of the statistical model.

**HIV status and prevalence.** All RCCS participants were offered free HIV testing. Prior to October 2011, HIV testing was performed through enzyme immunoassays (EIAs) with confirmation via western blot and DNA polymerase chain reaction (PCR). After October 2011, testing was performed through a combination of three rapid tests with confirmation of positives, weakly positives and discordant results by at least two EIAs and western blot or DNA PCR[61]. Overall, 99.7% participants took up the test offer across survey rounds, and Supplementary Table 1 documents the number of participants with HIV. From these survey data, we estimated HIV prevalence (that is, probability for a participant to have HIV) with a non-parametric Bayesian model over the age of participants independently for both genders and survey round. Specifically, we used a binomial likelihood on the number of participants with HIV parameterized by the number of participants and HIV prevalence in each 1-year age band. The HIV prevalence parameter was modelled on the logit scale by the sum of a baseline term and a zero-mean Gaussian process on the age space. The prior on the baseline was set to a zero-mean normal distribution with a standard deviation of 10. The covariance matrix of the Gaussian Process was defined with a squared exponential kernel, using a zero-mean half-normal prior with a standard deviation of 2 on the scale parameter of the squared exponential kernel and a zero-mean half-normal prior with a standard deviation of 11.3 ($= (49 − 15)/3$) on the lengthscale of the squared exponential kernel. The model was fitted with Rstan (release 2.21.0) using Stan's adaptive Hamiltonian Monte Carlo (HMC) sampler[62] with 10,000 iterations, including 500 iterations of warm-up. Convergence and mixing were good, with highest R-hat value of 1.0029, and lowest effective sample size of 830. The model represented the data well, with 98.57% of data points inside 95% posterior predictive intervals, indicating that the data met the assumptions of the statistical model. For the mathematical modelling of transmission flows, we assumed that age- and gender-specific HIV prevalence were the same in non-participants in the RCCS communities as in the participants in these communities.

**ART use.** The RCCS measures ART use through participant reports since survey round 11. Self-reported ART use reflected viral suppression with high specificity and a sensitivity around 70% in the study population

(Supplementary Table 9). We took the following pre-processing steps. For survey round 10, we assumed self-reported ART use to have been on the same levels as in survey round 11. Next, the ART use field was adjusted to 'yes' for the participants with HIV who did not report ART use but who had a viral load measurement below 1,000 copies per millilitre of blood plasma. Further, we considered it likely that with increasingly comprehensive care and changing treatment guidelines[14,63], ART use in individuals with HIV who did not participate increased substantively over time, and this prompted us to consider as proxy of ART use in non-participants the observed ART use in first-time participants with HIV. Overall, first-time participants represented 15.3–39.9% of all participants across survey rounds. Extended Data Fig. 8a,b exemplifies the self-reported ART use data in male participants and male first-time participants. The ART use rate estimates for participants and first-time participants were obtained using the same Bayesian non-parametric model as for HIV prevalence fitted independently on the reported ART use data of participants and first-time participants. Convergence and mixing were good, with highest R-hat value of 1.0025 and lowest effective sample size of 978 for the participants, and 1.0027, 521 respectively for the first-time participants. The model represented the data well, with 99.67% of data points inside the corresponding 95% posterior predictive intervals for the participants, and 99.24% for the first-time participants, indicating that the data met the assumptions of the statistical model. The resulting, estimated ART use rates in infected men and women are shown in Extended Data Fig. 8c.

**Viral suppression.** Since survey round 15, HIV-1 viral load was measured on stored serum/plasma specimens from infected participants using the Abbott real-time m2000 assay (Abbott Laboratories), which is able to detect a minimum of 40 copies ml$^{-1}$. Viral suppression was defined as a viral load measurement below 1,000 copies ml$^{-1}$ plasma blood following recommendations of the WHO[34]. To estimate virus suppression levels in the infected non-participants, we considered again as proxy data on infected first-time participants. Overall, viral load measurements were obtained from 19.3% of participants with HIV in survey round 15 and nearly all (>97.71%) participants with HIV since survey round 16[64–66]. From these data we estimated the proportion of individuals in the study population with HIV who had suppressed virus in participants and first-time participants (used as proxy for non-participants), using the same Bayesian non-parametric model as for HIV prevalence and ART use. Convergence and mixing were good with the lowest R-hat value of 1.0016 and lowest effective sample size of 461 for the participants and 1.0052, 844 respectively for the first-time participants. The model represented the data well, with 98.19% of data points inside 95% posterior predictive intervals and 97.99% for the first-time participants, indicating that the data met the assumptions of the statistical model. For the purpose of mathematical modelling of transmission flows, we next considered the earlier survey rounds 10 to 14, for which viral load measurements were not available. On average, 93% of individuals reporting ART use also had suppressed virus (Supplementary Table 9), leading us to estimate the number of individuals with suppressed virus before 2011 from corresponding ART use data. Specifically, we estimated the proportion of the study population with HIV that was virally suppressed by adjusting the estimated ART use data with the sensitivity of being virally suppressed given self-reported ART use and the specificity of being virally suppressed given self-reported no ART use estimated from round 15 when available, and otherwise from round 16 (Supplementary Table 9). Specificity and sensitivity values by 1-year age bands were linearly interpolated between the midpoints of the age brackets in Supplementary Table 9. The resulting, estimated virus suppression levels in men and women with HIV are shown in Extended Data Fig. 8d, illustrating that the gap in virus suppression levels increased over time.

**Sexual behaviour.** RCCS participants reported to interviewers in each round on aspects of sexual behaviour, including the number of sexual partners in the past 12 months within the same community, the number of partners outside the community, and in round 15 the demographic characteristics of up to four partners (Supplementary Table 8). To interpret HIV transmission flows in the context of typical sexual contact networks, we focused on the detailed behaviour data collected in round 15 and estimated sexual contact intensities between men and women by 1-year age band, defined as the expected number of sexual contacts of one individual of gender $g$ and age $a$ with the population of the opposite gender $h$ and age $b$ in the same community. Estimates were obtained with the Bayesian rate consistency model (version 1.0.0), using default prior specifications[67]. We noted along with previous work[68–71] that women tended to report considerably fewer contacts than men (Supplementary Table 8), prompting us to include in the linear predictor of contact rates additional age-specific random effects to capture under-reporting behaviour in women. Further, community-specific baseline parameters were added to allow for variation in the average level of contact rates in each community, but the age-specific structure of contact rates was assumed to be identical across communities. The resulting model was fitted to all data pertaining to within-community sexual contacts in the last year, including reports of within-community contacts for which information on the partners remained unreported. Contacts reported with partners from outside the same community were excluded, because male-female contacts have to add up to female-male contacts only in the same population denominator, and hence under-reporting could only be adjusted for when within-community contacts are considered. The model was fitted with CmdstanR (version 0.5.1)[72] using Stan's adaptive HMC sampler[62] with 4 chains, where each chain runs 2,800 iterations, including 300 warm-up iterations. Convergence and mixing were good, with highest R-hat value of 1.003, and lowest effective sample size of 1,745. The model represented the data well, with >99% of data points inside 95% posterior predictive intervals, indicating that the data met the assumptions of the statistical model. Supplementary Table 8 reports the estimated sexual contact intensities from men and women in survey round 15, and shows that the estimated, under-reporting adjusted sexual contact intensities in women were considerably higher than those directly reported. The table also shows that the estimated number of sexual contacts from men to women equal those from women to men, and the estimated age distribution of sexual contacts is shown in Fig. 2 and Extended Data Fig. 7.

### Longitudinal HIV incidence cohort

**Data and outcomes from the incidence cohort.** RCCS encompasses both a full census of the study communities and a population-based survey in each surveillance round, which enables identification and follow-up of unique individuals over time, and thus provides a comprehensive sampling frame to measure HIV incidence. The RCCS incidence cohort comprises all RCCS study participants who were HIV-negative at their first visit (baseline) and had at least one subsequent follow-up visit (Supplementary Fig. 1). Individuals in the incidence cohort were considered to be at risk of acquiring HIV after their first visit, and stopped accruing risk at the date of HIV acquisition or the date of last visit. Exposure times were estimated from data collected at survey visit times similarly as in ref. 14. Individuals in the incidence cohort who remained negative until the last survey round contributed their time between the first and last survey visit to their exposure period. Individuals in the incidence cohort who were found to have acquired HIV must have done so between the visit date of the last round in which they were negative and the visit date of the current round, and the infection date was imputed at random between the two dates. This included incident cases who had no missed visit between the last negative and current visit (type 1) or one missed visit (type 2) as in ref. 14, but also cases who had more than one missed visit (type 3). Unknown dates were imputed at random 50 times, and individual exposure periods and incident cases were then attributed to each survey round, summed over the cohort, and then

averaged over imputations. Supplementary Table 3 and Extended Data Fig. 2 illustrate the age- and gender-specific exposure times and incidence events in each survey round. In sensitivity analyses, we considered only those individuals in the incidence cohort who resided in one of the 28 inland communities that were continuously surveyed across survey rounds 10 to 18, and found similar incidence dynamics with slightly faster declines in incidence rates in younger men, although this difference was not statistically significant. No statistical methods were used to pre-determine sample sizes but our sample sizes are similar to those reported in previous publications[14].

**Modelling and analysis.** The primary statistical objective was to estimate longitudinal age-specific HIV incidence rates by 1-year age bands across (discrete) survey rounds, separately for each gender. We used a log-link mixed-effects Poisson regression model, with individual-level exposure times specified as offset on the log scale, common baseline fixed effect and further random effects. The random effects comprised a one-dimensional smooth function on the age space, a one-dimensional smooth function on the survey round space, and an interaction term between age and survey round. The functions were specified as one-dimensional Gaussian processes. Alternative specifications, including two-dimensional functions over the participant's age and survey round, and without interaction terms between age and survey rounds were also tried. We did not consider incidence trends in continuous calendar time because study communities were surveyed in turn, and so the incidence data within each round are structured by communities, which would require further modelling assumptions to account for. Owing to the large number of individual observations, models were fitted using maximum-likelihood estimation (MLE) with the R package mgcv (version 1.8-38)[73], to each of the 50 datasets with imputed exposure times for each gender independently. Numerical convergence was examined with the gam.check function. Within- and between-sample uncertainties in parameter estimates, from the variability of the estimation procedure and the data imputation procedure, were incorporated in the age-, gender- and survey-round-specific incidence rate estimates by drawing 1,000 replicate incidence rate estimates from the MLE model mean parameter and associated standard deviation obtained on each of the 50 imputation datasets, and then calculating median estimates and 95% prediction intervals over the 1,000 × 50 Monte Carlo estimates (Fig. 1c). Model fits were evaluated by comparing predicted HIV incidence infections estimates to the empirical data. To assess model fit, incident cases were predicted using the Poisson model parameterized by replicate MLE incidence estimates. Overall, model fit was very good, with 98.80% (98.10–99.49) data points inside the 95% prediction intervals across the 50 imputed datasets and the fitted model was consistent with the available data (Extended Data Fig. 6), indicating that the data met the assumptions of the statistical model. The Akaike information criterion was used to identify the best model for each gender, and the best model was as described above (Supplementary Table 4).

**Longitudinal viral phylogenetic transmission cohort**
**Data from the transmission cohort.** Within RCCS, we also performed population-based HIV deep sequencing spanning a period of more than 6 years, from January 2010 to April 2018. The primary purpose of viral deep sequencing was to reconstruct transmission networks and identify the population-level sources of infections, thus complementing the data collected through the incidence cohort.

The RCCS viral phylogenetic transmission cohort comprises all participants with HIV for whom at least one HIV deep-sequence sample satisfying minimum quality criteria for deep-sequence phylogenetic analysis is available (Supplementary Fig. 1). For survey rounds 14 to 16 (PANGEA-HIV 1), viral sequencing was performed on plasma samples from participants with HIV who had no viral load measurement and self-reported being ART-naive at the time of the survey, or who had a viral load measurement above 1,000 copies per ml of plasma. We used

this criterion because viral deep sequencing was not possible within our protocol on samples with virus less than 1,000 copies per ml of plasma, and because self-reported ART use was in this population found to be a proxy of virus suppression with reasonable specificity and sensitivity[14,21]. Plasma samples were shipped to University College London Hospital for automated RNA sample extraction on QIAsymphony SP workstations with the QIAsymphony DSP Virus/Pathogen Kit (catalogue number 937036, 937055; Qiagen), followed by one-step reverse transcription PCR (RT–PCR)[74]. Amplification was assessed through gel electrophoresis on a fraction of samples, and samples were shipped to the Wellcome Trust Sanger Institute for HIV deep sequencing on Illumina MiSeq and HiSeq platforms in the DNA pipelines core facility. Primers are publicly available[74]. For survey rounds 17 and 18 (PANGEA-HIV 2), viral load measurements were available for all infected participants and viral sequencing was performed on plasma samples of individuals who had not yet been sequenced and who had a viral load measurement above 1,000 copies per ml of plasma. Plasma samples were shipped to the Oxford Genomics Centre for automated RNA sample extraction on QIAsymphony SP workstations with the QIAsymphony DSP Virus/Pathogen Kit (937036, 937055; Qiagen), followed by library preparation with the SMARTer Stranded Total RNA-Seq kit v2 - Pico Input Mammalian (Clontech, TaKaRa Bio), size selection on the captured pool to eliminate fragments shorter than 400 nucleotides (nt) with streptavidin-conjugated beads[75] to enrich the library with fragments desirable for deep-sequence phylogenetic analysis, PCR amplification of the captured fragments, and purification with Agencourt AMPure XP (Beckman Coulter), as described in the veSEQ-HIV protocol[76]. Sequencing was performed on the Illumina NovaSeq 6000 platform at the Oxford Genomics Centre, generating 350 to 600 base pair (bp) paired-end reads. Sequencing probes are publicly available[77]. A subset of samples from survey rounds 14 to 16 with low quality read output under the PANGEA-HIV 1 procedure was re-sequenced with the veSEQ-HIV protocol. To enhance the genetic background used in our analyses, additional samples from the spatially neighbouring MRC/UVRI/LSHTM surveillance cohorts and other RCCS communities were also included. For sequencing, the following software were used, QuantStudio Real-Time PCR System v1.3, Agilent TapeStation Software Analysis 4.1.1, Clarity Version 4.2.23.287, FreezerPro 7.4.0-r14598, and LabArchives Electronic Lab Notebook 2023. We restricted our analysis to samples from 2,172 individuals that satisfied minimum criteria on read length and depth for phylogeny reconstruction and subsequent inferences. Specifically, deep sequencing reads were assembled with the shiver sequence assembly software, version 1.5.7[78]. Next, phyloscanner version 1.8.1[25] was used to merge paired-end reads, and only merged reads of at least 250 bp in length were retained in order to generate 250 bp deep-sequence alignments as established in earlier work[21].

Deep sequencing was performed from 2010 (survey round 14) onwards, but because sequences provide information on past and present transmission events, we also obtained information on transmission in earlier rounds and calculated sequence coverage in participants that were ever deep-sequenced at minimum quality criteria for phylogenetic analysis. Specifically, we required that individuals had a depth of ≥30 reads over at least 3 non-overlapping 250 bp genomic windows. Individuals who did not have sequencing output meeting these criteria were excluded from further analysis, and these were largely individuals sequenced only in PANGEA-HIV 1, and were primarily associated with low viral load samples[76,79]. In total, we deep-sequenced virus from 1,978 participants with HIV of who 559 were also in the incidence cohort. Supplementary Table 5 characterizes HIV deep-sequencing outcomes in more detail. No statistical methods were used to pre-determine sample sizes but our sample sizes are similar to those reported in previous publications[20,27,50].

**Reconstruction of transmission networks and source–recipient pairs.** The HIV deep-sequencing pipeline provided sequence fragments that capture viral diversity within individuals, which

enables phylogenetic inference into the direction of transmission from sequence data alone[21,78,80]. First, potential transmission networks were identified, and in the second step transmission networks were confirmed and the transmission directions in the networks were characterized as possible. In this study, the first step was modified from previous protocols[21] to ease computational burden, while the second step was as before performed with phyloscanner version 1.8.1.

In the first step[81], to identify potential transmission networks, HIV consensus sequences were generated as the most common nucleotide in the aligned deep-sequence fragments that were derived for each sample. We then calculated similarity scores between all possible combinations of consensus sequences in consecutive 500 bp genomic windows rather than the entire genome to account for the possibility of recombination events and divergent virus in parts of the genome. Similarity score thresholds to identify putative, genetically close pairs were derived from data of long-term sexual partners enrolled in the RCCS cohort similarly as in refs. 21,81, and then applied to the population-based sample of all possible combinations of successfully sequenced individuals. Overall, 2,525 putative, genetically close individuals were identified, and these formed 305 potential transmission networks.

In the second step, we confirmed the potential transmission networks in phylogenetic deep-sequence analyses. We updated the background sequence alignment used in phyloscanner to a new sequence dataset that included 113 representatives of all HIV subtypes and circulating recombinant forms and 200 near full-genome sequences from Kenya, Uganda and Tanzania, obtained from the Los Alamos National Laboratory HIV Sequence Database (http://www.hiv.lanl.gov/). The deep-sequence alignment options were updated to using MAFFT (version 7.475) with iterative refinement[82], and additional iterative re-alignment using consistency scores in case a large proportion of gap-like columns in the first alignment was detected. Deep-sequence phylogeny reconstruction was updated to using IQ-TREE (version 2.0.3) with GTR+F+R6 substitution model, resolving the previously documented deep-sequence phylogenetics branch length artefact[20,83]. Confirmatory analyses of the potential transmission networks were updated to using phyloscanner (version 1.8.1) with input argument zeroLengthAdjustment set to TRUE. From phyloscanner output, we calculated pairwise linkage scores that summarize how frequently viral phylogenetic subgraphs of two individuals were adjacent and phylogenetically close in the deep-sequence phylogenies corresponding to all 250 bp genomic windows that contained viral variants from both individuals[21,25]. Similarly we calculated pairwise direction scores that summarize how frequently viral phylogenetic subgraphs of one individual were ancestral to the subgraphs of the other individual in the deep-sequence phylogenies corresponding to all 250 bp genomic windows that contained viral variants from both individuals and in which subgraphs had either ancestral or descendant relationships[21,25]. Phylogenetically likely source–recipient pairs with linkage scores ≥0.5 and direction scores ≥0.5 were extracted, and only the most likely source–recipient pair with highest linkage score was retained if multiple likely sources were identified for a particular recipient. The resulting source–recipient pairs were checked further against sero-history data from both individuals where available. If sero-history data indicated the opposite direction of transmission, the estimated likely direction of transmission was set to that indicated by sero-history data.

**Infection time estimates.** The shape and depth of an individual's subgraph in deep-sequence phylogenies also provide information on the time since infection, and since the sequence sampling date is known thus also on the infection time[84] and the age of both individuals at the time of the infection event. We used the phyloTSI random forest estimation routine with default options, which was trained on HIV seroconverter data from the RCCS and other cohorts, and uses as input the output of the phyloscanner software[26]. Individual-level time since infection estimates were associated with wide uncertainty (Extended Data Fig. 4a), and for this reason we refined estimates for the phylogenetically likely recipient in source–recipient pairs using the inferred transmission direction, age data, and where available longitudinal sero-history data. Specifically, we refined plausible infection ranges as indicated in the schema in Supplementary Fig. 2. Here, the dotted red rectangle illustrates the 2.5% and 97.5% quantiles of the phyloTSI infection time estimates for the phylogenetically likely recipient (x axis) and transmitting partner (y axis). We incorporated evidence on the direction of transmission by requiring that the date of infection of the phylogenetically likely recipient is after that of the transmitting partner (filled red triangle). Sero-history and demographic data were incorporated as follows. For both the recipient and the transmitting partner, the upper bound of the infection date was set as the thirtieth day prior to the first positive test of the participant[85]. The lower bound of the infection date was set to the largest of the following dates, the date of last negative test if available, the fifteenth birthday, or the date corresponding to 15 years prior the upper bound[86]. The refined uncertainty range of the infection time estimates of the phylogenetically likely transmitting partner and recipient are illustrated as the purple triangle in the schema above, and obtained as follows. Firstly, we defined individual-level plausible ranges, by intersecting the range of dates consistent with the phyloTSI predictions and sero-history data. If the intersection was empty, we discarded the phyloTSI estimates. Then we intersected the rectangle given by the cartesian product of the plausible intervals for source and recipient with the half-plane consistent with the direction of transmission. Finally, infection dates were sampled at random from the refined uncertainty range, so that the median infection date estimates correspond to the centre of gravity of the triangle (cross). In sensitivity analyses, we further integrated estimates of transmission risk by stage of infection[87], though this had limited impact on the estimates (see 'Sensitivity analyses' section). In cases where the likely transmitting partner in one heterosexual pair was the recipient partner in another heterosexual pair, the above infection date refinement algorithm was applied recursively so that the refined infection date estimates were consistent across pairs. Finally, the transmission events captured by each source–recipient pair were attributed to the survey round into which the posterior median infection time estimate of the recipient fell, and in cases where the median estimate fell after the start time of a round and the end time of the preceding round, the event was attributed to the preceding round.

In total, we identified 539 source–recipient pairs that involved participants from the 36 survey communities and further individuals from the background dataset. In 13 of the 539 source–recipient pairs, available dates of last negative tests indicated that only the opposite transmission direction was possible and in these cases the inferred direction of transmission was set to the opposite direction. The resulting pairs included 501 unique recipient partners, and for reach we retained the most likely transmitting partner. To identify pairs capturing transmission events within the RCCS inland communities, we restricted analysis initially to 236 heterosexual source–recipient pairs in whom both individuals were ever resident in the 36 survey communities. Of these, 142 pairs were from men to women and 94 from women to men. Infection times were estimated for all sampled individuals and refined for the recipient partners in the 236 heterosexual source–recipient pairs. For 4 recipient partners, the phyloTSI estimates were ignored as they were incompatible with inferred transmission direction and survey data, and was based on sero-history data only. The phylogenetically most likely location of both individuals at time of transmission was estimated as their location at the RCCS visit date that was closest to the posterior median infection time estimate. Using this location estimate, 233 of the 236 heterosexual source–recipient pairs were estimated to capture transmission events in RCCS inland communities and were retained

for further analysis. A further six recipient partners had posterior median infection time estimates outside the observation period from September 2003 to May 2018 and were excluded, leaving for analysis 227 heterosexual source–recipient pairs that captured transmission events in RCCS inland communities during the observation period. This excluded 88 potential source–recipient pairs from our study due to ethical considerations and prior analyses suggesting these pairs most likely represent partially sampled transmission chains (that is, 'false positives')[21].

### Transmission flow analysis

**Statistical framework.** We next estimated the sources of the inferred population-level HIV incidence dynamics from the dated, source–recipient pairs in the viral phylogenetic transmission cohort. Overall, inference was done in a Bayesian framework using a semi-parametric Poisson flow model similar ref. 28, that was fitted to observed counts of transmission flows $Y_{p,i,j}^{g \to h}$ with transmission direction $g \to h$ (male-to-female or female-to-male), time period $p$ (survey rounds 10–15 and 16–18) in which the recipient was likely infected, and 1-year age bands $i, j$ of the source and recipient populations respectively, where

$$i, j \in \mathcal{A} = \{15, 16, \ldots, 48, 49\} \tag{1a}$$

$$(g \to h) \in \mathcal{D} = \{\text{male-to-female, female-to-male}\}. \tag{1b}$$

The target quantity of the model is the expected number of HIV transmissions in the study population in transmission direction $g \to h$ (male-to-female or female-to-male), survey round $r$ (survey round 10 to 18) in which infection occurred, and 1-year age bands $i, j$ of the source and recipient populations respectively, which we denote by $\lambda_{r,i,j}^{g \to h}$. We considered that the expected number of HIV transmissions in the study population is characterized by transmission risk and modulated by the number of infectious and susceptible individuals, which prompted us to express $\lambda_{r,i,j}^{g \to h}$ in the form of a standard discrete-time susceptible-infected (SI) model,

$$\lambda_{r,i,j}^{g \to h} = \beta_{r,i,j}^{g \to h} \times S_{r,j}^{h} \times I_{r,i}^{g} \times |(t_r^{\text{end}} - t_r^{\text{start}})|, \tag{2}$$

where $\beta_{r,i,j}^{g \to h} > 0$ is the transmission rate exerted by one infected, virally unsuppressed individual of gender $g$ and age $i$ on one person in the uninfected ('susceptible') population of the opposite gender $h$ and age $j$ in a standardized unit of time in survey round $r$. Additionally, $S_{r,j}^{h}$ is the number of susceptible individuals of gender $h$ and age $j$ in survey round $r$ and $I_{r,i}^{g}$ is the number of infected, virally unsuppressed individuals of gender $g$ and age $i$ in survey round $r$. With equation (2), we express expected transmission flows with a population-level mechanism of how transmission rates from individuals with unsuppressed HIV act on the susceptible population, and we preferred equation (2) over a purely phenomenological model of the $\lambda_{r,i,j}^{g \to h}$ for the generalizing insights it provides. The main simplifying approximations in equation (2) are that all quantities on the right-hand side of equation (2) are in discrete time and constant in each round, meaning we approximate over changes in population size, HIV prevalence and viral suppression at a temporally finer scale, and assume further that one generation of transmissions occurs from individuals with unsuppressed HIV in each round. Importantly, in this framework, we can then relate the expected transmission flows to the HIV incidence dynamics and the data from the longitudinal incidence cohort by summing in equation (2) over the sources of infections,

$$\sum_i \lambda_{r,i,j}^{g \to h} = \left( \sum_i \beta_{r,i,j}^{g \to h} \times I_{r,i}^{g} \right) \times S_{r,j}^{h} \times |(t_r^{\text{end}} - t_r^{\text{start}})| \tag{3a}$$

$$=: \kappa_{r,j}^{h} \times S_{r,j}^{h} \times |(t_r^{\text{end}} - t_r^{\text{start}})|, \tag{3b}$$

where $\kappa_{r,j}^{h}$ is the incidence rate per susceptible person of gender $h$ and age $j$ in survey round $r$ (that is, $S_{r,j}^{h}$) and per unit time ($|(t_r^{\text{end}} - t_r^{\text{start}})|$). Estimates of $\kappa_{r,j}^{h}$ were calculated in units of 100 PY as described above and shown in Fig. 1c, and we will constrain the semi-parametric Poisson flow model using these estimates. From the model output, we are primarily interested in the transmission flows and transmission sources during each round as quantities out of 100%, defined respectively by

$$\pi_{r,i,j}^{g \to h} = \lambda_{r,i,j}^{g \to h} \Big/ \left( \sum_{i' \in \mathcal{A}, j' \in \mathcal{A}, k \in \mathcal{D}} \lambda_{r,i',j'}^{k} \right) \tag{4a}$$

$$\delta_{r,i,j}^{g \to h} = \pi_{r,i,j}^{g \to h} \Big/ \left( \sum_{i' \in \mathcal{A}} \pi_{r,i',j}^{g \to h} \right) \tag{4b}$$

$$\delta_{r,i}^{g \to h} = \sum_{j \in \mathcal{A}} \pi_{r,i,j}^{g \to h}. \tag{4c}$$

In words, equation (4b) quantifies the sources of infection in individuals of gender $h$ and age $j$ in survey round $r$ such that the sum of $\delta_{r,i,j}^{g \to h}$ across $i$ equals one, and equation (4c) quantifies the sources of infection in the entire population in survey round $r$ that originate from the group of individuals of gender $g$ and age $i$ such that the sum of $\delta_{r,i}^{g \to h}$ across $g$ and $i$ equals one. The width of the boxplots in Fig. 2b shows equation (4b) and Fig. 2a,c show equation (4c).

**Specification of susceptible and infected individuals.** The number $S_{r,j}^{h}$ of the susceptible population of gender $h$ and age $j$ was calculated by multiplying the smoothed estimate $N_{r,j}^{h}$ of the census-eligible population of gender $h$ and age $j$ (shown in Extended Data Fig. 1a,b) with 1 minus the posterior median estimate of HIV prevalence $\rho_{r,j}^{h}$ in census-eligible individuals of gender $h$ and age $j$ in survey round $r$ (calculated as described further above). To specify the number $I_{r,i}^{g}$ of individuals with unsuppressed HIV of gender $g$ and age $i$, we multiplied the smoothed estimate $N_{r,i}^{g}$ of the census-eligible population of gender $g$ and age $i$ in survey round $r$ (shown in Extended Data Fig. 1a,b) with the posterior median estimate of HIV prevalence in the census-eligible population of gender $g$ and age $i$ ($\rho_{r,i}^{g}$) with 1 minus the posterior median estimate $v_{r,i}^{g}$ of the proportion of census-eligible individuals of gender $g$ and age $i$ in survey round $r$ that have suppressed HIV (calculated as described further above and shown in Extended Data Fig. 8d). The start and end times of each survey round, $t_r^{\text{start}}$ and $t_r^{\text{end}}$ were set as shown in Fig. 1b and specified in units of years, so that the transmission intensity is also expressed in units of years.

**Bayesian model.** We first present the likelihood of the observed counts of transmission flows $Y_{p,i,j}^{g \to h}$ under the semi-parametric Poisson flow model that is parameterized in terms of equation (2). The phylogenetically reconstructed source–recipient pairs capture only a subset of incidence events, and so it is important to characterize the sampling frame. As in ref. 28, we consider the unknown transmission events $Z_{r,i,j}^{g \to h}$ in survey round $r$ and assume these are sampled at random within each strata with probabilities that factorize into sampling probabilities of sources of age $i$ and gender $g$ and sampling probabilities of recipients of age $j$ and gender $h$, $Y_{r,i,j}^{g \to h} \sim \text{Binomial}(Z_{r,i,j}^{g \to h}, \xi_{r,g,i}^{1} \xi_{r,h,j}^{2})$. Using equation (4a), we also let $Z_{r,i,j}^{g \to h} \sim \text{Multinomial}(Z_r, \pi_{r,i,j}^{g \to h})$, where $Z_r$ is the total number of infection events in round $r$.

Because we have data from both the transmission and incidence cohorts, we are able to constrain the sampling problem with the detection probabilities of incidence events. Specifically, setting $Y_{r,j}^{h} = \sum_{i \in \mathcal{A}} Y_{r,i,j}^{g \to h}$ and $Z_{r,j}^{h} = \sum_{i \in \mathcal{A}} Z_{r,i,j}^{g \to h}$, we let $Y_{r,j}^{h} \sim \text{Binomial}(Z_{r,j}^{h}, \zeta_{r,j}^{h})$ and set the detection probability to the proportion of the expected number of incident cases of gender $h$ and age $j$ that could be phylogenetically reconstructed in time period $p$,

$$\zeta_{p,j}^h = \left( \sum_{r \in p, i \in \mathcal{A}} Y_{r,i,j}^{g \to h} \right) \Bigg/ \left( \sum_{r \in p} \kappa_{r,j}^h \times S_{r,j}^h \times |(t_r^{\text{end}} - t_r^{\text{start}})| \right), \tag{5}$$

for all rounds $r$ in the two time periods rounds 10–15 and 16–18. We set the detection probability of round $r$ to match that of its corresponding period $p$. We focused in equation (5) on time periods due to the limited phylogenetic count data. The advantage in constraining the transmission model with the detection probabilities (equation (5)) is that the estimates of the transmission model will be consistent with the incidence dynamics that we already estimated with data from the incidence cohort. Re-arranging terms between binomial and multinomial models, we obtain

$$Y_{r,i,j}^{g \to h} \sim \text{Multinomial}\left(Z_r, \xi_{r,g,i}^1 \xi_{r,h,j}^2 \pi_{r,i,j}^{g \to h}\right) \tag{6a}$$

$$\xi_{r,h,j}^2 = \frac{\sum_{i \in \mathcal{A}} \pi_{r,i,j}^{g \to h}}{\sum_{i \in \mathcal{A}} \xi_{r,g,i}^1 \pi_{r,i,j}^{g \to h}} \zeta_{r,j}^h, \tag{6b}$$

which shows that the sampling probabilities of recipients $\xi_{r,h,j}^2$ can be expressed in terms of the detection probability of infection events, weighted by the relative contribution and sampling of source-specific transmission events to the same incidence group. We still need to specify $\xi_{r,h,i}^1$ to complete the sampling model. Here, we approximated the sampling probability of sources with the proportion of individuals of age $i$ and gender $h$ with unsuppressed virus in round $r$ that were ever deep-sequenced. Note that the sampling model (equation (6)) will alter the posterior mean transmission flows $\pi_{r,i,j}^{g \to h}$ only when the sampling probabilities $\xi_{r,h,i}^1$ and $\zeta_{r,j}^h$ differ between age and gender strata in the same round. Extended Data Fig. 5 visualizes our specifications of $\zeta_{r,j}^h$ and $\xi_{r,h,j}^1$, and shows that the sampling differences between age and gender groups are relatively modest in any given round, which suggests that the adjustments on the inferred transmission flows based on our modelled sampling probabilities will be modest.

In the semi-parametric Poisson flow model of ref. 28, the sampling model (equation (6)) can be analytically integrated out based on standard thinning properties, which in turn allows us to express the likelihood of observing the phylogenetic data with

$$Y_{p,i,j}^{g \to h} \sim \text{Poisson}\left(\sum_{r \in p} \xi_{r,g,i}^1 \xi_{r,h,j}^2 \lambda_{r,i,j}^{g \to h}\right) \tag{7a}$$

$$\lambda_{r,i,j}^{g \to h} = \beta_{r,i,j}^{g \to h} \times S_{r,j}^h \times F_{r,i}^g \times |(t_r^{\text{end}} - t_r^{\text{start}})| \tag{7b}$$

$$\log \beta_{r,i,j}^{g \to h} = \hat{c}^{g \to h}(i,j) + \gamma_0 + \gamma_g + \gamma_r + \gamma_{p(r)} + f_0^{g \to h}(i,j)$$
$$+ f_r^{g \to h}(j) + f_{p(r)}^{g \to h}(i) \tag{7c}$$

$$\xi_{r,h,j}^2 = \frac{\sum_{i \in \mathcal{A}} \lambda_{r,i,j}^{g \to h}}{\sum_{i \in \mathcal{A}} \xi_{r,g,i}^1 \lambda_{r,i,j}^{g \to h}} \zeta_{r,j}^h, \tag{7d}$$

where $\hat{c}^{g \to h}(i,j)$ is the posterior median estimate of the log rate of sexual contacts within communities in one year between one person of age $i$ and gender $g$ and one person of age $j$ and gender $h$ that we estimated from the sexual behaviour data, and the remaining terms quantify the transmission probability per sexual contact on the log scale. The model is designed in such a way that the log sexual contact rates describe a fixed age-specific non-zero mean surface, and the remaining parameters describe age-specific random deviations around the mean surface. With this approach, any inferred deviations in

transmission rates relative to sexual contact rates are informed by the phylogenetic data and robust to prior specifications on the random deviations. Specifically, $\gamma_0$ is the baseline parameter characterizing overall transmission risk per sexual contact, $\gamma_g$ is a gender-specific offset which is set to zero in the female-to-male direction and a real value in male-to-female direction, $\gamma_r$ a round-specific offset which is set to zero for the first survey round 10, and $\gamma_p$ is a time period specific offset which is set to zero for the first time period. We assume the age-specific structure of transmission rates in terms of the transmitting partners (denoted by $i$) and recipients (denoted by $j$) are similar across similar ages, and so we can exploit regularizing prior densities[28] to learn smooth, latent transmission rate surfaces from the sparse data shown in Extended Data Fig. 3. In detail, we modelled the age-specific structure of transmission rates non-parametrically with 2 time-invariant random functions $f_0^{g \to h}$ with two-dimensional inputs on the domain $[15, 50] \times [15, 50]$ that characterize age–age interactions in transmission risk for each gender, $2 \times 8$ random functions $f_r^{g \to h}$ with one-dimensional inputs that characterize time trends in the age of recipients for each gender for survey rounds after round 10, and 2 random functions $f_p^{g \to h}$ with one-dimensional inputs that characterize time trends in the age of transmitting partners for each gender for the second time period. We attach to each of these random functions computationally efficient B-splines projected Gaussian process (GP) priors[88], which we constructed by describing the random functions with cubic B-splines over equidistant knots and modelling the prior relationship of the B-splines parameters with GPs with squared exponential kernels with variance and lengthscale hyper-parameters, denoted respectively by $\sigma^2$ and $\ell$. The prior densities of our Bayesian model are

$$\gamma_0 \sim \mathcal{N}(0, 10^2) \tag{8a}$$

$$\gamma_{\text{male}} \sim \mathcal{N}(0, 1) \tag{8b}$$

$$\gamma_r \sim \mathcal{N}(0, 1) \qquad \text{for } r > \text{R10} \tag{8c}$$

$$\gamma_p \sim \mathcal{N}(0, 1) \qquad \text{for } p = \text{R16-R18} \tag{8d}$$

$$f_0^{g \to h} \sim \text{2D-B-splines-GP}\left(\sigma_0^{g \to h}, \ell_{0,1}^{g \to h}, \ell_{0,2}^{g \to h}\right) \tag{8e}$$

$$f_r^{g \to h} \sim \text{1D-B-splines-GP}\left(\bar{\sigma}_r^{g \to h}, \bar{\ell}_r^{g \to h}\right) \qquad \text{for } r > \text{R10} \tag{8f}$$

$$f_p^{g \to h} \sim \text{1D-B-splines-GP}\left(\breve{\sigma}^{g \to h}, \breve{\ell}^{g \to h}\right) \qquad \text{for } p = \text{R16-R18} \tag{8g}$$

$$\sigma_{0,i}^{g \to h}, \sigma_{0,j}^{g \to h}, \bar{\sigma}^{g \to h}, \breve{\sigma}^{g \to h} \sim \text{Half-Cauchy}\,(0, 1) \tag{8h}$$

$$\ell_{0,1}^{g \to h}, \ell_{0,2}^{g \to h}, \tilde{\ell}^{g \to h}, \breve{\ell}^{g \to h} \sim \text{Inv-Gamma}\,(2, 2), \tag{8i}$$

where the $2 \times 8$ recipient-specific time-varying 1D B-splines GPs each have squared exponential kernels with gender- and round-specific hyper-parameters $\bar{\sigma}_r^{g \to h}$, $\tilde{\ell}_r^{g \to h}$, the 2 source-specific time-varying 1D B-splines GPs each have squared exponential kernels with gender-specific hyper-parameters $\breve{\sigma}^{g \to h}$, $\breve{\ell}^{g \to h}$, and the 2 time-invariant 2D B-splines GPs each have squared exponential kernels with gender-specific hyper-parameters $\sigma_0^{g \to h}$, $\ell_{0,1}^{g \to h}$ and $\ell_{0,2}^{g \to h}$ decomposed as follows,

$$k_0^{g \to h}((i,j),(i',j')) = (\sigma_0^{g \to h})^2 \exp\left(-\frac{(i-i')^2}{2(\ell_{0,1}^{g \to h})^2}\right) \exp\left(-\frac{(j-j')^2}{2(\ell_{0,2}^{g \to h})^2}\right). \tag{9}$$

We constrain the model further with a pseudo-likelihood term so that the model's implied incidence rate $\kappa_{r,j}^h$ in equation (3b) is around the

MLE incidence rate estimate obtained from the incidence cohort. We took this approach in lieu of fitting the model to both the source–recipient and individual-level incidence exposure data to bypass extreme computational runtimes[12], and in the context that the source–recipient data are not informative of incidence dynamics[89]. Specifically, we fitted log-normal distributions to the 1,000 × 50 Monte Carlo replicate rate estimates for individuals of gender $h$ and age $j$ in round $r$ (see above) using the lognorm R package (version 0.1.6)[90], and then set

$$\frac{\sum_i \lambda_{r,i,j}^{g \to h}}{S_{r,j}^h \times |(t_r^{\text{end}} - t_r^{\text{start}})|} \sim \text{LogNormal}\left(\text{mean} - \hat{\kappa}_{r,j}^h, \text{var} - \hat{\kappa}_{r,j}^h\right), \quad (10)$$

where mean-$\hat{\kappa}_{r,j}^h$ and var-$\hat{\kappa}_{r,j}^h$ denote respectively the parameters of the fitted log-normal distributions, and the left-hand side is calculated from equation (7b) and matches the model's incidence rate $\kappa_{r,j}^h$ in equation (3b).

**Computational inference.** Model (7–10) was fitted with Rstan (version 2.21.0), using Stan's adaptive HMC sampler[62] with 4 chains for 3,500 iterations including 500 warm-up iterations. Convergence and mixing were good, with highest R-hat value of 1.0027 and lowest effective sample sizes of 1,444. The model presented the data well, with 99.63% data point inside 95% posterior predictive intervals and the fitted model was consistent with all the available data (Extended Data Fig. 6), indicating that the data met the assumptions of the statistical model. There were no divergent transitions, suggesting non-pathological posterior topologies.

**Counterfactual interventions**
We investigated – given the inferred transmission flows – the hypothetical impact of targeted counterfactual intervention scenarios $c$ on predicted incidence reductions in women in the most recent survey round 18. In the model, counterfactual interventions were implemented by calculating the expected number of transmission flows (equation (2)) into women under counterfactual $c$ that fewer men of age $i$ had remained with unsuppressed HIV in survey round 18, which we denote by $\tilde{I}_{\text{R18},i,c}^M$. We obtained the expected number of incident cases in women of age $j$ in round 18 in counterfactual $c$ via

$$\tilde{\lambda}_{\text{R18},j,c}^{M \to F} = \sum_i \int \hat{\beta}_{\text{R18},i,j}^{M \to F} \times \tilde{I}_{\text{R18},i,c}^M \times S_{\text{R18},j}^F \times |(t_{\text{R18}}^{\text{end}} - t_{\text{R18}}^{\text{start}})| \, d\hat{\beta}_{\text{R18},i,j}^{M \to F}, \quad (11)$$

where uncertainty in the posterior age-specific transmission rates after fitting model ((7)–(10)) is integrated out. The predicted incidence rate reductions were based on comparing the counterfactuals (11) to the inferred cases in women in the corresponding age group (3b), $1 - \left(\sum_j \tilde{\lambda}_{\text{R18},j,c}^{M \to F}\right) / \left(\sum_j \hat{\lambda}_{\text{R18},j}^{M \to F}\right)$.

**Closing half the gap in viral suppression rates in men relative to women.** In this scenario, we considered the impact of reducing by half the gap in the proportion of men with unsuppressed HIV compared to the same proportion in women. To this end, we first calculated for each 1-year age band the average of the estimated proportion of census-eligible infected men in round 18 with suppressed virus and the same proportion in women, $\bar{v}_{\text{R18},i}^M = (v_{\text{R18},i}^M + v_{\text{R18},i}^F)/2$. Next, we set $\tilde{I}_{\text{R18},i,1}^M$ to the smoothed estimate of census-eligible men of age $i$ in round 18 multiplied with the posterior median estimate of HIV prevalence in census-eligible men of age $i$, and with $1 - \bar{v}_{\text{R18},i}^M$.

**Closing the gap in viral suppression rates in men relative to women.** In this scenario, we considered the impact of achieving the same proportions of men with unsuppressed HIV as in women. To this end, we set $\tilde{I}_{\text{R18},i,2}^M$ to the smoothed estimate of census-eligible men of age $i$ in

round 18 multiplied with the posterior median estimate of HIV prevalence in census-eligible men of age $i$, and with $1 - v_{\text{R18},i}^F$.

**95-95-95 in men.** In this scenario, we considered the impact of achieving viral suppression in 85.7% (0.95 × 0.95 × 0.95) in each 1-year age group of men with HIV. The number of remaining men with unsuppressed HIV in round 18, $\tilde{I}_{\text{R18},i,3}^M$, was calculated by multiplying the smoothed estimate of the census-eligible men of age $i$ in round R18 with the posterior median estimate of HIV prevalence in the census-eligible men of age $i$, and with $1 - 0.95^3$.

**Sensitivity analyses**
**Sensitivity in incidence rate estimates to the GAM incidence model specification.** The longitudinal age-specific HIV incidence rates of the central analysis were estimated with a log-link generalized additive effects Poisson regression model with a linear predictor comprising relatively simple main and interaction effects by age and survey round, fitted to individual-level 0/1 incidence outcomes and exposure times specified as offset on the log scale. To assess sensitivity against the relatively simple linear predictor, we considered a more complex mean specification comprising independent LOESS smoothers to capture age-specific incidence trends in each survey round, and fitted this mean model for computational reasons to crude HIV incidence rates. Specifically, we fitted LOESS regressions as implemented in the R package stats (version 3.6.2) with span argument set to 0.7 across the age space independently to each of the crude gender- and round-specific HIV incidence rates in all 50 imputation datasets, and weighted by the corresponding, group-level aggregated exposure times. The HIV incidence rate estimates under the LOESS model had as expected a smaller mean absolute error against the crude estimates as compared against the GAM model (0.0048 (0.0046–0.0051) versus 0.0053 (0.0051–0.0056); Supplementary Fig. 3). Overall, the contribution of men to incidence was more variable across rounds while the shifts in the median age at infection were similar in the central and this sensitivity analysis (Supplementary Table 10).

**Sensitivity in incidence rate and transmission flow estimates to limited communities.** Over time some communities were added and others left RCCS (Supplementary Table 2). We repeated our analysis on the subset of 28 consecutively surveyed communities. We found similar incidence rates with slightly faster declines in male new infections and larger gender disparities (Supplementary Fig. 4). All other primary findings remained insensitive (Supplementary Table 10).

**Sensitivity in estimating transmission flows to uncertainty in infection time estimates.** In the central analysis, phyloTSI infection time estimates associated to source–recipient pairs were refined using the inferred transmission direction, age, and sero-history data. To assess sensitivity to the infection time estimates used, we inferred transmission flows on the basis of the raw phyloTSI infection time estimates as long as they were compatible with the inferred transmission direction, and otherwise on the basis of the refined estimates. Overall, we found source–recipient pairs were potentially allocated to earlier or later time periods reflecting the wide uncertainty in infection time estimates, though across the sample the age distribution of sources and recipients was remarkably stable (Extended Data Fig. 4). All primary findings were insensitive to using the raw infection time estimates (Supplementary Table 10).

**Sensitivity in time since infection estimates to higher transmissibility during acute infection.** In the central analysis, transmission flows were estimated using the centre of gravity of the uncertainty region associated with the refined infection time estimates. To account for higher transmission rates during acute infection of the transmitting partner[87], we assumed that the transmission hazard was 5 times higher in the first 2 months after infection of the transmitting partner as

compared to the following period, and obtained the resulting mean infection time estimate under this assumption by generalizing our Monte Carlo approach used in the central analysis to an importance sampling approach under piecewise linear transmission hazards. The primary results were insensitive to these changes as less than 5% of source–recipient pairs were attributed to different survey rounds (Supplementary Table 10).

**Sensitivity in estimating transmission flows to right censoring of likely transmission pairs.** The RCCS transmission cohort was defined retrospectively and so it is possible that some transmission events, especially in later rounds, remain as of yet unseen because the corresponding individuals are not yet in the survey or do not yet have virus deep-sequenced. To assess sensitivity to right censoring, we excluded from analysis those source–recipient pairs for which virus of the source or the recipient was deep-sequenced only after rounds 17, 16 and 15. The primary findings were insensitive to these analyses because the probabilities of detecting infection events in the phylogenetic data changed accordingly (Supplementary Table 10 and Supplementary Fig. 5).

**Sensitivity in estimated transmission flows to limited sample size of likely transmission pairs.** The number of observed infection events in the incidence cohort was approximately four times larger than the number of reconstructed transmission events, prompting us to explore the effect of sampling uncertainty on the transmission flow estimates. We bootstrap sampled source–recipient pairs at random with replacement three times, and repeated inferences on these bootstrap samples. Our primary findings remained insensitive (Supplementary Table 10).

**Sensitivity in estimated transmission flows to modelled sampling estimates.** The sampling adjustments in equation (6) require assumptions including that sampling is independent of infection and transmission, independent between source and recipient, at random within strata, and well approximated by approximating sources with individuals with unsuppressed virus. We repeated flow inferences without any adjustments and without adjustments for potentially unequal sampling of sources. Our primary findings were insensitive across these analyses (Supplementary Table 10).

**Sensitivity in transmission flow estimates to the phylo-SI model specification.** In the central analysis, the log transmission rates that underpin the estimated transmission flows were estimated using the linear predictor in equation (7c), and this model specification was associated with overall smallest mean absolute error and posterior predictive coverage as shown in Supplementary Table 6 against the following alternative models,

$$\log \beta_{r,i,j}^{g \to h} = \hat{c}^{g \to h}(i,j) + \gamma_0 + \gamma_g + \gamma_r + \gamma_{p(r)} + f_0^{g \to h}(i,j) + f_{p(r)}^{g \to h}(i), \quad (12a)$$

$$\log \beta_{r,i,j}^{g \to h} = \hat{c}^{g \to h}(i,j) + \gamma_0 + \gamma_g + \gamma_r + \gamma_{p(r)} + f_0^{g \to h}(i,j) + f_{p(r)}^{g \to h}(j), \quad (12b)$$

$$\log \beta_{r,i,j}^{g \to h} = \hat{c}^{g \to h}(i,j) + \gamma_0 + \gamma_g + \gamma_r + \gamma_{p(r)} + f_0^{g \to h}(i,j) + f_{p(r)}^{g \to h}(i,j), \quad (12c)$$

$$\log \beta_{r,i,j}^{g \to h} = \hat{c}^{g \to h}(i,j) + \gamma_0 + \gamma_g + \gamma_r + \gamma_{p(r)} + f_0^{g \to h}(i,j) + f_r^{g \to h}(j) \quad (12d)$$

$$\log \beta_{r,i,j}^{g \to h} = \hat{c}^{g \to h}(i,j) + \gamma_0 + \gamma_g + \gamma_r + \gamma_{p(r)} + f_0^{g \to h}(i,j) \\ + f_r^{g \to h}(j) + f_{p(r)}^{g \to h}(j), \quad (12e)$$

$$\log \beta_{r,i,j}^{g \to h} = \hat{c}^{g \to h}(i,j) + \gamma_0 + \gamma_g + \gamma_r + \gamma_{p(r)} + f_0^{g \to h}(i,j) \\ + f_r^{g \to h}(j) + f_{p(r)}^{g \to h}(i,j). \quad (12f)$$

Models specifying transmission rates without a round-specific random function on the age of infected individuals, equations (12a)–(12c), did not fit the data well (Supplementary Table 6). The remaining models, equations (12d)–(12f), performed as well as the model used in the central analysis (Supplementary Table 6) and our primary findings remained insensitive (Supplementary Table 10).

**Sensitivity in counterfactual intervention impacts to assumptions on viral suppression levels in non-participants.** Infection and viremia in the non-participant census-eligible population remained unknown and in the central analysis, we considered as proxy of virus suppression levels among non-participants data from first-time participants. We performed two sensitivity analyses, assuming first that all non-participants with HIV were also viremic across all rounds, and assuming second that virus suppression was identical among non-participants and participants of the same age, gender and survey round. Together, the two scenarios likely encompass the true, unknown viral suppression levels in non-participants. These scenarios were implemented by updating the number of individuals with viremia in equation (2), and refitting the model. The sensitivity analysis assuming all non-participants with HIV were viremic resulted in larger predicted incidence reductions in women around 75%, while the sensitivity analysis assuming virus suppression was the same among non-participants as among participants of the same age, gender and survey round resulted in similar predicted incidence reductions in women than in the central analysis (Supplementary Table 10).

**Sensitivity in counterfactual intervention impacts to potentially higher HIV prevalence in non-participants.** In the central analysis, we assumed that HIV prevalence was the same in participants and non-participants of the same age, gender and survey round. We considered three sensitivity analyses, assuming first that prevalence was 25% higher in male non-participants compared to male participants of the same age, gender and survey round, assuming second that prevalence was 25% higher in female non-participants compared to female participants of the same age, gender and survey round, and assuming third that prevalence was 25% higher in female and male non-participants compared to female and male participants of the same age, gender and survey round respectively. These scenarios were implemented by updating the number of virally unsuppressed individuals in equation (2), and refitting the model. Our primary findings remained insensitive (Supplementary Table 10).

**Sensitivity in counterfactual intervention impacts to lower viral suppression thresholds.** Different definitions of HIV suppression are currently operational, and we considered the effect of lower thresholds to define viral suppression (<200 copies ml⁻¹) than in the central analysis (<1,000 copies ml⁻¹). This scenario was implemented by re-estimating the age- and gender-specific proportions of individuals with HIV in the study population who had suppressed virus at the lower threshold, re-calculating gaps in viral suppression levels in men relative to women, and re-calculating the additional number of men needed to reach and maintain viral suppression in the counterfactual intervention scenarios. We found slightly smaller gender gaps in viral suppression at the lower threshold and the predicted incidence reduction in women in the counterfactual that assessed closing the suppression gap in men was around 45%, and all other findings remained insensitive (Supplementary Table 10).

### Ethics statement

The study was independently reviewed and approved by the Ugandan Virus Research Institute, Scientific Research and Ethics Committee, protocol GC/127/13/01/16; the Ugandan National Council of Science and Technology; and the Western Institutional Review Board, protocol

200313317. All study participants provided written informed consent at baseline and follow-up visits using institutional review board approved forms. This project was reviewed in accordance with Centers for Disease Control and Prevention (CDC) human research protection procedures and was determined to be research, but CDC investigators did not interact with human subjects or have access to identifiable data or specimens for research purposes. Participants in RCCS received 10,000 UGX (approximately US$2.50) in compensation for the baseline and follow-up surveys.

### Reporting summary

Further information on research design is available in the Nature Portfolio Reporting Summary linked to this article.

### Data availability

Pseudo-anonymized data from the RCCS incidence and transmission cohort as well as pseudo-anonymized deep-sequence phylogenies to reproduce all analyses are available from Zenodo (https://zenodo.org/record/8412741) as open-access dataset under the CC-BY-4.0 license[91]. HIV consensus sequences are available from Zenodo (https://zenodo.org/records/10075815) and the PANGEA-HIV sequence repository (https://github.com/PANGEA-HIV/PANGEA-Sequences) as open-access dataset under the CC-BY-4.0 license[92], with identifiers changed to ensure participants cannot be identified from this dataset.
Additional deep-sequence HIV-1 reads can be requested from PANGEA-HIV under a managed access policy due to privacy and ethical reasons, which aligns with UNAIDS ethical guidelines. The process for accessing data, the PANGEA-HIV data sharing policy and a detailed description of what data are available is laid out in full at (https://www.pangea-hiv.org/join-us). Briefly, applicants can apply to receive additional data by submitting a concept sheet proposal in which they explain the research question and how they will mitigate potential risks to participant privacy. In line with requirements for PANGEA members, applicants will be asked to present proof of human subject research training and comply with PANGEA-internal publication agreements. PANGEA encourages external applicants to collaborate with the researchers who generated the data. For more information contact PANGEA project manager Lucie Abeler-Dörner (lucie.abeler-dorner@bdi.ox.ac.uk). The time frame for a response to requests is 2–4 weeks.
Additional cohort data can be requested from RHSP. Because of privacy and ethical reasons, RHSP maintains a controlled access data policy for corresponding epidemiological metadata and corresponding data collection tools. In brief, RHSP policy requires individuals to submit an RHSP data request form (available upon request from info@rhsp.org or gkigozi@rhsp.org) and a brief concept note (one or two pages) detailing their research questions and methods. In addition, researchers are asked to provide a curriculum vitae/resume along with proof of human subjects research training. Concept sheets can be submitted to Godfrey Kigozi (gkigozi@rhsp.org), executive director of the RHSP. Only individuals named on the original data request and who provide the request, resume and human subjects research training, are permitted access to the data. Released data are not to be reused for other purposes outside of approved concepts. The time frame for a response to requests is 2–4 weeks.

### Code availability

Code to reproduce all analyses is freely available on GitHub version 1.1.2 under the GNU General Public License version 3.0 at the repository (https://github.com/MLGlobalHealth/phyloSI-RakaiAgeGender).

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

## Acknowledgements

We thank all contributors, programme staff and participants to the Rakai Community Cohort Study; all members of the PANGEA-HIV consortium, the Rakai Health Sciences Program, and CDC Uganda for comments on an earlier version of the manuscript; the Imperial College Research Computing Service (https://doi.org/10.14469/hpc/2232) and the Biomedical Research Computing Cluster at the University of Oxford for providing the computational resources to perform this study; the Office of Cyberinfrastructure and Computational Biology at the National Institute for Allergy and Infectious Diseases for data management support; and Zulip for sponsoring team communications through the Zulip Cloud Standard chat app. This study was supported by the Bill and Melinda Gates Foundation (OPP1175094 to C.F., OPP1084362 to D. Pillay); the National Institute of Allergy and Infectious Diseases (U01AI051171 to R.H.G., U01AI075115 to R.H.G., UM1AI069530-16 to M. G. Fowler, P. Musoke and A. Tobian, R01AI087409 to R.H.G., U01AI100031 to R.H.G., R01AI110324 to R.H.G., R01AI114438 to M.J.W., K25AI114461 to X. Kong, R01AI123002 to C. Liu, K01AI125086 to M.K.G., R01AI128779 A. Tobian, R01AI143333 to L.W.C., R21AI145682 to C. Kennedy, R01AI155080 to M.K.G., ZIAAI001040 to T.C.Q.); the National Institute of Mental Health (F31MH095649 to J. Wagman, R01MH099733 to N. Sacktor and M.J.W., R01MH107275 to L.W.C., R01MH115799 to M.J.W. and L.W.C., U19MH110001 to M. McKay and F. Ssewamala); the National Institute of Child Health and Development (R01HD038883 to R.H.G., R01HD050180 to M.J.W., R01HD070769 to M.J.W., R01HD091003 to J. Santelli); the Division of Intramural Research of the National Institute for Allergy and Infectious Diseases (T.C.Q., O.L. and S.J.R.); NIAID (K01AA024068 to J. Wagman); the National Heart, Lung, and Blood Institute (R01HL152813 to L.W.C.); the Fogarty International Center (D43TW009578 to R.H.G., D43TW010557 to L.W.C.); the Doris Duke Charitable Foundation to A. Tobian; the Johns Hopkins University Center for AIDS Research (P30AI094189 to R. Chaisson); the US President's Emergency Plan for AIDS Relief (PEPFAR) through the Centers for Disease Control and Prevention (NU2GGH000817 to Rakai Health Sciences Program); the Engineering and Physical Sciences Research Council through the EPSRC Centre for Doctoral Training in Modern Statistics and Statistical Machine Learning at Imperial College London and Oxford University (EP/S023151/1 to A. Gandy); and the Imperial College London President's PhD Scholarship fund to Y.C. The funders had no role in study design, data collection and analysis, decision to publish or preparation of the manuscript. The findings and conclusions in this report are those of the author(s) and do not necessarily represent the official position of the Centers for Disease Control and Prevention.

## Author contributions

O.R. and M.K.G. designed the study. O.R., M.K.G., J.K., P.G.-F. and D.S. oversaw the study. R.M.G., R.S., E.N.K., V.S., L.A.-D., D.B., L.W.C., C.F., T.G., R.H.G., J.C.J., G.K., L.A.M., O.L., T.C.Q., S.J.R., J. Santelli, N.K.S., J. Ssekasanvu, L.T., M.J.W., D.S., J.K. and M.K.G. oversaw and performed data collection. M.M., A. Brizzi, X.X., E.N.K., V.S., A.A., A. Blenkinsop, Y.C., S.D., T.G., M.H., S.E.F.S., L.T., M.K.G. and O.R. contributed to the analysis. M.M., A. Brizzi, X.X., A. Blenkinsop, Y.C., M.K.G. and O.R. wrote the first draft. All authors discussed the results and contributed to the revision of the final manuscript.

## Competing interests

O.R., M.K.G., C.F. and D.B. report grants from the Bill and Melinda Gates Foundation during the conduct of this study. M.K.G., M.J.W., R.H.G. and L.W.C. report grants from the National Institutes of Health during the conduct of this study. A. Brizzi, Y.C. and X.X. report an EPSRC PhD studentship during the conduct of this study. M.J.W. and R.H.G. are paid consultants to the Rakai Health Sciences Program and serves on its board of directors. This arrangement has been reviewed and approved by the Johns Hopkins University in accordance with its conflict of interest policies.

## Additional information

**Extended data** is available for this paper at https://doi.org/10.1038/s41564-023-01530-8.

**Correspondence and requests for materials** should be addressed to Joseph Kagaayi, M. Kate Grabowski or Oliver Ratmann.

**Mélodie Monod** [1,21], **Andrea Brizzi** [1,21], **Ronald M. Galiwango** [2,21], **Robert Ssekubugu** [2,21], **Yu Chen** [1,21], **Xiaoyue Xi** [1,21], **Edward Nelson Kankaka** [3,4,21], **Victor Ssempijja** [5,6,21], **Lucie Abeler-Dörner** [7], **Adam Akullian** [8], **Alexandra Blenkinsop** [1], **David Bonsall** [9,10], **Larry W. Chang** [2,3,11], **Shozen Dan** [1], **Christophe Fraser** [7,10], **Tanya Golubchik** [7,12], **Ronald H. Gray** [11], **Matthew Hall** [7], **Jade C. Jackson** [13], **Godfrey Kigozi** [2], **Oliver Laeyendecker** [14,15], **Lisa A. Mills** [16], **Thomas C. Quinn** [13,14,15], **Steven J. Reynolds** [2,14,15], **John Santelli** [17], **Nelson K. Sewankambo** [18], **Simon E. F. Spencer** [19], **Joseph Ssekasanvu** [11], **Laura Thomson** [7], **Maria J. Wawer** [2,11], **David Serwadda** [2,18], **Peter Godfrey-Faussett** [20,22], **Joseph Kagaayi** [2] ✉, **M. Kate Grabowski** [2,11,13,22] ✉, **Oliver Ratmann** [1,22] ✉, **Rakai Health Sciences Program\* & PANGEA-HIV consortium\***

[1]Department of Mathematics, Imperial College London, London, UK. [2]Rakai Health Sciences Program, Kalisizo, Uganda. [3]Division of Infectious Diseases, Johns Hopkins School of Medicine, Baltimore, MD, USA. [4]Research Department, Rakai Health Sciences Program, Rakai, Uganda. [5]Clinical Monitoring Research Program Directorate, Frederick National Laboratory for Cancer Research, Frederick, MD, USA. [6]Statistics Department, Rakai Health Sciences Program, Rakai, Uganda. [7]Big Data Institute, University of Oxford, Oxford, UK. [8]Bill and Melinda Gates Foundation, Seattle, WA, USA. [9]Wellcome Centre for Human Genomics, Nuffield Department of Medicine, University of Oxford, Oxford, UK. [10]Pandemic Sciences Institute, University of Oxford, Oxford, UK. [11]Department of Epidemiology, Johns Hopkins Bloomberg School of Public Health, Baltimore, MD, USA. [12]Sydney Infectious Diseases Institute, School of Medical Sciences, Faculty of Medicine and Health, University of Sydney, Sydney, Australia. [13]Department of Pathology, Johns Hopkins School of Medicine, Baltimore, MD, USA. [14]Division of Intramural Research, National Institute of Allergy and Infectious Diseases, National Institutes of Health, Bethesda, MD, USA. [15]Department of Medicine, Johns Hopkins School of Medicine, Baltimore, MD, USA. [16]Division of Global HIV and TB, US Centers for Disease Control and Prevention, Kampala, Uganda. [17]Population and Family Health and Pediatrics, Columbia Mailman School of Public Health, New York, NY, USA. [18]College of Health Sciences, School of Medicine, Makerere University, Kampala, Uganda. [19]Department of Statistics, University of Warwick, Coventry, UK. [20]Department of Infectious and Tropical Diseases, London School of Hygiene & Tropical Medicine, London, UK. [21]These authors contributed equally: Mélodie Monod, Andrea Brizzi, Ronald M. Galiwango, Robert Ssekubugu, Yu Chen, Xiaoyue Xi, Edward Nelson Kankaka, Victor Ssempijja. [22]These authors jointly supervised this work: Peter Godfrey-Fausett, M. Kate Grabowski, Oliver Ratmann. \*A list of authors and their affiliations appears at the end of the paper. ✉e-mail: jkagayi@rhsp.org; mgrabow2@jhu.edu; oliver.ratmann@imperial.ac.uk

**Rakai Health Sciences Program**

Larry W. Chang[2,3,11], Ronald M. Galiwango[2,21], M. Kate Grabowski[2,11,13], Ronald H. Gray[11], Jade C. Jackson[13], Joseph Kagaayi[2], Edward Nelson Kankaka[3,4,21], Godfrey Kigozi[2], Oliver Laeyendecker[14,15], Thomas C. Quinn[13,14,15], Steven J. Reynolds[2,14,15], John Santelli[17], David Serwadda[2,18], Nelson K. Sewankambo[18], Joseph Ssekasanvu[11], Robert Ssekubugu[2,21], Victor Ssempijja[5,6,21] & Maria J. Wawer[2,11]

**PANGEA-HIV consortium**

Lucie Abeler-Dörner[7], David Bonsall[9,10], Christophe Fraser[7,10], Tanya Golubchik[7,12], M. Kate Grabowski[2,11,13,22], Joseph Kagaayi[2], Thomas C. Quinn[13,14,15], Oliver Ratmann[1,22] & Maria J. Wawer[2,11]

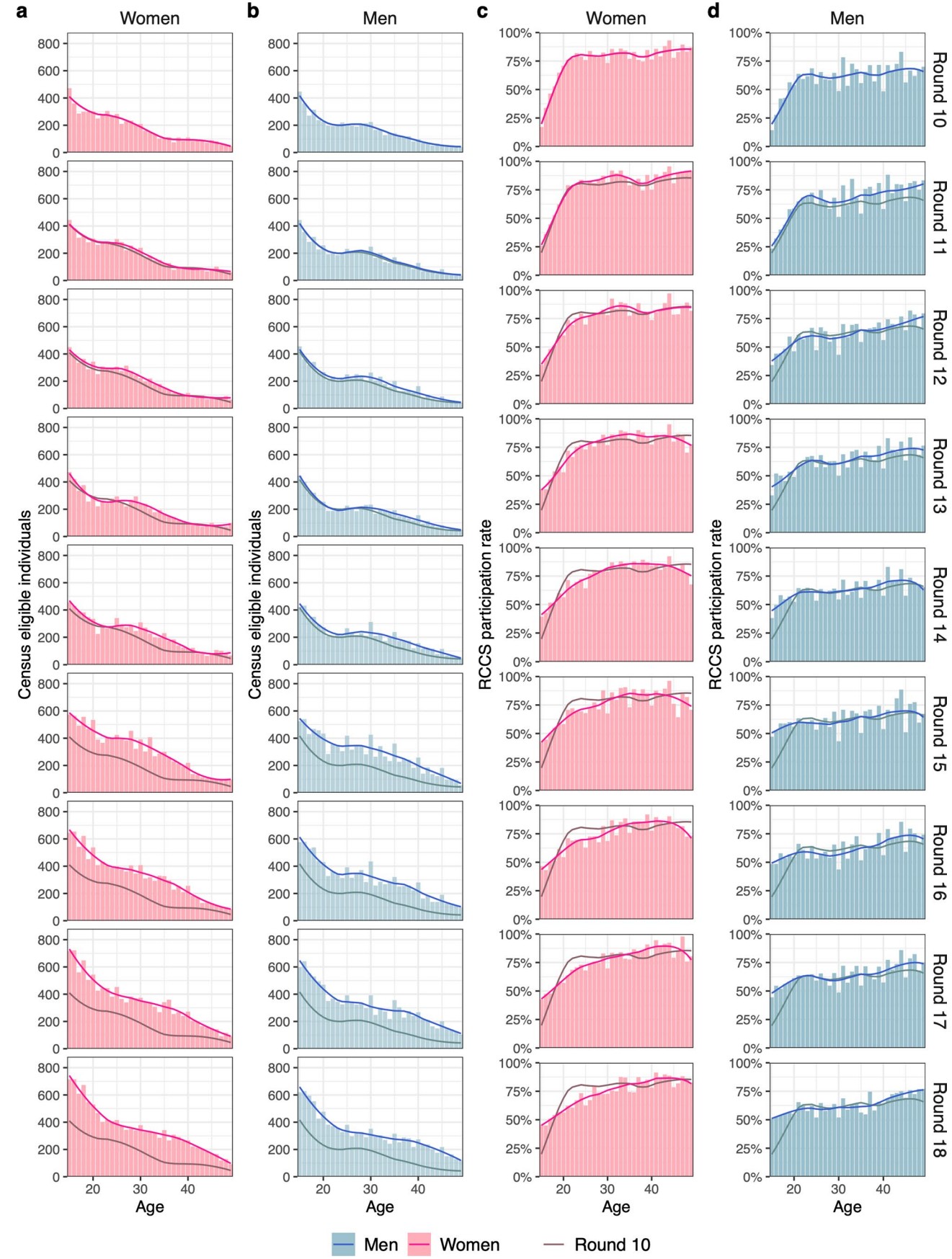

**Extended Data Fig. 1 | See next page for caption.**

**Extended Data Fig. 1 | Characteristics of the RCCS study population by age, gender, and time.** (**a-b**) Population size. Counts of the aggregated individual-level census data by 1-year age group (bars) are shown along LOESS smoothed population size estimates (line) for men and women (see text). (**c-d**) RCCS participation rates. Rates relative to the aggregated census data by 1-year age band (bars) are shown along LOESS smoothed participation rates (line) for both men and women. For reference, round 10 values are indicated in each subsequent plot in darker colours. The timeline of the survey rounds is shown in Fig. 1b.

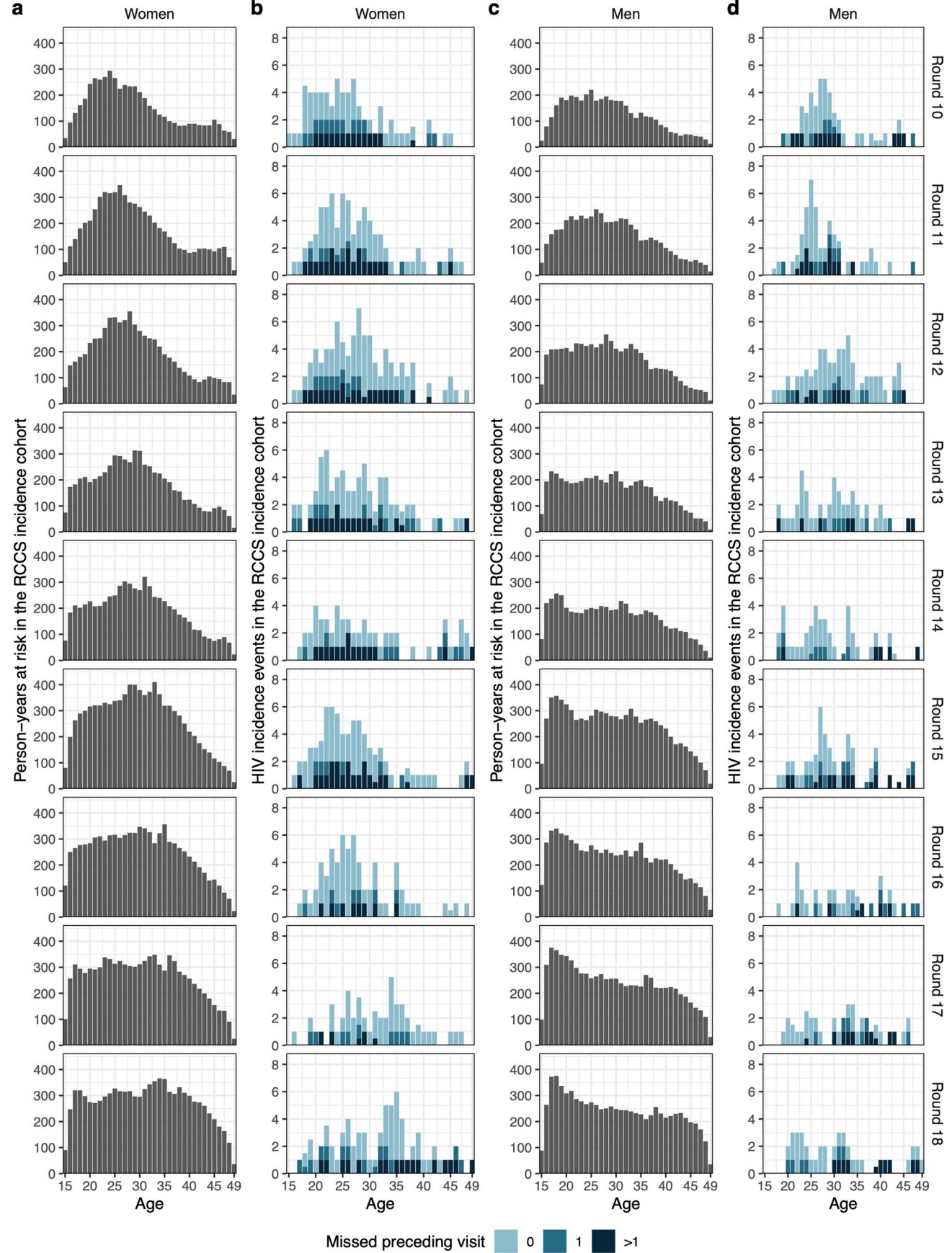

**Extended Data Fig. 2 | Age- and gender-specific person-years at risk and HIV incidence events in the RCCS incidence cohort.** Person-years at risk in the RCCS incidence cohort among (**a**) women and (**c**) men. HIV incidence events in the RCCS incidence cohort among (**b**) women and (**d**) men.

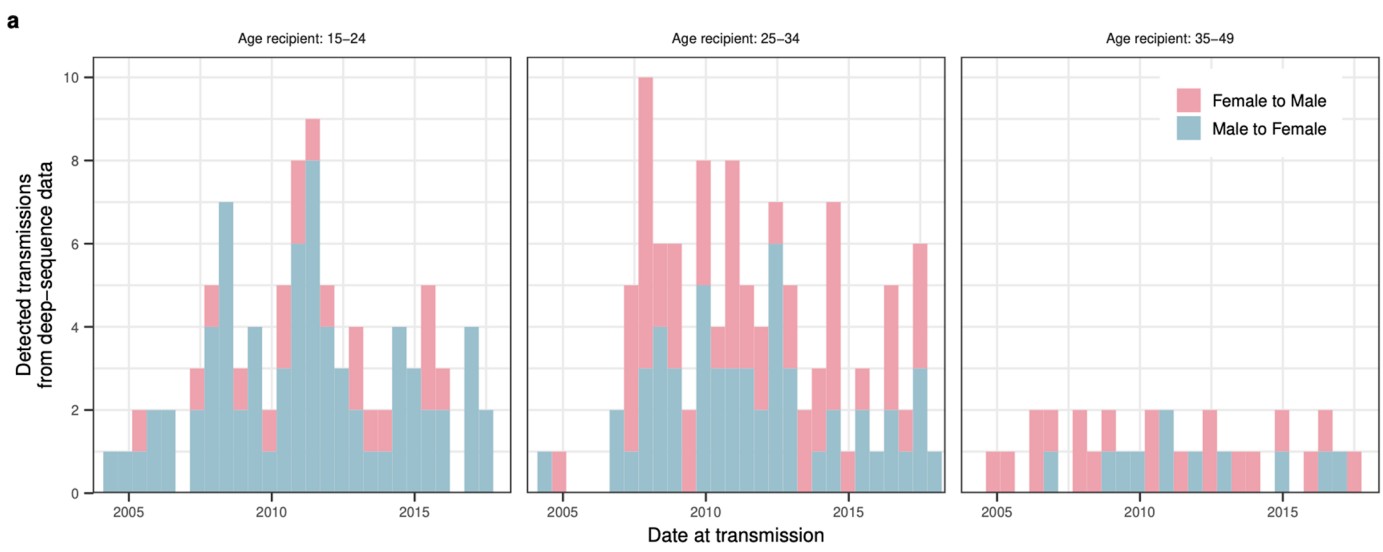

**Extended Data Fig. 3 | Phylogenetically reconstructed source–recipient pairs.** (a) Number of heterosexual source–recipient pairs by the date of infection of the recipient (x-axis), the age of the recipient at infection (panel), and transmission direction (colour). (b) Heterosexual source–recipient pairs by the age of the recipient (x-axis) and the age of the source (y-axis) at the median infection time estimate by the round (colour) in which transmission was estimated to have occurred. The number of phylogenetically reconstructed source–recipient pairs is indicated in the top-left corner.

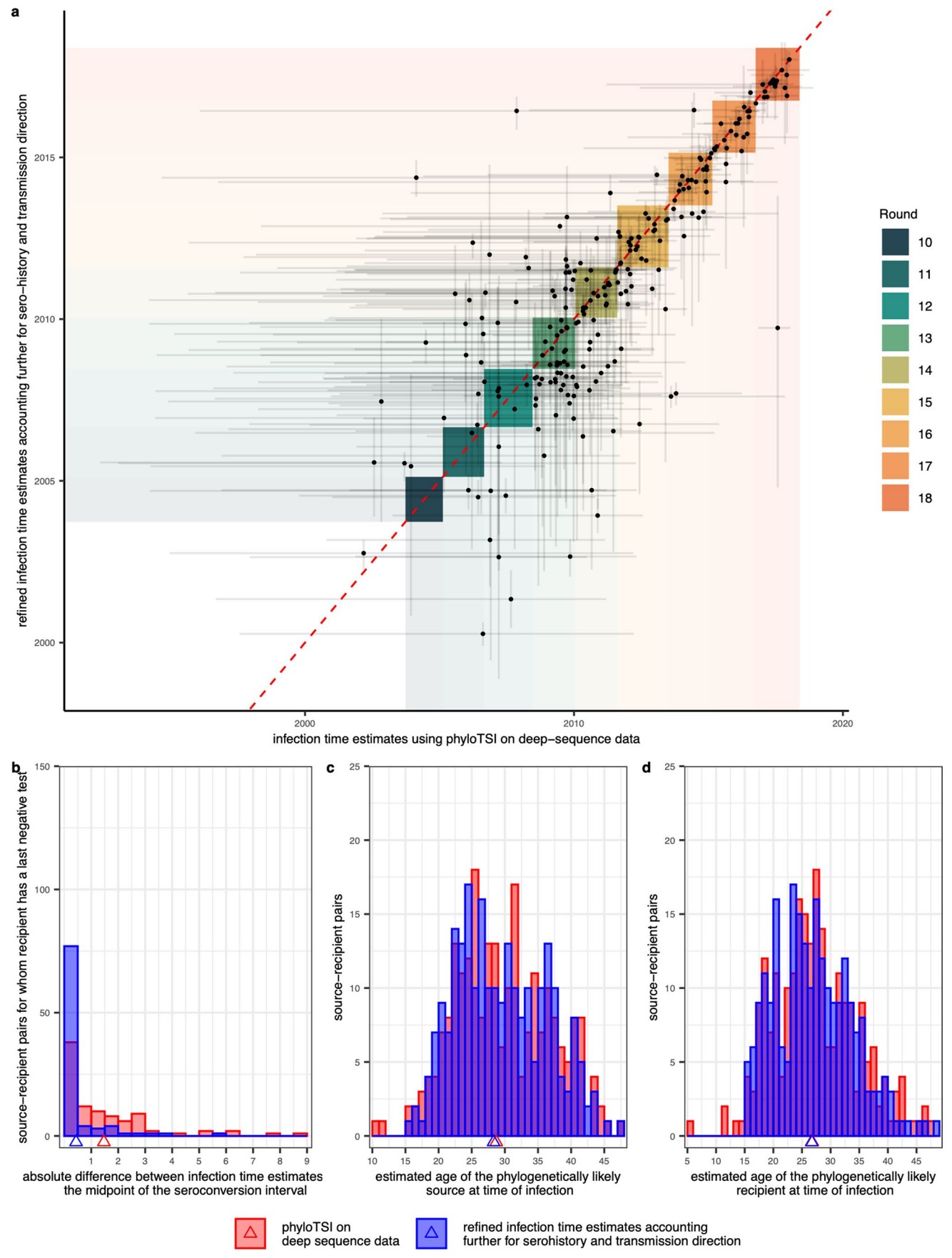

**Extended Data Fig. 4 | See next page for caption.**

**Extended Data Fig. 4 | Comparison of estimated infection dates in phylogenetically reconstructed source–recipient pairs.** (**a**) Estimated infection times of the recipient in the n = 227 phylogenetically reconstructed source–recipient pairs from `phyloTSI` based on deep-sequence data alone (x-axis) against refined estimates accounting for sero-history and inferred direction of transmission (y-axis). Median estimates (dots) are shown along 95% uncertainty ranges (lines). (**b**) Histogram of absolute difference (bars) and mean absolute differences (triangle) between infection time estimates and the midpoint of seroconversion intervals in 98 source–recipient pairs in which the recipient had a last negative test, across the two methods (colour). (**c**) Histogram (bars) and median (triangle) age of the phylogenetically likely recipient, across the two methods (colours). (**d**) Histogram (bars) and median (triangle) age of the phylogenetically likely source, across the two methods (colours).

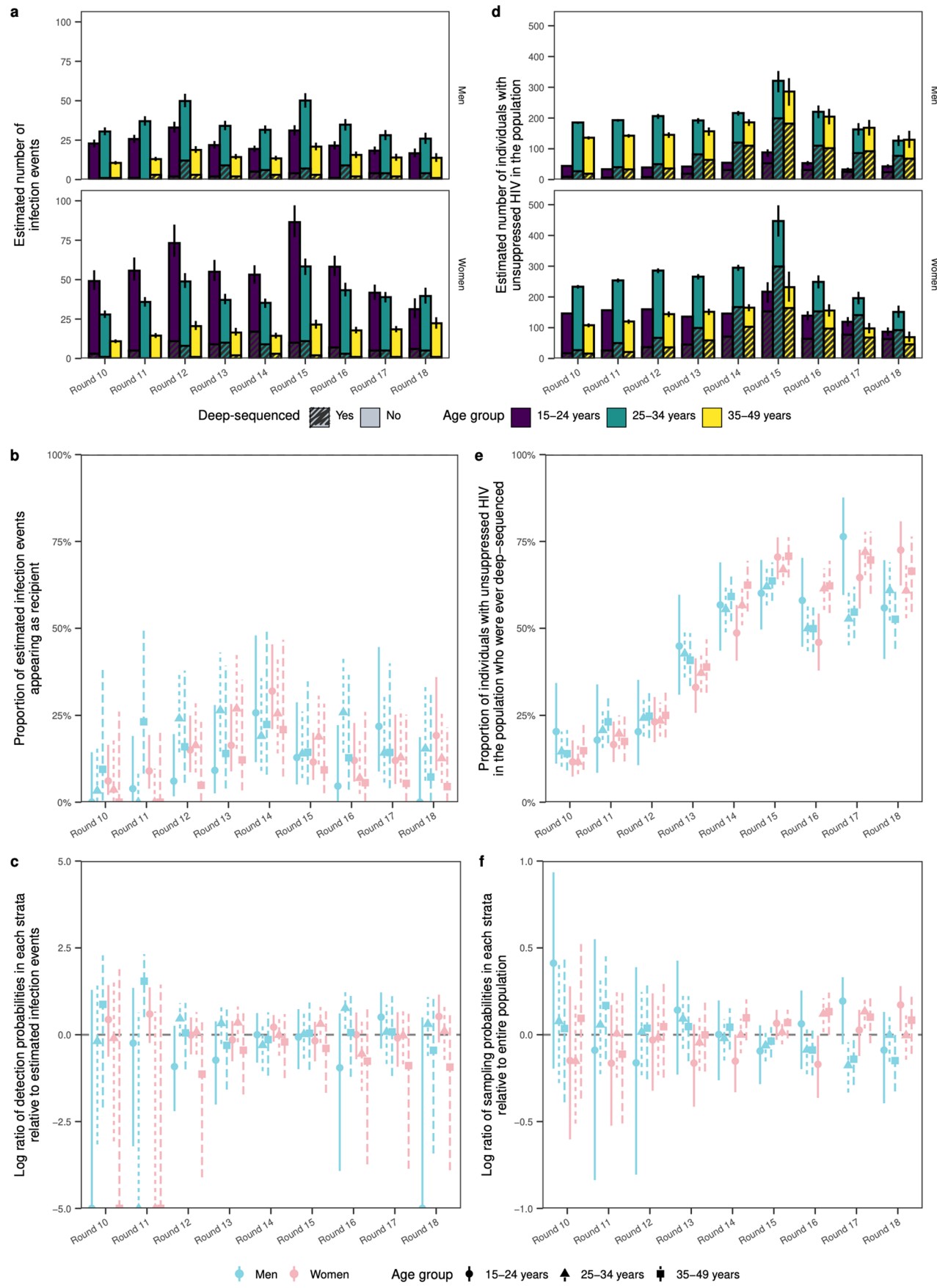

**Extended Data Fig. 5 | See next page for caption.**

**Extended Data Fig. 5 | Sampling estimates of transmission events and sources of infections.** (**a-c**) The sampling cascade of transmission events was modelled by comparing the number of phylogenetically reconstructed source–recipient events to the estimated number of infection events under the incidence model, by gender, age band and survey round of infected individuals. Throughout, shown are the number of sampled and unsampled transmission events, the estimated proportion of transmission events that were ever deep-sequenced, and log ratios of estimated proportion of transmission events that were ever deep-sequenced in any strata relative to the overall average across strata (point estimates: dots, 95% confidence intervals: linebars). Estimates are based on n = 227 source–recipient pairs and n = 1,117 observed incidence events. (**d-f**)

Additional differences in source sampling were modelled by considering unsuppressed individuals as potential sources of infection, and calculating the number of unsuppressed individuals in a round that were ever deep-sequenced. Throughout, shown are the number of sampled and unsampled possible transmission sources, the estimated proportion of possible sources that were ever deep-sequenced, and log ratios of estimated proportion of possible sources that were ever deep-sequenced in any strata relative to the overall average across strata (point estimates: dots, 95% confidence intervals: linebars). Estimates are based on n = 227 source–recipient pairs and n = 3,265 participants with HIV and measured viral load.

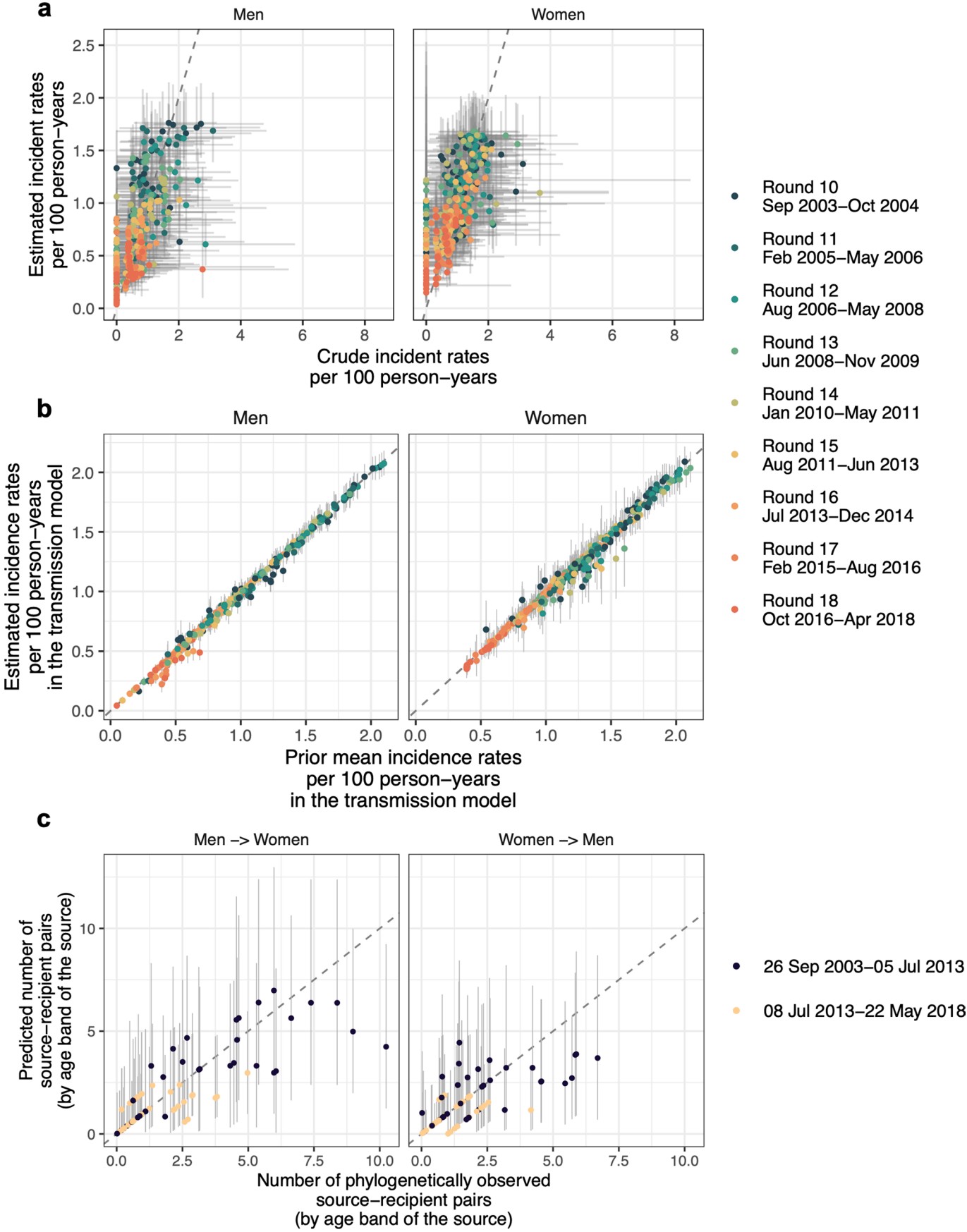

**Extended Data Fig. 6 | See next page for caption.**

**Extended Data Fig. 6 | Validation of the incidence rate and transmission flow models.** (**a**) Empirical HIV incidence rates were obtained for each of the 50 data sets with imputed exposure times and compared to the estimated HIV incidence rates under the Poisson model. The median (point) and 95% range (horizontal error bars) of the crude HIV incidence rates are plotted against the posterior median (point) and 95% range (vertical error bars) of estimated HIV incidence rates for each gender, age and round. (**b**) Prior incidence rates as specified according to the outputs of the incidence rate and used in the transmission model are compared versus the posterior incidence rates obtained with the transmission model. Shown are medians (point) and 95% credible intervals (error bars) by gender, age and round. (**c**) Observed transmission flow counts are compared to posterior predictive estimates under the transmission model. Shown are medians (point) and 95% credible intervals (error bars) by direction of transmission, time period, and age of the phylogenetically likely source. Throughout all subfigures, empirical and modelled incidence estimates are based on n = 1,117 individuals in the incidence cohort and n = 227 source–recipient pairs among n = 1,978 individuals in the transmission cohort.

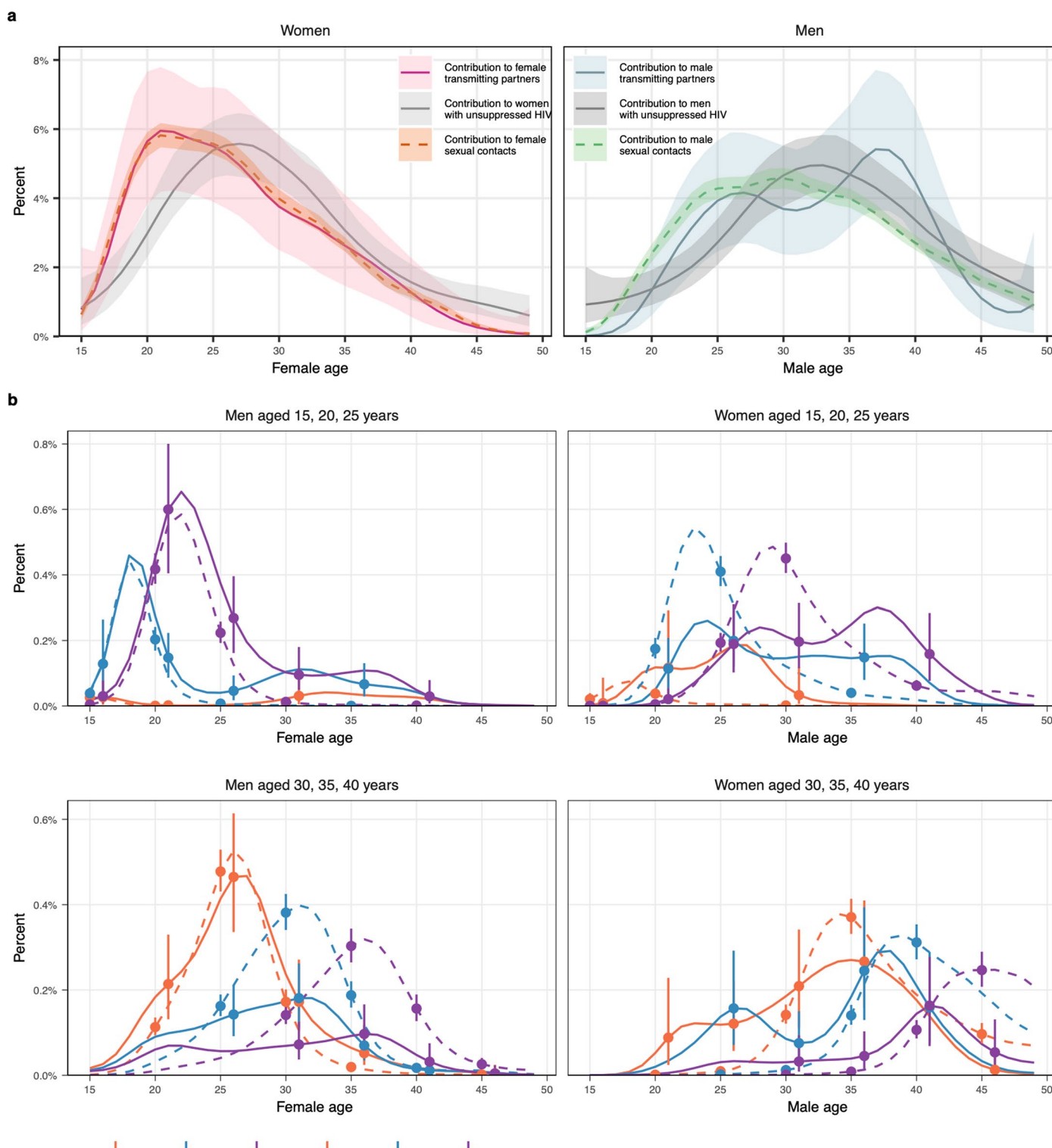

**Extended Data Fig. 7 | Age contributions to sexual contacts, viral suppression and transmission. (a)** Estimated age contributions from women to men of all ages (left) and from men to women of all ages (right) to sexual contacts in round 15, viral suppression in round 18, and transmission in round 18 (posterior median: line, 95% credible interval: ribbon). Age contributions sum to 100% separately for women and men. **(b)** Estimated age contributions from women to men of specific ages (left) and from men to women of specific ages (right) to sexual contacts in round 15, and transmission in round 18 (posterior median: line, 95% credible interval: error bars). Age contributions sum to 100% for women and men combined. Throughout all subfigures, empirical and modelled incidence estimates are based on n = 1,117 individuals in the incidence cohort and n = 227 source–recipient pairs among n = 1,978 individuals in the transmission cohort.

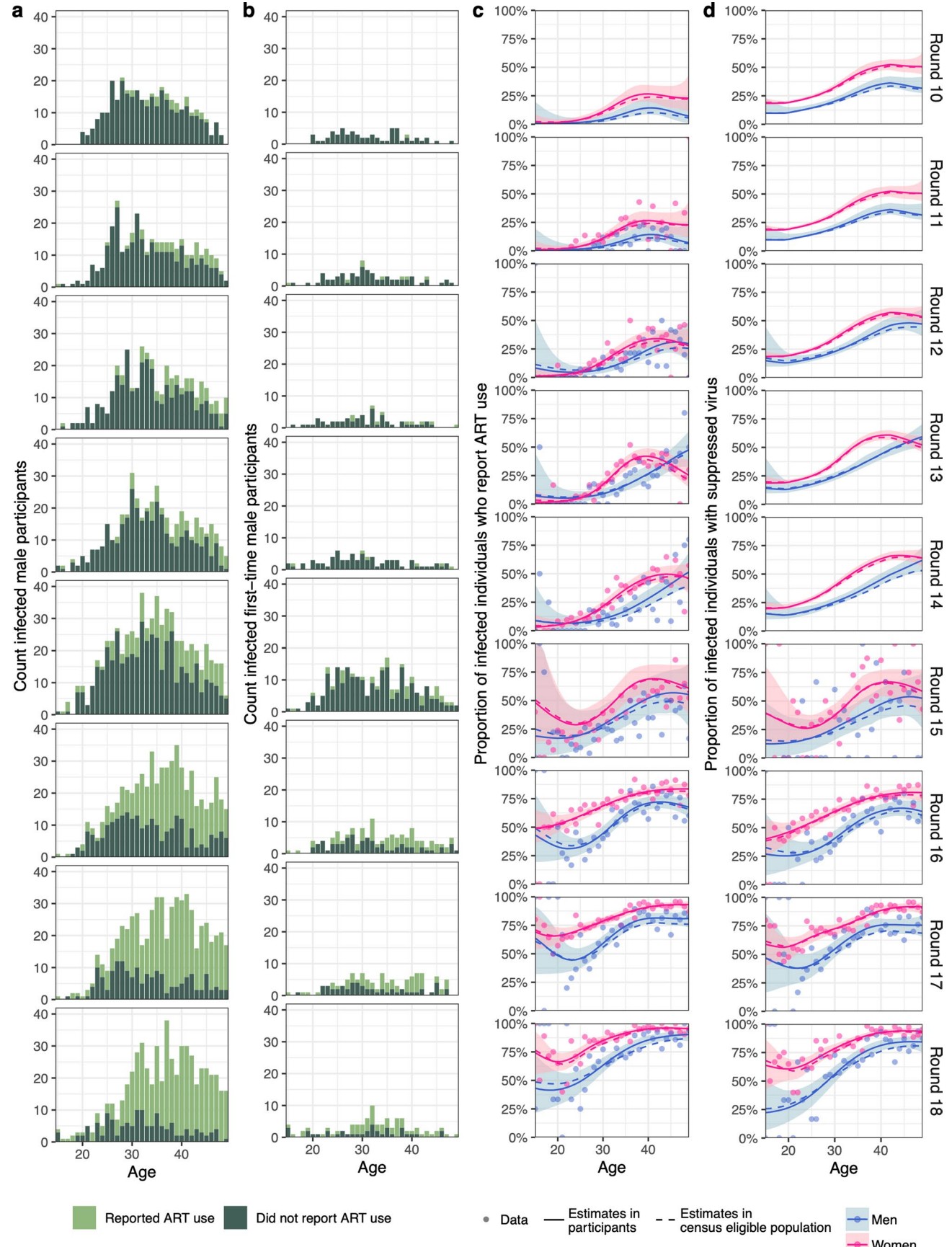

Extended Data Fig. 8 | See next page for caption.

**Extended Data Fig. 8 | ART use and virus suppression in the RCCS study population by age, gender, and time.** (**a**) HIV-positive male participants, by whether they reported ART use (colour), by 1-year age band (x-axis) and survey round (rows). (**b**) HIV-positive male first-time participants, by whether they reported ART use (colour), by 1-year age band (x-axis) and survey round (panel). (**c**) Estimates of ART use in men (blue) and women (pink) in the study population by 1-year of age. Data from participants (dots) are shown along smoothed posterior median estimates (solid line) and 95% credible intervals (ribbon) in participants, and along posterior median estimates in the census-eligible population (dashed line), using data from first-time participants as proxy of ART use in non-participants (see text). (**d**) Estimates of virus suppression, defined as a viral load measurement below 1,000 copies of HIV per millilitre plasma blood, in men (blue) and women (pink) in the study population by 1-year of age. Data from participants (dots) are shown along smoothed posterior median estimates (solid line) and 95% credible intervals (ribbon) in participants, and along posterior median estimates in the census-eligible population (dashed line), using data from first-time participants as proxy of ART use in non-participants. Throughout all subfigures, estimates are based on n = 38,749 participants including n = 3,924 participants with HIV who report ART status and n = 3,265 participants with HIV and measured viral load.

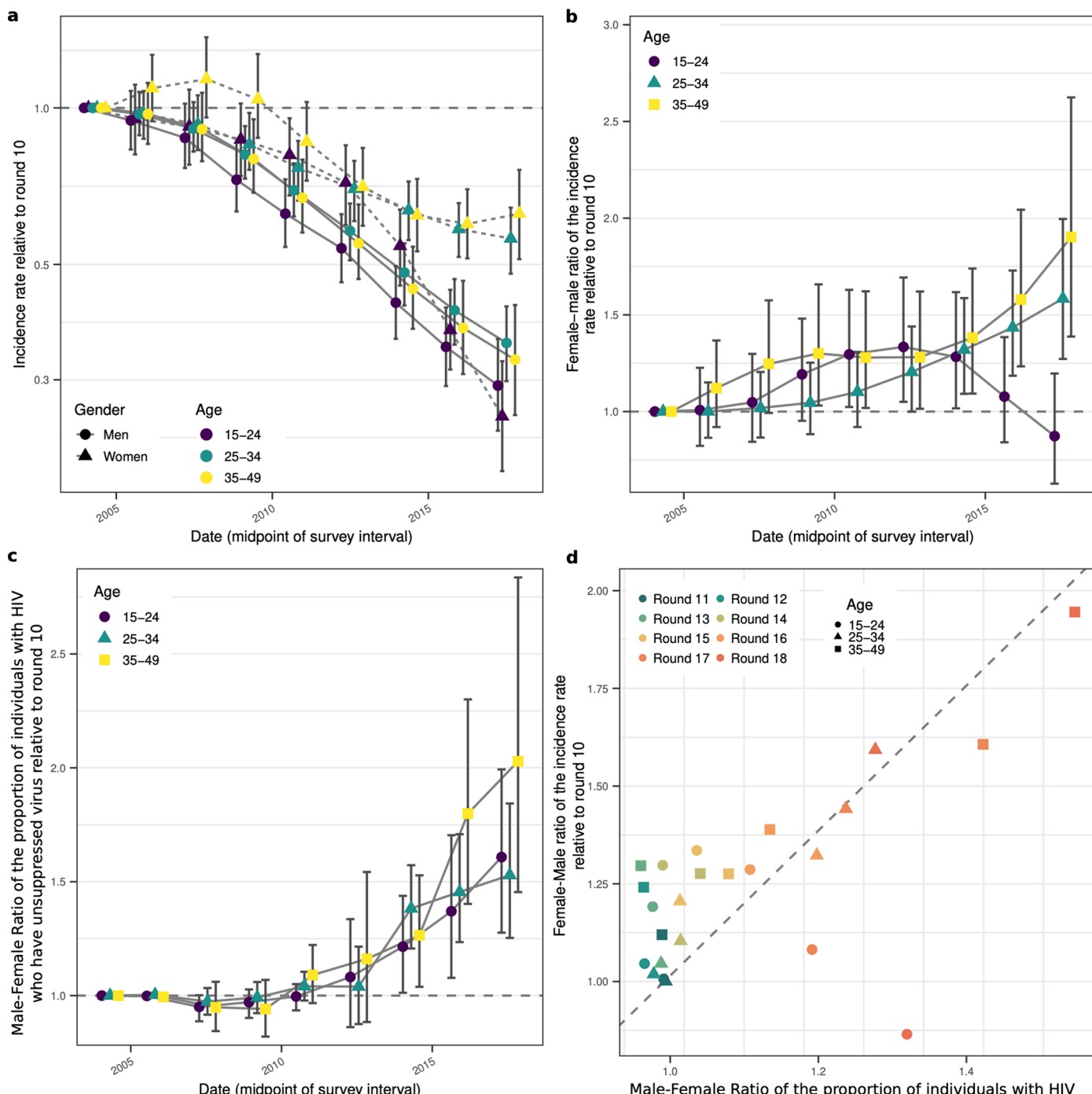

**Extended Data Fig. 9 | Longitudinal changes in viral suppression and incidence rates in the RCCS study population since 2003.** (a) Changes in incidence rates relative to round 10, that is Sep 2003 to Oct 2004 (posterior median: dots, 95% confidence interval: error bars). (b) Female-to-male ratio in changes in incidence rates relative to round 10 (posterior median: dots, 95% credible interval: error bars). (c) Male-to-female ratio in changes in the proportion of individuals with HIV who have unsuppressed virus relative to round 10 (posterior median: dots, 95% credible interval: error bars). (d) Scatter plot between the female-to-male ratio in changes in incidence rates as shown in (b) and the male-to-female ratio in changes in the proportion of individuals with HIV who have unsuppressed virus relative to round 10 as shown in (c). Throughout all subfigures, estimates are based on n = 1,117 individuals in the incidence cohort and n = 3,265 participants with HIV and measured viral load.

# Reporting Summary

## Statistics

For all statistical analyses, confirm that the following items are present in the figure legend, table legend, main text, or Methods section.

| n/a | Confirmed | |
|---|---|---|
| ☐ | ☒ | The exact sample size (*n*) for each experimental group/condition, given as a discrete number and unit of measurement |
| ☐ | ☒ | A statement on whether measurements were taken from distinct samples or whether the same sample was measured repeatedly |
| ☐ | ☒ | The statistical test(s) used AND whether they are one- or two-sided<br>*Only common tests should be described solely by name; describe more complex techniques in the Methods section.* |
| ☐ | ☒ | A description of all covariates tested |
| ☐ | ☒ | A description of any assumptions or corrections, such as tests of normality and adjustment for multiple comparisons |
| ☐ | ☒ | A full description of the statistical parameters including central tendency (e.g. means) or other basic estimates (e.g. regression coefficient) AND variation (e.g. standard deviation) or associated estimates of uncertainty (e.g. confidence intervals) |
| ☐ | ☒ | For null hypothesis testing, the test statistic (e.g. *F*, *t*, *r*) with confidence intervals, effect sizes, degrees of freedom and *P* value noted<br>*Give P values as exact values whenever suitable.* |
| ☐ | ☒ | For Bayesian analysis, information on the choice of priors and Markov chain Monte Carlo settings |
| ☐ | ☒ | For hierarchical and complex designs, identification of the appropriate level for tests and full reporting of outcomes |
| ☒ | ☐ | Estimates of effect sizes (e.g. Cohen's *d*, Pearson's *r*), indicating how they were calculated |

*Our web collection on statistics for biologists contains articles on many of the points above.*

## Software and code

Policy information about availability of computer code

| Data collection | For sequencing, the following software were used, QuantStudio Real-Time PCR System v1.3, Agilent TapeStation Software Analysis 4.1.1, Clarity Version 4.2.23.287, FreezerPro® (7.4.0-r14598), LabArchives ELN (Electronic Lab Notebook) 2023. All epidemiologic data collected through the Rakai Community Cohort Study are stored in a database running Microsoft SQL server 2019 and Microsoft Access version 2016. |
|---|---|
| Data analysis | All data were analyzed with R version 4.1.2, the R package stats version 3.6.2, the R package Rstan version 2.21.0, the R package cmdstanR version 0.5.1, the R package mgcv version 1.8-38, shiver version 1.5.7, phyloscanner version 1.8.1, MAFFT version 7.475, IQ-Tree version 2.0.3, phyloTSI version 1.0.0, ; and using the custom code scripts freely available at https://github.com/MLGlobalHealth/phyloSI-RakaiAgeGender. Team communications were supported through the Zulip chat app 5.10.2 (https://zulip.com/). |

For manuscripts utilizing custom algorithms or software that are central to the research but not yet described in published literature, software must be made available to editors and reviewers. We strongly encourage code deposition in a community repository (e.g. GitHub). See the Nature Portfolio guidelines for submitting code & software for further information.

# Data

Policy information about availability of data

All manuscripts must include a data availability statement. This statement should provide the following information, where applicable:

- Accession codes, unique identifiers, or web links for publicly available datasets
- A description of any restrictions on data availability
- For clinical datasets or third party data, please ensure that the statement adheres to our policy

Pseudo-anonymised data from the RCCS incidence and transmission cohort as well as pseudo-anonymised deep-sequence phylogenies to reproduce all analyses are available from Zenodo (https:/zenodo.org/record/8412741) as open-access data set under the CC-BY-4.0 license. HIV consensus sequences are available from Zenodo (https://zenodo.org/records/10075815) and the PANGEA-HIV sequence repository (https://github.com/PANGEA-HIV/PANGEA-Sequences) as open-access data set under the CC-BY-4.0 license, with identifiers changed to ensure participants cannot be identified from this data set.

Additional deep-sequence HIV-1 reads can be requested from PANGEA-HIV under a managed access policy due to privacy and ethical reasons, as the risks to the participants outweigh the benefits. The risk is to accidentally disclose evidence of transmission, or of not transmission, therefore making light evidence of sexual contact and transmission that could jeopardise relationships, and in some instances lead to criminalisation which is against UNAIDS ethical guidelines. The process for accessing data, the PANGEA-HIV Data Sharing Policy and a detailed description of what data are available is laid out in full at (https://www.pangea-hiv.org/join-us). Briefly, applicants can apply to receive additional data by submitting a concept sheet proposal in which they explain the research question and how they will mitigate potential risks to participant privacy. In line with requirements for PANGEA members, applicants will be asked to present proof of human subject research training and comply with PANGEA-internal publication agreements. PANGEA encourages external applicants to collaborate with the researchers who generated the data. For more information contact PANGEA project manager Lucie Abeler-Dörner (lucie.abeler-dorner@bdi.ox.ac.uk). The time frame for a response to requests is 2-4 weeks.

Additional cohort data can be requested from RHSP. Because HIV transmission is criminalized in Uganda and due to further privacy considerations, RHSP maintains a controlled access data policy for corresponding epidemiological metadata and corresponding data collection tools. In brief, RHSP policy requires individuals to submit an RSHSP data request form (available upon request) and a brief concept note (1-2 pages) detailing their research questions and methods. In addition, researchers are asked to provide a curriculum vitae/resume along with proof of human subjects research training. Concept sheets can be submitted to Dr. Godfrey Kigozi (gkigozi@rhsp.org), executive director of the RHSP. Only individuals named on the original data request and who provide the request, CV/resume and HSR training, are permitted access to the data. Released data are not to be reused for other purposes outside of approved concepts. The time frame for a response to requests is 2-4 weeks.

# Human research participants

Policy information about studies involving human research participants and Sex and Gender in Research.

| | |
|---|---|
| Reporting on sex and gender | The results presented in this study derive from data collected through nine consecutive survey rounds of the Rakai Community Cohort Study (RCCS) between September 2003 and May 2018. Participants self-reported their gender, birth date, and age at visit. |
| Population characteristics | Following consent, participants reported on demographics, behavior, health, and health service use. All participants were offered free voluntary counseling and HIV testing as part of the survey. Rapid tests at the time of the survey and confirmatory enzyme immunoassays were performed to determine HIV status. All participants were provided with pre-test and post-test counseling, and referrals of individuals who were HIV-positive for ART. Additionally, all consenting participants, irrespective of HIV status, were offered a venous blood sample for storage/future testing, including viral phylogenetic studies. Table S1 summarises the characteristics of the RCCS participants and HIV-positive participants by age and gender. Within the RCCS, we also performed population-based HIV deep-sequencing spanning a period of more than 6 years, from August 2011 to April 2018. The primary purpose of viral deep sequencing was to reconstruct transmission networks and identify the population-level sources of infections, thus complementing the data collected through the incidence cohort. Participants are characterised in Supplementary Table S1. |
| Recruitment | For each survey round, the RCCS did a household census, and subsequently invited all individuals that were of age 15-49 years and residents for at least 1 month to participate in the open, longitudinal RCCS survey. Eligible individuals first attended group consent procedures, and individual consent was obtained privately by a trained RCCS interviewer. While our analyses accounted for participation biases by age, gender and community we cannot rule out the possibility that other unmeasured factors associated with age and gender and also HIV serostatus, viral load suppression, and onward transmission may have been related to study participation, potentially biasing results. |
| Ethics oversight | The study was independently reviewed and approved by the Ugandan Virus Research Institute, Scientific Research and Ethics Committee, protocol GC/127/13/01/16; the Ugandan National Council of Science and Technology; and the Western Institutional Review Board, protocol 200313317. All study participants provided written informed consent at baseline and follow-up visits using institutional review board approved forms. This project was reviewed in accordance with CDC human research protection procedures and was determined to be research, but CDC investigators did not interact with human subjects or have access to identifiable data or specimens for research purposes. Participants in the RCCS received 10,000UGX (approximately 2.50USD) in compensation for the baseline and follow-up surveys. |

Note that full information on the approval of the study protocol must also be provided in the manuscript.

# Field-specific reporting

Please select the one below that is the best fit for your research. If you are not sure, read the appropriate sections before making your selection.

☐ Life sciences ☒ Behavioural & social sciences ☐ Ecological, evolutionary & environmental sciences

For a reference copy of the document with all sections, see nature.com/documents/nr-reporting-summary-flat.pdf

# Behavioural & social sciences study design

All studies must disclose on these points even when the disclosure is negative.

| | |
|---|---|
| Study description | The results presented in this study derive from data collected through nine consecutive survey rounds of the Rakai Community Cohort Study (RCCS) between September 2003 and May 2018. All data collected through the surveys were of quantitative nature. |
| Research sample | Individuals that were of age 15-49 years and residents for at least 1 month in inland and fishing communities in South-central Uganda. In total, 38749 participants were enrolled. Participants are characterized by survey round, gender and age in Supplementary Table S1. Sampling was representative of the population except for individuals away for work or at school. |
| Sampling strategy | The entire eligible population was invited to participate in the RCCS; sampling was thus population-based and survey participation was voluntary. Viral sequencing was performed on plasma samples from participants with HIV who had no viral load measurement and self-reported being ART-naïve at the time of the survey, or who had a viral load measurement above 1,000 copies/mL plasma. |
| Data collection | Between September 2003 and May 2018, nine consecutive survey rounds of the Rakai Community Cohort Study (RCCS) were conducted in 36 inland communities in south-central Uganda. For each survey round, the RCCS did a household census, and subsequently invited all individuals that were of age 15-49 years and residents for at least 1 month to participate in the open, longitudinal RCCS survey; and so data collection was not randomized, and data collection was blind relative to previous interactions with individuals or any personal characteristics apart from age and residency status, and any research questions. Eligible individuals first attended group consent procedures, and individual consent was obtained privately by a trained RCCS interviewer. Following consent, participants reported in a private location, typically a tent at the survey hub, on demographics, behavior, health, and health service use. All participants were offered free voluntary counseling and HIV testing as part of the survey. Rapid tests at the time of the survey and confirmatory enzyme immunoassays were performed to determine HIV status. All participants were provided with pre-test and post-test counseling, and referrals of individuals who were HIV-positive for ART. Additionally, all consenting participants, irrespective of HIV status, were offered a venous blood sample for storage/future testing, including viral phylogenetic studies. All epidemiologic data collected through RCCS are stored in a database running Microsoft SQL server 2019 and Microsoft Access version 2016. Further details for the survey methods are described in Grabowski, M. K. et al. HIV prevention efforts and incidence of HIV in Uganda. New England Journal of Medicine 377, 2154–2166 (2017) |
| Timing | Surveys were conducted between September 2003 and May 2018. The first survey round considered in this analysis was Round 10, September 26, 2003 - November 23, 2004; followed by Round 11, February 15, 2005 - June 30, 2006; Round 12, August 30, 2006 - June 06, 2008; Round 13, June 17, 2008 - July 12, 2009; Round 14, January 18, 2010 - June 21, 2011; Round 15, August 10, 2011 - July 05, 2013; Round 16, July 08, 2013 - January 30, 2015; Round 17, February 23, 2015 - September 02, 2016; and Round 18, October 03, 2016 - May 22, 2018. |
| Data exclusions | Viral sequence data were excluded if they had not sufficient depth or length for the purpose of deep-sequence phylogenetic analyses. We required that individuals had a depth of ≥ 30 such reads over at least 3 non-overlapping 250bp genomic windows. 88 transmission pairs had to be excluded due to ethical considerations. |
| Non-participation | Non-participation rates among eligible individuals in the communities considered were as follows. Round 10: 4,569/11,976; Round 11 4,255/12,528; Round 12 4,966/13,718; Round 13 4,715/13,433; Round 14 5,165/14,828; Round 15 7,217/20,806; Round 16 7,815/21,887; Round 17 7,836/22,929; Round 18 8,216/23,269. Participation rates varied by age and gender and are described in Supplementary Table S1 and Extended Data Fig 1, and the most common reason for non-participation was being away for work or school. |
| Randomization | Participants were not allocated into experimental groups. |

# Reporting for specific materials, systems and methods

We require information from authors about some types of materials, experimental systems and methods used in many studies. Here, indicate whether each material, system or method listed is relevant to your study. If you are not sure if a list item applies to your research, read the appropriate section before selecting a response.

## Materials & experimental systems

| n/a | Involved in the study |
|-----|----------------------|
| ☒ | Antibodies |
| ☒ | Eukaryotic cell lines |
| ☒ | Palaeontology and archaeology |
| ☒ | Animals and other organisms |
| ☒ | Clinical data |
| ☒ | Dual use research of concern |

## Methods

| n/a | Involved in the study |
|-----|----------------------|
| ☒ | ChIP-seq |
| ☒ | Flow cytometry |
| ☒ | MRI-based neuroimaging |

