## [Peer Review File · Nature Microbiology]

Peer Review Information

Journal: Nature Microbiology

Manuscript Title: Longitudinal population-level HIV epidemiologic and genomic surveillance highlights growing gender disparity of HIV transmission in Uganda

Corresponding author name(s): Oliver Ratmann, Joseph Kagaayi, M Kate Grabowski

Editorial Notes:

This manuscript has been previously reviewed at another journal. This document only contains reviewer comments, rebuttal and decision letters for versions considered at Nature Microbiology. Mentions of the other journal have been redacted.

Parts of this Peer Review File have been redacted as indicated to maintain patient confidentiality.

Reviewer Comments & Decisions:

Decision Letter, initial version:

16th May 2023

Dear Oliver,

Thank you for your patience while your manuscript "Growing gender disparity in HIV infection in Africa: sources and policy implications" was under peer-review at Nature Microbiology. It has now been seen by 3 referees, whose expertise and comments you will find at the of this email. You will see from their comments below that while they find your work of interest, some important points are raised. We are

very interested in the possibility of publishing your study in Nature Microbiology, but would like to consider your response to these concerns in the form of a revised manuscript before we make a final decision on publication.

In particular, you will see that reviewer #1 raises concerns regarding the transmission analysis and estimation of age differences within the cohort. In addition, reviewer #3 raises concerns regarding biases while picking participants from the Rakai cohort.

The rest referees' reports are clear and the remaining issues should be straightforward to address.

If you have not done so already please begin to revise your manuscript so that it conforms to our Article format instructions at <http://www.nature.com/nmicrobiol/info/final-submission/>

The usual length limit for a Nature Microbiology Article is six display items (figures or tables) and 3,000 words. We have some flexibility, and can allow a revised manuscript at 3,500 words, but please consider this a firm upper limit. There is a trade-off of ~250 words per display item, so if you need more space, you could move a Figure or Table to Supplementary Information.

Some reduction could be achieved by focusing any introductory material and moving it to the start of your opening 'bold' paragraph, whose function is to outline the background to your work, describe in a sentence your new observations, and explain your main conclusions. The discussion should also be limited. Methods should be described in a separate section following the discussion, we do not place a word limit on Methods.

Nature Microbiology titles should give a sense of the main new findings of a manuscript, and should not contain punctuation. Please keep in mind that we strongly discourage active verbs in titles, and that they should ideally fit within 90 characters each (including spaces).

Please include a data availability statement as a separate section after Methods but before references, under the heading "Data Availability". This section should inform readers about the availability of the data used to support the conclusions of your study. This information includes accession codes to public repositories (data banks for protein, DNA or RNA sequences, microarray, proteomics data etc...), references to source data published alongside the paper, unique identifiers such as URLs to data repository entries, or data set DOIs, and any other statement about data availability. At a minimum, you should include the following statement: "The data that support the findings of this study are available from the corresponding author upon request", mentioning any restrictions on availability. If DOIs are provided, we also strongly encourage including these in the Reference list (authors, title, publisher (repository name), identifier, year). For more guidance on how to write this section please see: <http://www.nature.com/authors/policies/data/data-availability-statements-data-citations.pdf>

To improve the accessibility of your paper to readers from other research areas, please pay particular attention to the wording of the paper's opening bold paragraph, which serves both as an introduction and as a brief, non-technical summary in about 150 words. If, however, you require one or two extra sentences to explain your work clearly, please include them even if the paragraph is over-length as a result. The opening paragraph should not contain references. Because scientists from other sub-disciplines will be interested in your results and their implications, it is important to explain essential but specialised terms concisely. We suggest you show your summary paragraph to colleagues in other fields to uncover any problematic concepts.

If your paper is accepted for publication, we will edit your display items electronically so they conform to our house style and will reproduce clearly in print. If necessary, we will re-size figures to fit single or double column width. If your figures contain several parts, the parts should form a neat rectangle when assembled. Choosing the right electronic format at this stage will speed up the processing of your paper and give the best possible results in print. We would like the figures to be supplied as vector files - EPS, PDF, AI or postscript (PS) file formats (not raster or bitmap files), preferably generated with vector-graphics software (Adobe Illustrator for example). Please try to ensure that all figures are non-flattened and fully editable. All images should be at least 300 dpi resolution (when figures are scaled to approximately the size that they are to be printed at) and in RGB colour format. Please do not submit Jpeg or flattened TIFF files. Please see also 'Guidelines for Electronic Submission of Figures' at the end of this letter for further detail.

Figure legends must provide a brief description of the figure and the symbols used, within 350 words, including definitions of any error bars employed in the figures.

Please include a statement before the acknowledgements naming the author to whom correspondence and requests for materials should be addressed.

Finally, we require authors to include a statement of their individual contributions to the paper -- such as experimental work, project planning, data analysis, etc. -- immediately after the acknowledgements. The statement should be short, and refer to authors by their initials. For details please see the Authorship section of our joint Editorial policies at http://www.nature.com/authors/editorial_policies/authorship.html

- * include a point-by-point response to any editorial suggestions and to our referees. Please include your response to the editorial suggestions in your cover letter, and please upload your response to the referees as a separate document.
- * ensure it complies with our format requirements for Letters as set out in our guide to authors at www.nature.com/nmicrobiol/info/gta/
- * state in a cover note the length of the text, methods and legends; the number of references; number and estimated final size of figures and tables
- * resubmit electronically if possible using the link below to access your home page:

[redacted]

*This url links to your confidential homepage and associated information about manuscripts you may have submitted or be reviewing for us. If you wish to forward this e-mail to co-authors, please delete this link to your homepage first.

Please ensure that all correspondence is marked with your Nature Microbiology reference number in the subject line.

Nature Microbiology is committed to improving transparency in authorship. As part of our efforts in this direction, we are now requesting that all authors identified as 'corresponding author' on published papers create and link their Open Researcher and Contributor Identifier (ORCID) with their account on the Manuscript Tracking System (MTS), prior to acceptance. This applies to primary research papers only. ORCID helps the scientific community achieve unambiguous attribution of all scholarly contributions. You can create and link your ORCID from the home page of the MTS by clicking on 'Modify my Springer Nature account'. For more information please visit www.springernature.com/orcid.

We hope to receive your revised paper within three weeks. If you cannot send it within this time, please let us know.

Yours sincerely,

[redacted]

Reviewer Expertise:

Referee #1: Epidemiology, Sequencing, HIV

Referee #2: Clinical, HIV

Referee #3: Demographics, HIV

Reviewers Comments:

Reviewer #1 (Remarks to the Author):

Monod and colleagues present a thorough investigation of partnerships and viral suppression across a 15-year cohort in Uganda. These are important data and findings, which shed light on a changing HIV epidemic East Africa. I commend the authors for the scope and thoroughness of this study. The findings regarding the focus on viral suppression in men in East Africa are of particular note and novelty.

I have some concerns about the partner age analysis. First, it is not clear how age differences are calculated. Are these differences in age at enrollment for each round or differences in birth year. Birth year would be preferred, as each round lasts 2 years and differences seen are in this order of magnitude. Extended Data Figure 4 places source-recipient pairs in at specific points in time. Can this date be used to estimate the age of both partners at the time of transmission?

Also of concern is the acknowledgement in the methods that reported ages tended towards multiples of 5, indicating a high degree of imprecision and the potential for different biases in reported age among different age-cohorts and by sex. Given that the difference found between women and men in transmission pairs differed by 0-6 years, reported ages that round to multiples of 5 is potentially concerning (even when analyzing age cohorts a decade wide). I am not suggesting fault in data collection, but I am wary of policy suggestions that directly result from such ambiguous data. Unless I am misunderstanding the nature of these data and analyses, I would suggest the authors more appropriately couch these age related findings

I commend the authors for explicitly acknowledging the limitations of their genomic epidemiological approach for inferring directionality between individuals and highlighting the importance of additional sources of data to draw conclusions. That said, I think their own results presented here suggest even this level of inference may be overstated. Specifically, the [redacted] who are appropriated excluded from the analysis suggest that the remaining 227 [redacted] pairs almost certainly include individuals who are not actually transmission partners. If nearly 20% of inferred partner pairs are demonstrably false, what are to we make the rest?

A sensitivity analysis using the more biologically plausible viral threshold for suppression that would preclude onward transmission (<200copies/mL) would be helpful.

Minor comments:

“aged 0-6 years old” is a cumbersome turn of phrase.

Using the term “evolve” to describe the HIV epidemic as part of a study that employs genomic epidemiology is confusing.

Line 120, directed transmission networks are inferred or estimated, not “measured”.

Line 292, “biological susceptibility” should specify heterosexual contact.

For stylistic purposes, there is no need to start paragraphs with declarative sentences like “Our study had several strengths.” and “This study had important limitations.” Please remove.

Figure 2b. “Aig 2011” should be “Aug 2011”

Reviewer #2 (Remarks to the Author):

Reviewer's Comments

In this manuscript Monod et.al present a compelling study of changing HIV transmission dynamics by age and gender in the Rakai community cohort in Uganda. Using a combination of longitudinal population surveys, deep sequencing, phylogenetics and mathematical modelling they demonstrate the evolving transmission dynamics for HIV in this community cohort highlighting the shift in transmission flows. Their main results indicate that men with unsuppressed HIV viremia now contribute most to new HIV infections in this cohort marking a shift over time between 2003 and 2018. Furthermore, their mathematical models suggest that interventions which reduce the gender gap in viral suppression could reduce HIV incidence in women significantly. These findings have important implications for policies which aim to achieve the 90-90-90 targets for the HIV epidemic set by the WHO. I will like to commend the authors for very well done and rigorous study, which they have presenting a clear and very well written manuscript. It was a joy to read.

Please see below comments from this reviewer for your consideration to clarify some areas of your study and hopefully improve the overall quality of this excellent piece of work:

Major comments:

Methods:

1). Modelling and analysis - These are very well explained by the authors however the model assumes that the risk of HIV infection is constant over the course of a survey round which may not be realistic in practice and should be acknowledged as a limitation. Additionally, the model does not account for potential confounding variables which may influence risk of HIV infection e.g. substance use, alcohol abuse, this limitation should also be acknowledged.

2). The authors' use of deep-sequencing and phylogenetic analyses is appropriate for examining HIV transmission dynamics. However, in Table S6, the authors provide only the minimum sampling criteria for sequencing and do not describe how the deep sequencing data was generated. It is unclear how quality and coverage of the sequences were assessed. These Important details on the phylogenetic analyses should be provided.

3) I may have missed this in the supplementary documents but I did not see anywhere in the manuscript an explanation for why the present analyses focuses mainly on inland communities.

What was the rationale for this and does this approach significantly change any of the key findings?

Results

4) In line 177-181 the authors state that they did not further examine [redacted] and provide a logical explanation for this decision. However, I wonder why they chose not to further examine [redacted]. Although [redacted] transmission may be predominant in this cohort, I think they may be an opportunity to glean information from [redacted] on possible [redacted] in this cohort. In settings were [redacted] this is unlikely to be captured in [redacted] and phylogenetic analyses may provide some insight into this.

Discussion

5). The authors make a compelling case for prioritizing targeted interventions to improve virus suppression rates in men as a way of reducing HIV incidence in women and closing the gender gaps in infection burden. I agree that this is of high importance and should inform policies on HIV incidence reduction strategies. An aspect that I found missing in the discussion of areas where such targeted interventions could be directed was PreP and how this could potentially influence transmission flows. The focus of the modelling analysis and subsequent reported results is on virus suppression post HIV infection. PreP is a highly effective intervention which reduces HIV acquisition significantly yet access and uptake in most of sub-saharan Africa remains abysmally low. This is certainly an underutilized intervention in this population which has the potential to influence transmission flows and reduce HIV incidence in men and women. It could be informative to explore the potential impact of PreP on transmission flow and HIV incidence where possible by incorporating this variable into the model if and where possible or at least mention this in the discussion

Minor Comments

-The results section uses different units for reporting HIV incidence rates (per 100 person-years vs per 1000 person-years). Please use consistent units throughout the sections for clarity and ease of comparison. Line 132 and 142.

-Please harmonize choice of spelling throughout the manuscript for consistency i.e. UK or US English. For example; “behaviour” in UK English and “behavior” in US English.

-I appreciate the authors sharing the raw code used to generate the figures and making this publicly available in a GitHub repository. The sections of the raw R script could be commented out with hashtags (#) to explain what each code chunk is doing. Without these comments it is difficult to follow details of the analysis, the written functions, loops, and iterations. Creating a separate R script that concatenates the different R scripts would show a consistent flow of the analysis and enhance its readability and reproducibility. An Rmarkdown file might be a useful alternative the authors might want to consider.

Reviewer #3 (Remarks to the Author):

This is a comprehensive and sophisticated analysis that is clearly explained and interpreted. There is so much information to convey within a tight word limit that much of the detail has been given in the extensive supplementary material.

There are necessarily a number of assumptions and approximations needed to produce the estimates presented here and most of these are articulated and sensitivity analyses done to support the assumptions used.

The findings are well supported by the analyses presented but I do have a substantive concern. A key assumption is that the people whose data were included in the transmission cohort are the same as those who were not and therefore that the observed transmissions are a good representation of all transmissions.

This isn't fully discussed in one place, and isn't given much attention in the main paper. It should be in the main paper – the central analysis is thorough and clearly presented but the interpretation of the findings rests heavily on the assumption that this sub-sample has accurately captured the infection dynamics in the entire Rakai population. It is therefore important to demonstrate how the subset of participants included in this analysis compares to the wider population and to do this in a non-technical way.

Based on what is presented in the paper and supplementary material, this is only partially achieved by ensuring the transmission flows model fits the observed patterns of incidence by age and sex and including a detection probability.

Firstly, unless I have misunderstood, the observed recipients in the transmission flows analysis could be a mix of observed incident cases and people who were only ever observed as prevalent infections in the RCCS data. Therefore there is potentially a mismatch when adjusting all transmission flow recipients to match the age/sex characteristics of only the observed incident cases rather than weighting the data from PLHIV who were sequenced to all PLHIV.

Secondly, fitting the age and sex distribution assumes that the PLHIV who were not included in the transmissions flows analysis had the same transmission patterns as those who were. If there is a selection bias for the transmission flow analysis then this assumption could be incorrect.

A selection bias seems almost inevitable to me, but there is no information to assess whether or not it really matters.

My concern is that, because sequencing isn't possible for those with a low viral load, those people are likely to be omitted from the transmission flows analysis but they could still be an important part of the transmission network prior to initiation of treatment. The transmission flows analysis can identify historic transmissions, even from prevalent cases within RCCS, but only so long as both source and recipient remained untreated for long enough to be sampled for sequencing.

The chances of ending up in the transmission flows analysis is therefore dependent on ART use of both parties. Truncation by ART use is thus informative in this instance because it is associated both with the

likelihood of transmission of infection and the chance of being included in the analysis. We know that both transmission and ART uptake and adherence varies by age and sex. If most transmissions are from people who remain untreated for longer than average (those who have the highest chance of ever being sequenced) then few transmission events will be missed and the bias would be minimal – providing that the recipients of infection are also always sampled before starting ART.

However there could be a bias if a significant proportion of transmissions occur between infection and treatment in people who start treatment promptly because those will be under-represented in this sample. Or even where most transmission are from people who remain untreated for a long time but where the recipients of infection are promptly treated and thus less likely to be included in the analysis. If time from infection to treatment varies by age, sex and sexual behaviour, then the transmissions from one person with delayed treatment initiation to another who also delays treatment initiation will be over-represented in this sample compared to the population. This could introduce a bias because characteristics that are associated with prompt treatment are also associated with transmission of infection.

To support the generalisation of the results obtained in the transmission flows analysis to the Rakai population I think some more information and discussion is needed to acknowledge the possibility of a selection bias, and to refute it.

I would like to see information on those whose data weren't included in the transmission cohort because they were not eligible for sequencing. I couldn't work out from the methods or the results how the number sequenced in each round was obtained from all the people who tested positive in that round.

From the extended methods it seems that everyone who tested positive in the round were considered for sequencing and the criteria are given for rounds 15-16 and 17-18 but I couldn't see the criteria for the preceding rounds. I think it would be useful to include a flowchart for each sequencing protocol, showing the numbers excluded at each stage, including the reasons for each exclusion i.e.

Present in census -> aged 15-49 and resident ≥ 1 month -> surveyed-> agreed to HIV testing -> provided venous blood sample -> tested HIV positive -> reported ART use/non-use -> eligible for sequencing -> sequenced.

I would also like to see how the characteristics of those included in the transmission flows analysis compare to those who were not included. It would be important to present this by calendar year or round of first positive test, even if sequencing was done at a later date, so that each person is only included once. A table showing the number eligible, number surveyed, number tested for HIV, number HIV+, number where seroconversion was observed in RCCS, number on ART (newly positive and others) and number (eventually) sequenced, for each survey round and by age, sex and number of sexual partners.

Within the RCCS data it would be possible to directly examine the chances of inclusion in the analysis, for example with a competing risks analysis following people from first HIV positive test to first report of

ART or sequencing and see if that varies by key characteristics since if it doesn't there is much less likely to be a bias.

Sensitivity analysis was done for right censoring of data leading to missed transmission events. But, incomplete sequencing coverage of all PLHIV is likely to be a bigger issue: cohort participation is high and incidence relatively low so the number of new transmissions not yet observed is probably in the tens, whereas the existing PLHIV with no sequence obtained is in the 100s meaning that there are many more unobserved transmission pairs than observed.

Minor points

- Supplementary Table S6- footnote says "Totals by round include individuals seen in other rounds.". This is ambiguous- the total isn't cumulative, is it people who were sequenced in each round but could have been first identified as HIV+ in previous rounds?
- Supplementary material line 777: Smoothing of ages- why not just use date of birth to get an coherent and precise age for each round instead of smoothing each survey round independently, which could introduce error, a single date of birth can be derived for each individual, ensuring that repeat participants age appropriately across rounds. I wasn't sure where the smoothed age estimates were used so it's hard to know if this would have an impact on the results.
- Supplementary material line 444 says "We do not know the impact of decreasing sequence coverage over time on our analyses." but Table S6 shows increasing coverage over time- is this referring to the decreasing coverage when eligibility is taken into account?
- Supplementary Table S9 is referred to in the results (main manuscript line 203) and to support the statement in the discussion (line 449) but its contents are not explained. What does it show? I don't think it supports the statement and the reference should instead be to Extended Data Fig. 6.
- The comparison with the empirical data shows that the models are underestimating the number of transmission events- is this significant? And is this based on the entire RCCS population?
- Sp table 6. what is the relevance of ART use at first visit- presumably some people who said no ART then were still not sequenced as they were on ART by the time it was available to them. The n column for those sequenced is misaligned
- The numbers in the incidence cohort numbers don't match previous publication (<https://www.nejm.org/doi/full/10.1056/nejmoa1702150>). Can understand the cohort getting larger as more survey rounds are completed and more people have a subsequent test and can be included but in the earlier rounds the data used in this paper includes fewer individuals than in the previous paper.
- The bottom right panel in Figure 2 (c, although it isn't labelled) combines a lot of different information and there is no legend for the line graph. Would be better to separate this from the bars and put a legend next to the graph. The bar graphs take longer to read than just giving the two numbers with CI and I think that would be a more straightforward way to present that information.
- Extended Data Fig. 9: panel c has no legend and could be more clearly explained

Author Rebuttal to Initial comments

26 October 2023

Imperial College
London

Dear reviewers and Dr [redacted]

Thank you for the opportunity to revise our study "*Growing gender disparity in HIV infection in Africa: sources and policy implications*" for consideration of publication in Nature Microbiology as Article, as well as your kind support throughout the revision process. Please find our point to point response to your comments on the first version of our manuscript below.

Reviewer Expertise:

Referee #1: Epidemiology, Sequencing, HIV

Referee #2: Clinical, HIV

Referee #3: Demographics, HIV

Reviewers Comments:

Reviewer #1 (Remarks to the Author):

14. Monod and colleagues present a thorough investigation of partnerships and viral suppression across a 15-year cohort in Uganda. These are important data and findings, which shed light on a changing HIV epidemic East Africa. I commend the authors for the scope and thoroughness of

this study. The findings regarding the focus on viral suppression in men in East Africa are of particular note and novelty.

Our response: We thank your reviewer for their time, critical review, and positive evaluation.

15. I have some concerns about the partner age analysis. First, it is not clear how age differences are calculated. Are these differences in age at enrollment for each round or differences in birth year. Birth year would be preferred, as each round lasts 2 years and differences seen are in this order of magnitude. Extended Data Figure 4 places source-recipient pairs in at specific points in time. Can this date be used to estimate the age of both partners at the time of transmission?

Our response: Thank you for raising this important query. For each source-recipient pair we estimate the calendar time of infection. We also know the birth date of individuals participating in the cohort. Together, this allows us to calculate the age of both individuals at the time of the transmission event. We have clarified the main text as follows (line 148):

“We further estimated the likely infection date from deep-sequence data (Methods), which enabled us to place the source-recipient pairs in calendar time and consider their age at the time of infection”

We further clarified the Methods section, subparagraph “Infection time estimates” (line 719):

“The shape and depth of an individual's subgraph in deep-sequence phylogenies also provide information on the time since infection, and since the sequence sampling date is known thus also on the infection time and the age of both individuals at the time of the infection event.”

The legends of Extended Data Figures 4 and 5 noted “age at infection”, and no further changes were made.

16. Also of concern is the acknowledgement in the methods that reported ages tended towards multiples of 5, indicating a high degree of imprecision and the potential for different biases in reported age among different age-cohorts and by sex. Given that the difference found between

women and men in transmission pairs differed by 0-6 years, reported ages that round to multiples of 5 is potentially concerning (even when analyzing age cohorts a decade wide). I am not suggesting fault in data collection, but I am wary of policy suggestions that directly result from such ambiguous data. Unless I am misunderstanding the nature of these data and analyses, I would suggest the authors more appropriately couch these age related findings

Our response: This is a thoughtful query. There are two processes here. First, during census, household heads are asked to report the age of all household members. Second, those individuals who consent to participate report either their birth date or their current age, and documentary evidence such as their family book or baptism certificate is requested though not always present. We observed age heaping on the census data reported by household heads (our Extended Data Fig 1a-b), but not on the data reported by participants (Figure R1 and R2 below).

We clarified the Methods section, subparagraph "Population size estimates" as follows (line 417):

"The age reported by household heads in the census surveys tended to reflect grouping patterns towards multiples of 5, suggesting that individuals reported their age only approximately."

We clarified the Methods section, subparagraph "Participation rates" as follows (line 428):

"Following consent, participants reported either their birth date or current age themselves, and accompanying documentary evidence was requested. There were no obvious age grouping patterns of multiple of 5 among participants."

Please see the following two figures Fig. R1 and R2.

nature portfolio

Figure R1. Number of RCCS participants by gender and self-reported age at time of survey. Numbers in black indicate year of age to facilitate visual detection of potential age hearing effects; none were identified.

Figure R2. Number of RCCS participants with HIV by gender and self-reported age at time of survey. Numbers in black indicate year of age to facilitate visual detection of potential age heaping effects; none were identified.

- I commend the authors for explicitly acknowledging the limitations of their genomic epidemiological approach for inferring directionality between individuals and highlighting the importance of additional sources of data to draw conclusions. That said, I think their own results presented here suggest even this level of inference may be overstated. Specifically, the [redacted] who are appropriated excluded from the analysis suggest that the remaining 227 [redacted] pairs almost certainly include individuals who are not actually transmission partners. If nearly 20% of inferred partner pairs are demonstrably false, what are to we make the rest?

Our response: Thank you for raising this point. [redacted] (Ratmann et al, Nat Comm 2019; Hall et al medrxiv 2022; Zhang et al. CID 2020).

Importantly, please note that as stated in the main text, section “The proportion of transmissions from men is increasing” (line 151):

“Deep-sequence phylogenetics cannot prove direction of transmission between two persons^{34,42,43}, but in aggregate these data are able to capture HIV transmission flows by age and gender over time at a population level^{29,44}.”

In our analysis here, please note further as stated in the Methods, section “Longitudinal viral phylogenetic transmission cohort” (line 76off):

“The resulting source-recipient pairs were checked further against sero-history data from both individuals where available. If sero-history data indicated the opposite direction of transmission, the estimated likely direction of transmission was set to that indicated by sero-history data.”

which we expect further improves our inferences of transmission direction among [redacted].

Another point to consider is that where possible, phylogenetic analyses done using HIV consensus sequences have so far returned results consistent with those obtained from phylogenetic deep-sequence analyses and source-recipient pairs, for example see Ratmann et al. Lancet HIV 2020 and Bbosa et al. Scientific Reports 2019.

Finally, we would like to note in response to the concern raised that we recently developed an alternative transmission flow estimation procedure that considers the transmission direction status as unknown latent variable (<https://arxiv.org/abs/2302.11567>). In this approach, transmission direction is not determined by the thresholds used in our analysis here, but instead inferred from the linkage and direction scores for each source-recipient pair and, importantly, all other source-recipient pairs with similar age characteristics also provide information on the latent linkage and transmission direction status of a given pair. In effect, for some pairs the posterior probabilities of their inferred linkage and/or transmission direction status are different to those determined by the thresholding approach in our analysis here. Reassuringly, we found the inferred population-level transmission flows are essentially the same under this alternative approach, and we believe across the PANGEA-HIV consortium that the phylogenetically reconstructed source-recipient pairs provide valid insights into transmission dynamics at a population level.

No changes were made to the manuscript.

18. A sensitivity analysis using the more biologically plausible viral threshold for suppression that would preclude onward transmission (<200copies/mL) would be helpful.

Our response: Thank you for this important comment. This analysis was performed (please see the very end of the Methods section) and the findings are reported in table S11, last row. To recap from the Methods section, paragraph "*Sensitivity in counterfactual intervention impacts to lower viral suppression thresholds*" (line 968):

"We found slightly smaller gender gaps in viral suppression at the lower threshold and the predicted incidence reduction in women in the counterfactual that assessed closing the suppression gap in men was around 45%, and all other findings remained insensitive."

No changes were made to the manuscript.

Minor comments:

19. "aged 0-6 years old" is a cumbersome turn of phrase.

Our response: Thank you for this comment. We discussed alternatives to our wording "0-6 years older" across co-authors and the PANGEA-HIV steering committee. The consensus view was that the current wording was the clearest over the alternatives "up to 6 years older" or "<7 years older".

We thus propose to leave as is but are happy to consider further suggestions.

20. Using the term "evolve" to describe the HIV epidemic as part of a study that employs genomic epidemiology is confusing.

Our response: Thank you for this helpful suggestion. We replaced all occurrences of "evolved" with "changed". For example, in the abstract:

"Here, we integrated population-based surveillance and longitudinal deep-sequence viral phylogenetics to assess how HIV incidence and the population groups driving transmission have changed over a 15 year period from 2003 to 2018 in Uganda."

21. Line 120, directed transmission networks are inferred or estimated, not "measured".

Our response: Thank you, and we agree. We changed the sentence to (line 107)

"This enabled us to infer directed transmission networks across age and gender, but with a primary focus on the time trends in infection dynamics and transmission networks during mass scale-up of HIV services in Africa"

22. Line 292, "biological susceptibility" should specify heterosexual contact.

Our response: Thank you. We modified the relevant sentence to (line 241)

"These findings are compatible with generally higher viral load in men than women that are expected to lead to higher transmission rates per sex act from men than women, heterogeneous contact patterns, higher biological susceptibility of women to HIV infection in heterosexual contacts, but also lower susceptibility of men to HIV infection following voluntary medical male circumcision."

23. For stylistic purposes, there is no need to start paragraphs with declarative sentences like "Our study had several strengths." and "This study had important limitations." Please remove.

Our response: Thank you for this comment. We removed "Our study had several strengths." but prefer to keep "This study had important limitations" as this signposting has been helpful in the previous publications.

24. Figure 2b. "Aig 2011" should be "Aug 2011"

Our response: Thank you for this comment. We have fixed this typo.

Reviewer #2 (Remarks to the Author):

Reviewer's Comments

25. In this manuscript Monod et.al present a compelling study of changing HIV transmission dynamics by age and gender in the Rakai community cohort in Uganda. Using a combination of longitudinal population surveys, deep sequencing, phylogenetics and mathematical modelling they demonstrate the evolving transmission dynamics for HIV in this community cohort highlighting the shift in transmission flows. Their main results indicate that men with unsuppressed HIV viremia now contribute most to new HIV infections in this cohort marking a shift over time between 2003 and 2018. Furthermore, their mathematical models suggest that interventions which reduce the gender gap in viral suppression could reduce HIV incidence in women significantly. These findings have important implications for policies which aim to achieve the 90-90-90 targets for the HIV epidemic set by the WHO. I will like to commend the authors for very well done and rigorous study, which they have presenting a clear and very well written manuscript. It was a joy to read. Please see below comments from this reviewer for your consideration to clarify some areas of your study and hopefully improve the overall quality of this excellent piece of work:

Our response: We thank your reviewer for their time, critical review, and this positive and helpful evaluation.

Major comments:

Methods:

26. Modelling and analysis - These are very well explained by the authors however the model assumes that the risk of HIV infection is constant over the course of a survey round which may not be realistic in practice and should be acknowledged as a limitation.

Our response: Thank you for raising this point, and we apologise for not having provided more detail in the first place to avoid confusion. The RCCS survey area comprises 36 communities which are visited in turn during each survey round, and so the visit times and dates at which individuals were last seen HIV negative and first seen HIV positive are structured by community ID. It is possible to formulate a model for estimating HIV incidence rates in continuous calendar time, but such a model would need to be carefully specified to account for confounding across communities, while also bearing in mind that population sizes are small in some communities. For these reasons, we modelled age- and gender-specific incidence dynamics by survey round at the area level.

We added the following final sentence to the Methods section, paragraph "Data and outcomes from the incidence cohort" (line 554):

"The primary statistical objective was to estimate longitudinal age-specific HIV incidence rates by 1-year age bands across (discrete) survey rounds, separately for each gender. We used a log-link mixed-effects Poisson regression model, with individual-level exposure times specified as offset on the log scale, common baseline fixed effect, and further random effects. The random effects comprised a one-dimensional smooth function on the age space, a one-dimensional smooth function on the survey round space, and an interaction term between age and survey round. The functions were specified as one-dimensional Gaussian processes, similar as in the model for estimating HIV prevalence. Alternative specifications, including two-dimensional functions over the participant's age and survey round, and without interaction terms between age and survey rounds were also tried. We did not consider incidence trends in continuous calendar time because study communities were surveyed in turn, and so the incidence data within each round are structured by communities, which would require further modelling assumptions to account for."

27. Additionally, the model does not account for potential confounding variables which may influence risk of HIV infection e.g. substance use, alcohol abuse, this limitation should also be acknowledged.

Our response: We thank the reviewer for raising this query. Our reviewer is absolutely right that incidence risk and transmission risk depend on a large number of additional factors beyond age and gender, which we now acknowledge in the Results section, paragraph “Men contribute more to transmission than population viral load suggests” (line 223):

“These findings are compatible with generally higher viral load in men than women that are expected to lead to higher transmission rates per sex act from men than women, heterogeneous contact patterns, higher biological susceptibility of women to HIV infection in heterosexual contacts, but also lower susceptibility of men to HIV infection following voluntary medical male circumcision.”

This noted, the primary aim of this analysis is to quantify age and gender specific transmission flows at population level, rather than a characterisation of risk factors. We are also explicit in the Discussion that of many evolving factors influence transmission, but here we focus on age and gender specific transmission flows (line 292):

“Here, we combined population-based incidence with deep-sequence viral phylogenetic surveillance data to characterize how HIV incidence and transmission sources have been changing by age and gender in a typical rural and semi-urban African setting.”

Due to lack of space, we opted to not elaborate further on these points in the Limitations section.

28. The authors' use of deep-sequencing and phylogenetic analyses is appropriate for examining HIV transmission dynamics. However, in Table S6, the authors provide only the minimum sampling criteria for sequencing and do not describe how the deep sequencing data was generated. It is unclear how quality and coverage of the sequences were assessed. These important details on the phylogenetic analyses should be provided.

Our response: Thank you for this assessment, and your query. We provide a summary on how the deep-sequencing data were generated in the Methods section, paragraph “Data from the transmission cohort”. In particular, for R15 and R16 (line 619ff):

“Plasma samples were shipped to University College London Hospital, London, United Kingdom, for automated RNA sample extraction on QIASymphony SP workstations with the QI- Asymphony DSP Virus/ Pathogen Kit (Cat. No. 937036, 937055; Qiagen, Hilden, Germany), followed by one-step reverse transcription polymerase chain reaction (RT- PCR)⁸⁹. Amplification was assessed through gel electrophoresis on a fraction of samples, and samples were shipped to the Wellcome Trust Sanger Institute, Hinxton, United Kingdom for HIV deep-sequencing on Illumina MiSeq and HiSeq platforms in the DNA pipelines core facility.”

And further for R17 and R18 (line 634ff):

“Plasma samples were shipped to the Oxford Genomics Centre, Oxford, United Kingdom, for automated RNA sample extraction on QIASymphony SP workstations with the QIASymphony DSP Virus/ Pathogen Kit (Cat. No. 937036, 937055; Qiagen, Hilden, Germany), followed by library preparation with the SMARTer Stranded Total RNA-Seq kit v2 - Pico Input Mammalian (Clontech, TaKaRa Bio), size selection on the captured pool to eliminate fragments shorter than 400 nucleotides (nt) with streptavidin-conjugated beads⁹⁰ to enrich the library with fragments desirable for deep-sequence phylogenetic analysis, PCR amplification of the captured fragments, and purification with Agencourt AMPure XP (Beckman Coulter), as described in the veSEQ-HIV protocol⁹¹. Sequencing was performed on the Illumina NovaSeq 6000 platform at the Oxford Genomics Centre, generating 350 to 600 base pair (bp) paired-end reads. A subset of samples from survey rounds 15 to 16 with low quality read output under the PANGEA-HIV 1 procedure was re-sequenced with the veSEQ-HIV protocol.”

The references provided include further detail on the validation experiments performed for each deep-sequencing approach. In particular, the second-generation veSEQ-HIV protocol was assessed against quantitative viral load standards, uniformity in sequencing success across subtypes, homogeneous sequencing success across the genome, transmission network recovery, and drug resistance mutation calling. The primary determinant of sequencing success is viral load (Fig 4 in Ref 91 and Table 3 in Ref 92). We added to the Methods section (line 665):

“Individuals who did not have sequencing output meeting these criteria were excluded from further analysis, and these were largely individuals sequenced only in PANGEA-HIV 1, and primarily associated with low viral load samples^{91,92}”

29. I may have missed this in the supplementary documents but I did not see anywhere in the manuscript an explanation for why the present analyses focuses mainly on inland communities. What was the rationale for this and does this approach significantly change any of the key findings?

Our response: Thank you for this important point, and we apologise for not being clearer from the start. We changed the wording in the Introduction as follows (line 101):

“Specifically, we integrate 15 years of data on HIV incidence and onward transmission to show how the drivers of the African HIV epidemic are changing by age and gender. We focus on a study population aged 15 to 49 years with an HIV risk profile typical across eastern and southern Africa [...], living in 36 semi-urban and rural agrarian communities that are part of the population-based Rakai Community Cohort Study (RCCS) in south-central Uganda (Figure1a).”

This key point is also repeated at the beginning of the Discussion (line 292):

“Here, we combined population-based incidence with deep-sequence viral phylogenetic surveillance data to characterize how HIV incidence and transmission sources have been changing by age and gender in a typical rural to semi-urban African setting. We show that ...”;

and (line 373)

“...Fourth, our findings on rural and semi-urban populations may not extend to populations with different demographics, risk profiles or healthcare access, and this includes populations in urban or metropolitan areas or key populations.”

and we further modified our final paragraph as follows (line 377):

“This study demonstrates shifting patterns in HIV incidence and in the drivers of HIV infection in communities typical of rural and semi-urban East Africa, providing key data for evidence-informed policy making.”

Results

30. In line 177-181 the authors state that they did not further examine [redacted] and provide a logical explanation for this decision. However, I wonder why they chose not to further examine [redacted]. Although [redacted] transmission may be predominant in this cohort, I think they may be an opportunity to [redacted] on possible [redacted] in this cohort. In settings were [redacted] this is unlikely to be captured in [redacted] and phylogenetic analyses may provide some insight into this.

Our response: This point is challenging to address in the context of [redacted]. We emphasise [redacted]. It is of course nonetheless possible that the phylogenetically reconstructed [redacted] events, but this remains speculative without decisive support in terms of the number of such events inferred at the population level. Our ethics team reviewed these data and [redacted]

Discussion

31. The authors make a compelling case for prioritizing targeted interventions to improve virus suppression rates in men as a way of reducing HIV incidence in women and closing the gender gaps in infection burden. I agree that this is of high importance and should inform policies on HIV incidence reduction strategies. An aspect that I found missing in the discussion of areas where such targeted interventions could be directed was PrEP and how this could potentially influence transmission flows. The focus of the modelling analysis and subsequent reported results is on virus suppression post HIV infection. PrEP is a highly effective intervention which reduces HIV acquisition significantly yet access and uptake in most of sub-saharan Africa remains abysmally low. This is certainly an underutilized intervention in this population which has the potential to influence transmission flows and reduce HIV incidence in men and women. It could be informative to explore the potential impact of PrEP on transmission flow and HIV incidence where possible by incorporating this variable into the model if and where possible or at least mention this in the discussion

Our response: Thank you for this comment. PrEP is an additional tool that is highly efficacious, but of course is focused on those who are not living with HIV. So, inevitably, PrEP has to be given to considerably more people to prevent acquisition of infection compared to treating the number who need treatment regardless of the prevention benefit. Other models (e.g. Stover et al. that we quote) show that the contribution of PrEP to population reductions in HIV incidence in East and Southern Africa are likely to be modest compared to the impact of more efficient and effective testing, linkage and treatment. The steady decline in new infections in the East and Southern African regions (and in Rakai papers) that follow scale up of treatment and VMMC, but predate widespread use of PrEP are encouraging, and lend credibility to the idea that more efficient testing, linkage and treatment could

have an even greater impact. Yet, we also recognize that combination prevention approaches are more impactful than a single highly effective intervention and to recognize this, we added the following sentence to the paragraph on male-centred interventions in the Discussion (line 330):

“Retention of men with HIV in treatment and care programs could be improved through male-centered differentiated service delivery. It is well-established that improving male engagement in HIV services leads to better health for men^{69,70}. We expect additional interventions such as voluntary medical male circumcision, condom promotion, or pre-exposure prophylaxis would lead to further reductions in new cases⁷¹.”

Minor Comments

32. The results section uses different units for reporting HIV incidence rates (per 100 person-years vs per 1000 person-years). Please use consistent units throughout the sections for clarity and ease of comparison. Line 132 and 142.

Our response: Thank you for this comment. We checked our results section and could not find an instance where results are described by 1000 person years?

33. Please harmonize choice of spelling throughout the manuscript for consistency i.e. UK or US English. For example; “behaviour” in UK English and “behavior” in US English.

Our response: Thank you. We used Grammarly to check for and convert UK to US English spelling, including “behavior”, “maximize”, “modeling”, “centered”, “color”.

34. I appreciate the authors sharing the raw code used to generate the figures and making this publicly available in a GitHub repository. The sections of the raw R script could be commented out with hashtags (#) to explain what each code chunk is doing. Without these comments it is difficult to follow details of the analysis, the written functions, loops, and iterations. Creating a separate R script that concatenates the different R scripts would show a consistent flow of the

analysis and enhance its readability and reproducibility. An Rmarkdown file might be a useful alternative the authors might want to consider.

Our response: Thank you so much for going into these details, and we greatly appreciate your time. We have substantially improved the readability of our README file to reproduce our analyses (<https://github.com/MLGlobalHealth/phyloSI-RakaiAgeGender>).

Reviewer #3 (Remarks to the Author):

35. This is a comprehensive and sophisticated analysis that is clearly explained and interpreted. There is so much information to convey within a tight word limit that much of the detail has been given in the extensive supplementary material.

There are necessarily a number of assumptions and approximations needed to produce the estimates presented here and most of these are articulated and sensitivity analyses done to support the assumptions used.

Our response: We thank your reviewer for their time, critical review, and this positive and helpful evaluation.

36. The findings are well supported by the analyses presented but I do have a substantive concern. A key assumption is that the people whose data were included in the transmission cohort are the same as those who were not and therefore that the observed transmissions are a good representation of all transmissions.

This isn't fully discussed in one place, and isn't given much attention in the main paper. It should be in the main paper – the central analysis is thorough and clearly presented but the interpretation of the findings rests heavily on the assumption that this sub-sample has accurately captured the infection dynamics in the entire Rakai population. It is therefore important to demonstrate how the subset of participants included in this analysis compares to the wider population and to do this in a non-technical way.

Based on what is presented in the paper and supplementary material, this is only partially achieved by ensuring the transmission flows model fits the observed patterns of incidence by age and sex and including a detection probability.

Our response: reviewer #3 is raising a fair criticism here. Please see our detailed responses to the more detailed queries below. In summary, we have conducted and are reporting additional analyses into the sampling cascade of infection events and sources of infections, please see the Methods section paragraph Bayesian Model (line 804), and Extended Data Figure 6. These lead us to the following changes to the Limitations section in the Discussion (line 360),

“Second, we were only able to deep-sequence a fraction of all transmission events, and these may not be representative of all transmissions. We characterised sampling probabilities under the assumption that individuals or events were ever deep-sequenced at random within age- and gender strata, and found that the sampling probabilities rarely differed substantially between strata in each round (Extended Data Fig. 6), so that the estimated transmission flows were not sensitive to our sampling probability adjustments (Supplementary Table S10). Of course, these sampling adjustments are modelled and it remains possible that missing data could bias our findings.”

37. Firstly, unless I have misunderstood, the observed recipients in the transmission flows analysis could be a mix of observed incident cases and people who were only ever observed as prevalent infections in the RCCS data. Therefore there is potentially a mismatch when adjusting all transmission flow recipients to match the age/sex characteristics of only the observed incident cases rather than weighting the data from PLHIV who were sequenced to all PLHIV.

Our response: We apologise for not being clear on this point. We have dated the reconstructed infection events in calendar time. This means that the observed source-recipient events are placed into the particular survey round during which transmission is estimated to have occurred. This means that from the perspective of the recipients, we are considering incidence events and the most appropriate sampling adjustment is thus the ratio of phylogenetically reconstructed incidence events to the number of estimated incidence events in the corresponding survey round in the population at risk. Please note that because of the dating, we can calculate the age at the time of infection of both the phylogenetically likely source and recipient individuals from survey data.

We have modified the Results section, paragraph “The proportion of transmission from men is increasing” as follows (line 146):

“We further estimated the likely infection date from deep-sequence data (Methods), which enabled us to place the source-recipient pairs in calendar time and consider their age at the time of infection (Extended Data Fig. 4).”

Extended Data Fig. 4 repeats this point, and visualises how the reconstructed transmission events are placed at the time at which transmission occurred.

38. Secondly, fitting the age and sex distribution assumes that the PLHIV who were not included in the transmissions flows analysis had the same transmission patterns as those who were. If there is a selection bias for the transmission flow analysis then this assumption could be incorrect. A selection bias seems almost inevitable to me, but there is no information to assess whether or not it really matters.

Our response: The reviewer raises a fair concern, which we addressed as follows. Based on the seminal findings of the past decade that underpin $U=U$, we start by considering that individuals with suppressed virus at a given time x cannot transmit the virus. This is a subtle but important modification of the concern raised, because we largely only fail to sequence in any survey round only those participants who are suppressed at time of survey visit and cannot transmit around the visit time. With this in mind, it is clear that we are through longitudinal sequencing capturing a large proportion of infected individuals that could act as sources in the same survey round when they were found to have unsuppressed virus, or indeed any other round.

Of course, these considerations apply since the time when longitudinal sequencing started and among participants only, but they make clear that the potential for bias is smaller than we communicated in the first version of this paper (by not elaborating on the issues properly).

A further important point – especially to our editor – is that if incident cases and their sources of different ages and different gender have the same constant sampling probability, then there cannot be bias in source attribution because the constant sampling probabilities just drop out.

To quantify this issue in depth, we are as suggested by our reviewer interested in the differences in sampling probabilities between age and gender strata. From our data, we can quantify the following two sampling probabilities: first, the probability of detecting incidence events (which involves two individuals), and second the probability of sampling individuals that could potentially act as sources because they were viremic (which involve one individual). These probabilities are reported in our new Extended Data Figure 5, and differences in the sampling probabilities between age and gender strata are reported in subfigures EDF 5c and EDF 5f. It turns out that information on these sampling probabilities is sufficient for deriving more appropriate adjustment factors in our model, which we have implemented. We found our results were largely insensitive to these adjustments, which is in line with the limited heterogeneity in the modelled sampling probabilities across age and gender in each round shown in Extended Data Figure 5.

In more detail, we considered as in the first version of the paper the detection probability of infection events defined as

phylogenetically reconstructed source-recipient pairs by age a and gender g of recipients at time of infection in round x /

infection events by age a and gender g of infected individuals in round x ,

with denominator estimated through the incidence model in the entire population. These detection probabilities are calculated as before and are now visualised in Extended Data Figure 6a-c.

To address the reviewer's comment and – with our response to #37 in mind – we only need to consider potential further sampling differences in source cases beyond the detection of infection events. To this end, we approximated the unknown inclusion probability of sources,

transmission sources of age a and gender g in round x that were ever deep sequenced /

transmission sources of age a and gender g in round x

with the ratio

unsuppressed individuals of age a and gender g in round x that were ever deep sequenced /

unsuppressed individuals of age a and gender g in round x .

We can approximate the numerator with the number of individuals with HIV of age a and gender g in round x that were ever deep sequenced, assuming that individuals who could be sequenced had unsuppressed virus in all previous survey rounds and recognising that individuals could only be sequenced if they participated in the RCCS. The numbers used to approximate the numerator are reported in Supplementary Table S1. With regards to the denominator, we used suppression status among first-time participants to approximate the suppression status of non-participants, and estimates were already made and reported in Extended Data Figure 8d. The resulting inclusion probabilities of potential sources are shown in Extended Data Figure 5d-f.

We revised the text in the Methods section, paragraph “Bayesian model” (line 804), updated our statistical framework to account for the refined approach outlined above in Equations (5) to (7), as well as all analyses. Our findings are essentially the same. We have also added the following sensitivity analyses to our Methods section, paragraph Sensitivity analyses:

“Sensitivity in estimated transmission flows to modelled sampling estimates. The sampling adjustments in (6) require assumptions including that sampling is independent of infection and transmission, independent between source and recipient, at random within strata, and well approximated by approximating sources with individuals with unsuppressed virus. We repeated flow inferences without any adjustments and without adjustments for potentially unequal sampling of sources. Our primary findings were insensitive across these analyses (Supplementary Table S10).”

For convenience, we report Extended Data Figure 5 here as well:

Extended Data Fig. 5. Sampling estimates of transmission events and sources of infections. (a-c) The sampling cascade of transmission events was modeled by comparing the number of

phylogenetically reconstructed source-recipient events to the estimated number of infection events under the incidence model, by gender, age band and survey round of infected individuals. (d-f) Additional differences in source sampling were modelled by considering unsuppressed individuals as potential sources of infection, and calculating the number of unsuppressed individuals in a round that were ever deep-sequenced. Throughout, shown are the number of sampled and unsampled transmission events/possible sources, the estimated proportion of events/possible sources that were ever deep-sequenced, and log ratios of estimated proportion of events/possible sources that were ever deep-sequenced in any strata relative to the overall average across strata. Points correspond to point estimates, and linebars to 95% Agresti-Coull confidence intervals.

39. My concern is that, because sequencing isn't possible for those with a low viral load, those people are likely to be omitted from the transmission flows analysis but they could still be an important part of the transmission network prior to initiation of treatment. The transmission flows analysis can identify historic transmissions, even from prevalent cases within RCCS, but only so long as both source and recipient remained untreated for long enough to be sampled for sequencing. The chances of ending up in the transmission flows analysis is therefore dependent on ART use of both parties. Truncation by ART use is thus informative in this instance because it is associated both with the likelihood of transmission of infection and the chance of being included in the analysis. We know that both transmission and ART uptake and adherence varies by age and sex. If most transmissions are from people who remain untreated for longer than average (those who have the highest chance of ever being sequenced) then few transmission events will be missed and the bias would be minimal – providing that the recipients of infection are also always sampled before starting ART. However there could be a bias if a significant proportion of transmissions occur between infection and treatment in people who start treatment promptly because those will be under-represented in this sample. Or even where most transmission are from people who remain untreated for a long time but where the recipients of infection are promptly treated and thus less likely to be included in the analysis. If time from infection to treatment varies by age, sex and sexual behaviour, then the transmissions from one person with delayed treatment initiation to another who also delays treatment initiation will be over-represented in this sample compared to the population. This could introduce a bias because characteristics that are associated with prompt treatment are also associated with transmission of infection.

Our response: The reviewer raises a fair concern, which we have addressed as described under point #38. Please note that Rakai's sequencing program is longitudinal, and so we estimate as shown in Extended Data Figure 5e that – from the time that the sequencing program started onwards – a large proportion (50-75%) of individuals with unsuppressed virus in a given round were ever deep sequenced.

These considerations make clear that the potential for bias is smaller than we communicated in the first version of this paper (see also Extended Data Figure 5f). It is true that we cannot rule out that our modelling assumptions are inappropriate for unseen reasons, and so we also added the following sentence to the Limitations section in the Discussion (line 366),

“Of course, these sampling adjustments are modelled and it remains possible that missing data could bias our findings.”

40. To support the generalisation of the results obtained in the transmission flows analysis to the Rakai population I think some more information and discussion is needed to acknowledge the possibility of a selection bias, and to refute it. I would like to see information on those whose data weren't included in the transmission cohort because they were not eligible for sequencing. I couldn't work out from the methods or the results how the number sequenced in each round was obtained from all the people who tested positive in that round.

Our response: Thank you for these comments. We agree it is useful to show a comparison between the individuals included in the transmission flow analysis, and those who were not included.

Please see our new flowcharts in Supplementary Fig. S1 that make transparent the specific reasons for exclusion at each step towards the transmission cohort.

We also have included in our manuscript a table that shows by round, age, and gender the number of census eligible individuals, participants, etc., so that for each round each individual is counted once as Supplementary Table S1. More specifically, and to address the final point raised, (1) in R10, participants were not asked about ART status and viral loads were not measured. In R11-R14, participants reported their ART status and viral loads were not measured. In R15, participants reported both their ART status and a subset of viral loads were measured. In R16-R18, participants reported both their ART status and viral loads were measured comprehensively in participants with HIV. (2) Samples were selected for deep-sequencing from participants who had no viral load measured and reported being ART-naive or participants with viral load above 1,000 copies/mL plasma. Individuals participated across rounds, so for individuals participating in a given round, samples for sequencing could also be obtained in other rounds and we tabulate the proportion of participants ever deep-sequenced. We added these clarifications as stated here into the footnote of Supplementary Table S1.

We have furthermore updated Supplementary Table S5 and now report sequence sampling coverages by round, age and gender among participants with HIV.

41. From the extended methods it seems that everyone who tested positive in the round were considered for sequencing and the criteria are given for rounds 15-16 and 17-18 but I couldn't see the criteria for the preceding rounds.

Our response: Thank you for this query. We explained in the Results section, paragraph "The proportion of transmissions from men is increasing" the following, which due to space limitations we have now moved to the Methods section line 660:

"Deep-sequencing was performed from 2010 (survey round 14) onwards, but because sequences provide information on past and present transmission events, we also obtained information on transmission in earlier rounds and calculated sequence coverage in participants that were ever deep-sequenced at minimum quality criteria."

No changes were made to the manuscript.

42. I think it would be useful to include a flowchart for each sequencing protocol, showing the numbers excluded at each stage, including the reasons for each exclusion i.e.

Present in census -> aged 15-49 and resident ≥ 1 month -> surveyed-> agreed to HIV testing -> provided venous blood sample -> tested HIV positive -> reported ART use/non-use -> eligible for sequencing -> sequenced.

Our response: Thank you for this comment. We agree a flowchart that also makes the specific reasons for exclusion transparent is instrumental to comprehend the final sample sizes, as well as potential limitations of our analyses. We now provide these as Supplementary Fig. S1, and for ease of reference show one of these here:

Round 18

Supplementary Fig. S1: Flowchart of census eligible individuals through to individuals in the incidence and transmission cohorts.

43. I would also like to see how the characteristics of those included in the transmission flows analysis compare to those who were not included. It would be important to present this by calendar year or round of first positive test, even if sequencing was done at a later date, so that each person is only included once. A table showing the number eligible, number surveyed, number tested for HIV, number HIV+, number where seroconversion was observed in RCCS, number on ART (newly positive and others) and number (eventually) sequenced, for each survey round and by age, sex and number of sexual partners.

Our response: Thank you for this helpful suggestion. We have included tables that show by round, age, and gender the number of census eligible individuals, participants, etc., so that for each round each individual is counted once (see Supplementary Table S1 and Supplementary Table S5).

44. Within the RCCS data it would be possible to directly examine the chances of inclusion in the analysis, for example with a competing risks analysis following people from first HIV positive test to first report of ART or sequencing and see if that varies by key characteristics since if it doesn't there is much less likely to be a bias.

Our response: Thank you for this well intended suggestion. We looked at this in a slightly different way that we find is more appropriate for our transmission analysis. Please see our response to your query 38.

45. Sensitivity analysis was done for right censoring of data leading to missed transmission events. But, incomplete sequencing coverage of all PLHIV is likely to be a bigger issue: cohort participation is high and incidence relatively low so the number of new transmissions not yet observed is probably in the tens, whereas the existing PLHIV with no sequence obtained is in the 100s meaning that there are many more unobserved transmission pairs than observed.

Our response: Thank you for this query. Please see our response to your query #38.

Minor points

46. Supplementary Table S6- footnote says "Totals by round include individuals seen in other rounds.". This is ambiguous- the total isn't cumulative, is it people who were sequenced in each round but could have been first identified as HIV+ in previous rounds?

Our response: Thank you for this helpful suggestion. We have removed Supp Table S6, please see our response to query #51.

47. Supplementary material line 777: Smoothing of ages- why not just use date of birth to get an coherent and precise age for each round instead of smoothing each survey round independently, which could introduce error, a single date of birth can be derived for each individual, ensuring that repeat participants age appropriately across rounds. I wasn't sure where the smoothed age estimates were used so it's hard to know if this would have an impact on the results.

Our response: We apologise for the lack of clarity here. Reviewer #1 raised a very similar query. Please kindly see our response to question #16.

48. Supplementary material line 444 says “We do not know the impact of decreasing sequence coverage over time on our analyses.” but Table S6 shows increasing coverage over time- is this referring to the decreasing coverage when eligibility is taken into account?

Our response: We apologise for the unclear wording, and modified the Discussion section, Limitations paragraph as follows (line 360):

“Second, we were only able to deep-sequence a fraction of all transmission events, and these may not be representative of all transmissions. We characterized sampling probabilities under the assumption that individuals or events were ever deep-sequenced at random within age- and gender strata, and found that the sampling probabilities rarely differed substantially between strata in each round (Extended Data Fig. 5), so that the estimated transmission flows were not sensitive to our sampling probability adjustments (Supplementary Table S10). Of course, these sampling adjustments are modeled and it remains possible that missing data could bias our findings.”

49. Supplementary Table S9 is referred to in the results (main manuscript line 203) and to support the statement in the discussion (line 449) but its contents are not explained. What does it show? I don't think it supports the statement and the reference should instead be to Extended Data Fig. 6.

Our response: Thank you for this kind and helpful comment. This table (now Supplementary Table S8) describes the estimated contributions of age and gender groups to transmission in 2004, 2014 and 2018. It should not be referenced in line 203; however we added a reference to the table to the following statement (line 175):

“In 2003 the largest transmission flows were to women aged 15-24 years from male partners 0-6 years older (16.0% [12.8-19.3]) and from male partners 6+ years older (15.5% [12.2-18.8]) (Supplementary Table S8).”

Supplementary Table S8 also should not be referenced in line 449. We clarified the Discussion text as outlined in our response to point #48.

50. The comparison with the empirical data shows that the models are underestimating the number of transmission events - is this significant? And is this based on the entire RCCS population?

Our response: We thank our reviewer for this query. Extended Data Figure 6a shows the empirical incidence rates for each of the 50 data sets with imputed exposure times on the x-axis against the estimated HIV incidence rates under the Poisson model on the y-axis. These rates are based on the incidence cohort among RCCS participants. Because the model includes regularising smoothing splines, the outliers associated with a large number of infection events are smoothed out. The reverse is also true, the crude transmission events at 0 are estimated to be higher than 0. Model fit was good, with 98.8% of data points inside the 95% prediction intervals across all imputed data sets. There was no indication of model inconsistency by round, gender, age.

Extended Data Figure 6b shows the prior incidence rates used in the transmission model versus the posterior incidence rates obtained with the transmission model. Fit was good, demonstrating that the transmission flow model is calibrating to the estimates of the separate incidence model. Overall, the incidence rates of the fitted transmission model calibrated well around the prior incidence rates that were obtained under the incidence model, with 97.0% of the prior means within the posterior prediction intervals. There was no indication of model inconsistency by round, gender, age.

Extended Data Figure 6c shows the number of phylogenetically observed source-recipient pairs (by age band of the source) on the x-axis against the predicted number of source-recipient pairs (also by age band of the source) on the y-axis. The predictions are obtained under the fitted, generative model and the 95% prediction intervals should include the observed data in 95% of cases for the model to be well calibrated. We found that 99.57% of the observed data points are included in the 95% prediction intervals (see also Methods section, Transmission flow analysis). There was no indication of model inconsistency.

To improve clarity, we have updated the figure legend of Extended Data Fig. 6 as follows:

"Validation of the incidence rate and transmission flow models. (a) Empirical HIV incidence rates were obtained for each of the 50 data sets with imputed exposure times and compared to the estimated HIV incidence rates under the Poisson model. The median (point) and 95% range (horizontal error bars) of the crude HIV incidence rates are plotted against the posterior median (point) and 95% range (vertical error bars) of estimated HIV incidence rates for each gender, age and round. (b) Prior incidence rates as specified according to the outputs of the incidence rate and used in the transmission model are compared versus the posterior incidence rates obtained with the transmission model. Shown are medians (point) and 95% credible intervals (error bars) by gender, age and round. (c) Observed transmission flow counts are compared to posterior predictive estimates under the transmission model. Shown are medians (point) and 95% credible intervals (error bars) by direction of transmission, time period, and age of the phylogenetically likely source."

51. Supp table S6. What is the relevance of ART use at first visit - presumably some people who said no ART then were still not sequenced as they were on ART by the time it was available to them. The n column for those sequenced is misaligned

Our response: Thank you for this query. Our original intention was to report a proxy of the number of individuals that could be potential transmission sources (in the sense that those reporting not to be on ART tend to have unsuppressed virus). We find our new Extended Data Fig. 5 provides clearer insights into individuals that could be potential transmission sources. We also added the remaining information in Supplementary Table S6 to Supplementary Table S5, and deleted Supplementary Table S6 entirely.

52. The numbers in the incidence cohort numbers don't match previous publication (<https://www.nejm.org/doi/full/10.1056/nejmoa1702150>). Can understand the cohort getting larger as more survey rounds are completed and more people have a subsequent test and can be included but in the earlier rounds the data used in this paper includes fewer individuals than in the previous paper.

Our response: We would like to thank the reviewer in particular for their effort in checking the numbers that we report in our Supplementary tables. We made an indexing mistake when preparing the "Incidence cohort (n)" column in Supplementary Table S3, and this indexing error also affected the size of the incidence cohort reported in the manuscript, but not the person-years, incidence events, or estimated incidence rates.

For each survey round, the sample size of the incidence cohort considered in this analysis is always greater than that considered in Grabowski et al. NEJM (2017), for two reasons. First, in the NEJM analysis, the incidence cohort comprised HIV seroconverters with at most one missed survey round between the last HIV negative and HIV positive date, while in this analysis we relaxed this definition. In this analysis, the incidence cohort also comprised HIV seroconverters with more than one missed survey round between the last HIV negative and HIV positive date. Methodologically, this was possible as we now impute infection time events at random in a manner similar to that proposed in Vandormael et al, *Statistical Methods in Medical Research* (2019). Second, from R15 onwards we included in the main analysis of this manuscript additional communities that were not contiguously surveyed throughout the entire observation period; sensitivity analyses focused on the contiguously surveyed communities are reported in Supplementary Table S10. For R10 to R14, the communities considered in the NEJM analysis and this analysis were identical, though in this analysis we merged four spatially close communities into two as described in the Methods section.

The corrected numbers compare to those reported in the NEJM paper as follows:

	Round 10 (starting Sep 2003)	R11	R12	R13	R14	R15	R16	R17
NEJM	4693	4867	5001	5611	5742	6176	6277	7122
NEJM #communities	30	30	30	30	30	30	30	30
Ours (previously reported with indexing error)	3153	4359	5492	6544	7471	7961	8709	9077
Ours (with indexing error fixed)	7797	8513	9276	9320	10078	12451	13711	14448
ours ##communities after	28	28	28	28	28	33	35	35

merging								
---------	--	--	--	--	--	--	--	--

Along with this indexing error, we found another similar issue on the total number of unique participants. We corrected in line 112:

“From September 23, 2003 to May 22, 2018, 38,749 participants were enrolled in the Rakai Community Cohort Study. Of these participants, 22,724 tested HIV seronegative at first survey, ...”

53. The bottom right panel in Figure 2 (c, although it isn't labelled) combines a lot of different information and there is no legend for the line graph. Would be better to separate this from the bars and put a legend next to the graph. The bar graphs take longer to read than just giving the two numbers with CI and I think that would be a more straightforward way to present that information.

Our response: Thank you for this comment. Figure 2c compares the contribution of age/gender groups to transmission to the contribution of age/gender groups to unsuppressed viral load. We initially had the figure without the horizontal bar graph, and during internal review with PANGEA-HIV and RHSP it was felt that the overall contribution by gender is not clear without the bar graph. It would indeed be possible to state the two numbers in the main text. However we feel the figures should clearly communicate the main findings standing alone and for this reason we left the horizontal bars as part of Figure 2c. We added the label “c” to the figure.

54. Extended Data Fig. 9, panel c has no legend and could be more clearly explained

Our response: Thank you for this comment. We have clarified Extended Data Fig. 9 by removing panel (a) and showing instead the time trends in the male-to-female ratio of individuals with HIV who have unsuppressed virus. This in turn makes it more clear what the inset shows, and we have expanded the inset into subfigure d. We modified the figure legend as follows:

“Longitudinal changes in viral suppression and incidence rates in the RCCS study population since 2003. (a) Changes in incidence rates relative to round 10, i.e. Sep 2003 to Oct 2004 (posterior median: dots, 95% confidence interval: errorbars). (b) Female-to-male ratio in

changes in incidence rates relative to round 10 (posterior median: dots, 95% credible interval: errorbars). (c) Male-to-female ratio in changes in the proportion of individuals with HIV who have unsuppressed virus relative to round 10 (posterior median: dots, 95% credible interval: errorbars). (d) Correlation between the female-to-male ratio in changes in incidence rates as shown in (b) and the male-to-female ratio in changes in the proportion of individuals with HIV who have unsuppressed virus relative to round 10 as shown in (c)."

Decision Letter, first revision:

Our ref: NMICROBIOL-23030733A

7th September 2023

Dear Oliver and Kate,

Thank you for your e-mail and [redacted].

We have discussed the situation with our internal team, and came to an agreement that [redacted]. We suggest something along the lines: "Another limitation of our study is the exclusion of 88 participants from our study due to ethical requirements. Preliminary analysis of this data however suggested an increased number of false-positives within our analysis." [redacted].

Nevertheless, thank you for submitting your revised manuscript "Sources and policy implications of growing gender disparity in HIV infection in Africa" (NMICROBIOL-23030733A). It has been seen by the original referees and their comments are below. The reviewers find that the paper has improved in revision, and therefore we'll be happy in principle to publish it in Nature Microbiology, pending minor revisions to satisfy the referees' final requests and to comply with our editorial and formatting guidelines.

Besides the points raised by reviewer #1, please address the points from reviewer #3 by adding the information about people in both incidence and transmission cohorts to your flow diagrams, as well as mentioned it in the methods. Along the previous lines, when indicating exclusion of the 88 participants in the flow diagram, [redacted].

We are now performing detailed checks on your paper and will send you a checklist detailing our editorial and formatting requirements in about a week. Please do not upload the final materials until you receive this additional information from us.

Thank you again for your interest in Nature Microbiology. Please do not hesitate to contact me if you have any questions.

Sincerely,

[redacted]

Reviewer #1 (Remarks to the Author):

I appreciate the extensive edits made to the manuscript. It is much improved. Further, the treatment of age is more straightforward and interpretable.

I appreciate the extensive description of false discovery rates in the Response to Reviewer Comments. However, [redacted]. This section was important enough to be commented on by multiple reviewers. I think this edit, while making the manuscript a cleaner read, masks the nuance and ambiguity that is inherent in these types of phylogenetic studies. I appreciate [redacted]. But hiding these results from the reader presents a false-level of confidence in the results. And this overconfidence can have negative reverberations for this type of work [redacted]. I urge the authors to reconsider this edit and instead lean in to including the description of false-discovery rates communicated to the reviewers.

Minor Comment: I would suggest the avoiding the word “target” when describing people. The usage is acceptable when describing individual and country-level goals, etc. But targeting people has a negative connotation.

Reviewer #2 (Remarks to the Author):

I thank the authors for considering my reviewer comments and taking the time to carefully address all of them. These have been addressed to my satisfaction.

Reviewer #3 (Remarks to the Author):

The revised manuscript has addressed all the points I raised in my original review with the exception of two very minor things, which I think I didn't make clear initially.

For point 39 my question was a rather simple one. The people included in the incidence cohort (22,724) were required to have tested negative in their first study test and to have tested at least once at a later time. The transmission cohort was derived from the 2174 participants with HIV who were sequenced and these people were not required to have tested negative at some point in the past. I was wondering how many people were in both the incidence and transmission cohorts. Whilst the PLHIV who contributed samples for sequencing were all at some point incident cases, they may not have contributed to the incidence cohort observed in RCCS.

And for point 41 it would be nice just to have a sentence saying who was eligible for sequencing in round 14, to match the statements for the other rounds.

Author Rebuttal, first revision:

26 October 2023

Imperial College
London

Dear reviewers and Dr [redacted]

Thank you for the opportunity to revise our study "*Growing gender disparity in HIV infection in Africa: sources and policy implications*" for consideration of publication in Nature Microbiology as Article, as well as your kind support throughout the revision process. Please find our point to point response to your comments on the first version of our manuscript below.

Our ref: NMICROBIOL-23030733A

7th September 2023

Dear Oliver and Kate,

Thank you for your e-mail and [redacted].

We have discussed the situation with our internal team, and came to an agreement that [redacted]. We suggest something along the lines: "Another limitation of our study is the exclusion of 88 participants from our study due to ethical requirements. Preliminary analysis of this data however suggested an increased number of false-positives within our analysis." [redacted]

Nevertheless, thank you for submitting your revised manuscript "Sources and policy implications of growing gender disparity in HIV infection in Africa" (NMICROBIOL-23030733A). It has been seen by the original referees and their comments are below. The reviewers find that the paper has improved in revision, and therefore we'll be happy in principle to publish it in Nature Microbiology, pending minor revisions to satisfy the referees' final requests and to comply with our editorial and formatting guidelines.

Besides the points raised by reviewer #1, please address the points from reviewer #3 by adding the information about people in both incidence and transmission cohorts to your flow diagrams, as well as mentioned it in the methods. Along the previous lines, when indicating exclusion of the 88 participants in the flow diagram, [redacted].

We are now performing detailed checks on your paper and will send you a checklist detailing our editorial and formatting requirements in about a week. Please do not upload the final materials until you receive this additional information from us.

Thank you again for your interest in Nature Microbiology. Please do not hesitate to contact me if you have any questions.

Sincerely,

[redacted]

Our response:

Dear [redacted],

Thank you for your prompt and helpful response.

We have carefully revisited the Methods section and [redacted] and in addition as proposed added a sentence on further exclusion of 88 source-recipient pairs due to ethical considerations (great suggested wording - thank you!). We did not think that this exclusion poses a limitation to this study because the primary research objective of this manuscript is on characterising changes and trends in [redacted] transmission dynamics with massive scale-up of prevention interventions. We made the following edits:

[redacted]

We have further addressed the remaining comments as per your suggestions.

Please find attached our revised manuscript “NMICROBIOL-23030733A”, as well as a point-to-point response on the outstanding queries. We look forward to hearing back from you.

Warmly,
Oliver, Kate and Joseph on behalf of the study team

Reviewer #1 (Remarks to the Author):

I appreciate the extensive edits made to the manuscript. It is much improved. Further, the treatment of age is more straightforward and interpretable.

I appreciate the extensive description of false discovery rates in the Response to Reviewer Comments. However, [redacted]. This section was important enough to be commented on by multiple reviewers. I think this edit, while making the manuscript a cleaner read, masks the nuance and ambiguity that is inherent in these types of phylogenetic studies. I appreciate [redacted]. But hiding these results from the reader presents a false-level of confidence in the results. And this overconfidence can have negative reverberations for this type of work [redacted]. I urge the authors to reconsider this edit and instead lean in to including the description of false-discovery rates communicated to the reviewers.

Our response: Thank you for this comment. We have discussed the points raised further internally as well as with the editor [redacted]

We are committed to transparency on this point and added the following statement into the Methods Section around line 785:

“This excluded 88 possible source-recipient pairs from our study due to ethical considerations and prior analyses suggesting these pairs most likely represent partially sampled transmission chains (i.e., “false positives”).

Minor Comment: I would suggest the avoiding the word “target” when describing people. The usage is acceptable when describing individual and country-level goals, etc. But targeting people has a negative connotation.

Our response: We are grateful for this important minor comment. We modified the main text further around line 80,

“This study suggests that HIV programs to increase HIV suppression in men are critical to reduce incidence in women, close gender gaps in infection burden and improve men's health in Africa.”

around 264,

“Overall, we found slightly older men would have reached suppression in the scenarios closing the suppression gap as compared to the UNAIDS 95-95-95 scenario.”

around line 280,

“Thus, all three intervention scenarios involved”

Table 1 title:

“HIV prevalence, viral suppression, transmission sources, and impact of counterfactual interventions focussed on closing the suppression gap in men by age of male partner, round 18, Oct 2016-Apr 2018.”

Figure 4 title:

“Counterfactual modeling scenarios predicting the impact of interventions to increase HIV suppression in men on incidence reductions in women.”

Figure 4 legend:

“... Estimated additional number of men with HIV in the census-eligible population in round 18 that already had suppressed virus (light grey), those who would have achieved viral suppression in the counterfactual intervention scenarios (color), and those who would have remained with unsuppressed virus in the counterfactuals (dark grey). Posterior median: bars, 95% credible interval: errorbars. (c) Percent reduction in incidence in women of the census-eligible population in round 18 under the counterfactual targeted scenarios. Posterior median: bars, 95% credible interval: errorbars.”

Around line 840:

“We investigated ---given the inferred transmission flows--- the hypothetical impact of targeted counterfactual intervention scenarios”

Reviewer #2 (Remarks to the Author):

I thank the authors for considering my reviewer comments and taking the time to carefully address all of them. These have been addressed to my satisfaction.

Our response: Thank you for this positive evaluation.

Reviewer #3 (Remarks to the Author):

The revised manuscript has addressed all the points I raised in my original review with the exception of two very minor things, which I think I didn't make clear initially.

Our response: Thank you for this positive evaluation and please see our further responses below.

For point 39 my question was a rather simple one. The people included in the incidence cohort (22,724) were required to have tested negative in their first study test and to have tested at least once at a later time. The transmission cohort was derived from the 2174 participants with HIV who were sequenced and these people were not required to have tested negative at some point in the past. I was wondering how many people were in both the incidence and transmission cohorts. Whilst the PLHIV who contributed samples for sequencing were all at some point incident cases, they may not have contributed to the incidence cohort observed in RCCS.

Our response: We apologise that we didn't entirely follow the initial point that you were making. Following the further guidance by our editor,

- we now provide in our flow charts in Supplementary Figure S1 additional information on the number of individuals in the transmission cohort who are also in the incidence cohort;
- we added to the Methods section below line 660 the text “In total, we deep-sequenced virus from 1,980 participants with HIV of who 559 were also in the incidence cohort. Supplementary Table S5 characterizes HIV deep-sequencing outcomes in more detail.”

And for point 41 it would be nice just to have a sentence saying who was eligible for sequencing in round 14, to match the statements for the other rounds.

Our response: Thank you for this comment. We had in the first version of the manuscript indeed specified our eligibility criteria from R15 onwards. We rectified this error in the revised version of the manuscript, around line 610:

“Within the RCCS, we also performed population-based HIV deep-sequencing spanning a period of more than 6 years, from January 2010 to April 2018.”

and 620:

“For survey rounds 14 to 16 (PANGAEA-HIV 1), viral sequencing was performed,…”

and 660:

“Deep-sequencing was performed from 2010 (survey round 14) onwards, but because sequences provide information on past and present transmission events, we also obtained information on transmission in earlier rounds and calculated sequence coverage in participants that were ever deep-sequenced at minimum quality criteria.”

No changes were made to the manuscript.

Final Decision Letter:

16th October 2023

Dear Dr Ratmann,

I am pleased to accept your Article "Longitudinal population-level HIV genomic surveillance highlights growing gender disparity of HIV transmission in Uganda" for publication in Nature Microbiology. Thank you for having chosen to submit your work to us and many congratulations.

Acceptance of your manuscript is conditional on all authors' agreement with our publication policies (see <https://www.nature.com/nmicrobiol/editorial-policies>). In particular your manuscript must not be

published elsewhere and there must be no announcement of the work to any media outlet until the publication date (the day on which it is uploaded onto our website).

Please note that *Nature Microbiology* is a Transformative Journal (TJ). Authors may publish their research with us through the traditional subscription access route or make their paper immediately open access through payment of an article-processing charge (APC). Authors will not be required to make a final decision about access to their article until it has been accepted. [Find out more about Transformative Journals](https://www.springernature.com/gp/open-research/transformative-journals)

Authors may need to take specific actions to achieve [compliance](https://www.springernature.com/gp/open-research/funding/policy-compliance-faqs) with funder and institutional open access mandates. If your research is supported by a funder that requires immediate open access (e.g. according to [Plan S principles](https://www.springernature.com/gp/open-research/plan-s-compliance)) then you should select the gold OA route, and we will direct you to the compliant route where possible. For authors selecting the subscription publication route, the journal's standard licensing terms will need to be accepted, including [self-archiving policies](https://www.nature.com/nature-portfolio/editorial-policies/self-archiving-and-license-to-publish). Those licensing terms will supersede any other terms that the author or any third party may assert apply to any version of the manuscript.

nature portfolio

With kind regards,

[redacted]